# WITRAN: Water-wave Information Transmission and Recurrent Acceleration Network for Long-range Time Series Forecasting

**Yuxin Jia    Youfang Lin    Xinyan Hao    Yan Lin    Shengnan Guo    Huaiyu Wan**[*]
School of Computer and Information Technology, Beijing Jiaotong University, China
Beijing Key Laboratory of Traffic Data Analysis and Mining, Beijing, China
`{yuxinjia, yflin, xinyanhao, ylincs, guoshn, hywan}@bjtu.edu.cn`

## Abstract

Capturing semantic information is crucial for accurate long-range time series forecasting, which involves modeling global and local correlations, as well as discovering long- and short-term repetitive patterns. Previous works have partially addressed these issues separately, but have not been able to address all of them simultaneously. Meanwhile, their time and memory complexities are still not sufficiently low for long-range forecasting. To address the challenge of capturing different types of semantic information, we propose a novel Water-wave Information Transmission (WIT) framework. This framework captures both long- and short-term repetitive patterns through bi-granular information transmission. It also models global and local correlations by recursively fusing and selecting information using Horizontal Vertical Gated Selective Unit (HVGSU). In addition, to improve the computing efficiency, we propose a generic Recurrent Acceleration Network (RAN) which reduces the time complexity to $\mathcal{O}(\sqrt{L})$ while maintaining the memory complexity at $\mathcal{O}(L)$. Our proposed method, called Water-wave Information Transmission and Recurrent Acceleration Network (WITRAN), outperforms the state-of-the-art methods by 5.80% and 14.28% on long-range and ultra-long-range time series forecasting tasks respectively, as demonstrated by experiments on four benchmark datasets. The code is available at: `https://github.com/Water2sea/WITRAN`.

## 1   Introduction

Time series forecasting is a valuable tool across diverse fields, such as energy, traffic, weather and so on. Compared with short-range time series forecasting, long-range forecasting offers the advantage of providing individuals with ample time to prepare and make informed decisions. For instance, accurate long-range traffic and weather forecasts enable individuals to plan their travel arrangements and attire accordingly. To improve the accuracy of such forecasting, it is essential to utilize longer historical sequences as input for the forecasting models [Liu et al., 2021, Zeng et al., 2023].

Previous studies [Wu et al., 2023, Nie et al., 2023] have highlighted the importance of capturing semantic information in long-range time series to achieve accurate predictions. However, the semantic information in long-range time series is diverse, so how to analyze and capture it becomes a major challenge. Specifically, semantic information includes two main aspects: (1) Global and local correlations. The local semantic information usually contains short-range changes within the data, while the global semantic information reflects the long-range trends present in the time series [Wang et al., 2023], which can be referred to as global-local semantic information. (2) Long- and short-term repetitive patterns. Time series often exhibit repetitive patterns at different timescales [Lai

---

[*]Corresponding author

37th Conference on Neural Information Processing Systems (NeurIPS 2023).

et al., 2018, Liu et al., 2021], such as hourly or daily cycles, which can be identified as periodic semantic information. Furthermore, it is crucial to model both aspects of semantic information simultaneously and ensure that the modeling approach remains computationally efficient, avoiding excessive complexity.

Unfortunately, although the existing state-of-the-art methods have shown great performance, they encounter difficulties in addressing the aforementioned challenges simultaneously. Specifically, **CNN-based methods** [Bai et al., 2018, Franceschi et al., 2019, Sen et al., 2019, Wu et al., 2023, Wang et al., 2023] have linear complexity with respect to the sequence length $L$, but they are either limited by the size of the convolution receptive field [Wang et al., 2023] or constrained by the 1D input sequence [Wu et al., 2023], making it difficult to capture both important semantic information simultaneously. **Transformer-based methods** can be broadly classified into two categories based on whether the point-wise attention is used or not. The used ones [Vaswani et al., 2017, Zhou et al., 2021, 2022a] capture correlations between points in the sequence, yet face challenges in capturing hidden semantic information directly from point-wise input tokens [Nie et al., 2023, Wu et al., 2023]. The others [Li et al., 2019, Wu et al., 2021, Liu et al., 2021] struggles either with achieving sufficiently low computational complexity or with effectively capturing periodic semantic information. **Other methods** [Zhou et al., 2022b, Zeng et al., 2023] also exhibit limitations in capturing semantic information mentioned above. Further details can be found in Section 2.

**RNN-based methods** [Hochreiter and Schmidhuber, 1997, Chung et al., 2014, Rangapuram et al., 2018, Salinas et al., 2020] have significant advantages in capturing global and local semantic information through their recurrent structure, as shown in Figure 1(a), while maintaining linear complexity. However, they suffer from gradient vanishing/exploding [Pascanu et al., 2013] and information forgetting issues (refer to Appendix B for more details), making them less suitable for direct application in long-range forecasting tasks. Fortunately, it has been proven that by splitting the information transmissions into patches between a few adjacent time steps and processing them individually, it is possible to maintain both global and local semantic information [Nie et al., 2023, Wu et al., 2023]. This insight is highly inspiring, as it suggests that by dividing the inputs of RNNs into numerous subseries and processing them separately, we can effectively address the significant limitation mentioned earlier, without compromising the efficiency of long-range forecasting.

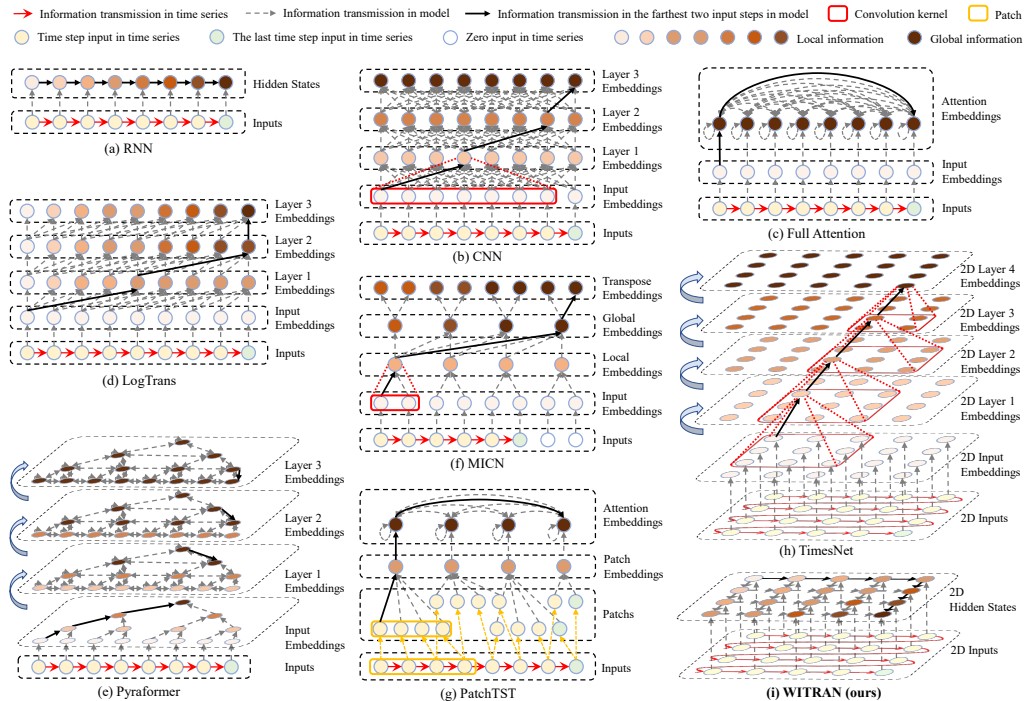

Figure 1: Information transmission process diagram of different forecasting models.

Based on the aforementioned insights, we propose a novel framework called **W**ater-wave **I**nformation **T**ransmission and **R**ecurrent **A**cceleration **N**etwork (**WITRAN**) which comprises two key components: The **W**ater-wave **I**nformation **T**ransmission (**WIT**) framework and the **R**ecurrent **A**cceleration **N**etwork (**RAN**). The overall information transmission process of WITRAN is illustrated in Figure 1(i). Firstly, to capture the periodic semantic information of long- and short-term, we rearrange the input sequences according to their natural period, as shown in Figure 2(a). This rearrangement allows for information transmission in two directions, resembling the propagation of water-wave energy, as shown in Figure 2(b). The horizontal red arrows indicate the intra-periodic information transmission along the time steps, while the vertical blue arrows indicate the inter-periodic information transmission. Secondly, to preserve the characteristics of long- and short-term periodic semantic information, we propose a novel **H**orizontal **V**ertical **G**ated **S**elective **U**nit (**HVGSU**) which incorporates **G**ated **S**elective **C**ells (**GSC**s) separately in both directions. To capture the correlation between periodic semantic information of both directions at each time step, we design fusion and selection operations in GSC. With a recurrent structure, HVGSU progressively captures more local semantic information until it encompasses the global semantic information. Thirdly, to improve efficiency, we propose the Recurrent Acceleration Network (RAN), which enables parallel processing of information transmission in both directions. Notably, RAN maintains a memory complexity of $\mathcal{O}(L)$ while reducing the time complexity to $\mathcal{O}(\sqrt{L})$. In addition, RAN could serve as a universal framework for integrating other models to facilitate information fusion and transmission. In summary, our main contributions are as follows:

- We propose a **W**ater-wave **I**nformation **T**ransmission and **R**ecurrent **A**cceleration **N**etwork (**WITRAN**), which represents a novel paradigm in information transmission by enabling bi-granular flows. We provide a comprehensive comparison of WITRAN with previous methods in Figure 1 to highlight its uniqueness. Furthermore, in order to compare the differences between WITRAN and the model (a)-(h) in Figure 1 more clearly, we have prepared Table 1 to highlight the advantages of WITRAN.

- We propose a novel **H**orizontal **V**ertical **G**ated **S**elective **U**nit (**HVGSU**) which captures long- and short-term periodic semantic information by using **G**ated **S**elective **C**ell (**GSC**) independently in both directions, preserving the characteristics of periodic semantic information. The fusion and selection in GSC can model the correlations of long- and short-term periodic semantic information. Furthermore, utilizing a recurrent structure with HVGSU facilitates the gradual capture of semantic information from local to global within a sequence.

- We present a **R**ecurrent **A**cceleration **N**etwork (**RAN**) which is a generic acceleration framework that significantly reduces the time complexity to $\mathcal{O}(\sqrt{L})$ while maintaining the memory complexity of $\mathcal{O}(L)$. We summarize the complexities of different methods in Table 2, demonstrating the superior efficiency of our method.

- We conduct extensive experiments on four benchmark datasets across various fields (energy, traffic and weather). The empirical studies demonstrate the remarkable performance of WITRAN, which achieves relative improvements of $5.80\%$ and $14.28\%$ in long-range and ultra-long-range forecasting respectively. In addition, the introduction of the generic RAN framework greatly improves the computational efficiency.

Table 1: Advantages of **WITRAN** compared to other methods.

| Advantages | (a) RNN | (b) CNN | (c) Full Attention | (d) LogTrans | (e) Pyraformer | (f) MICN | (g) PatchTST | (h) TimesNet | (i) WITRAN (ours) |
|---|---|---|---|---|---|---|---|---|---|
| Non point-wise semantic information capture | ✓ | ✓ | ✗ | ✓ | ✓ | ✓ | ✓ | ✓ | ✓ |
| Special design to capture long-term repetitive patterns | ✗ | ✗ | ✗ | ✗ | ✓ | ✗ | ✗ | ✓ | ✓ |
| Efficiently (1 or 2 layers) model global correlations | ✓(1) | ✗ | ✓(1) | ✗ | ✗ | ✓(2) | ✓(2) | ✗ | ✓(1) |
| Well solve the gradient vanishing/exploding problem of RNN | ✗ | – | – | – | – | – | – | – | ✓ |

Table 2: Complexity of forecasting models in training. $L$ is the sequence length, and $S$ is the stride in the PatchTST model.

| Methods | RNN | CNN | Transformer | LogTrans | Informer | Autoformer | Pyraformer | FEDformer | FiLM | PatchTST | MICN | WITRAN (ours) |
|---|---|---|---|---|---|---|---|---|---|---|---|---|
| Time | $\mathcal{O}(L)$ | $\mathcal{O}(L)$ | $\mathcal{O}(L^2)$ | $\mathcal{O}(L\log L)$ | $\mathcal{O}(L\log L)$ | $\mathcal{O}(L\log L)$ | $\mathcal{O}(L)$ | $\mathcal{O}(L)$ | $\mathcal{O}(L)$ | $\mathcal{O}((L/S)^2)$ | $\mathcal{O}(L)$ | $\mathcal{O}(\sqrt{L})$ |
| Memory | $\mathcal{O}(L)$ | $\mathcal{O}(L)$ | $\mathcal{O}(L^2)$ | $\mathcal{O}(L^2)$ | $\mathcal{O}(L\log L)$ | $\mathcal{O}(L\log L)$ | $\mathcal{O}(L)$ | $\mathcal{O}(L)$ | $\mathcal{O}(L)$ | $\mathcal{O}((L/S)^2)$ | $\mathcal{O}(L)$ | $\mathcal{O}(L)$ |

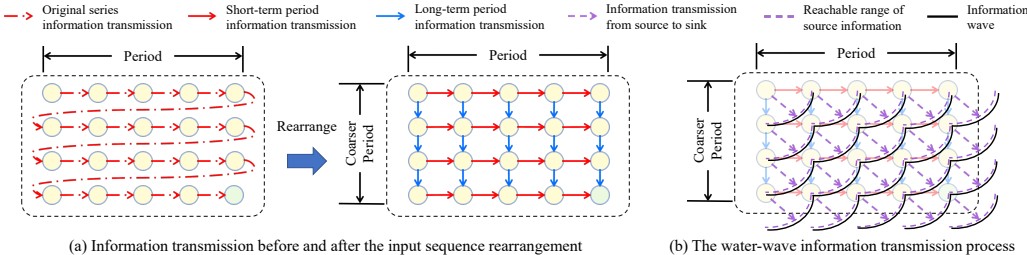

Figure 2: Input rearrangement and water-wave information transmission.

## 2 Related Work

Time series forecasting methods can be broadly categorized into statistical methods and neural network methods. Neural network methods, in turn, can be further classified into several subcategories based on the specific techniques employed, including RNN-based, CNN-based, Transformer-based, and other Deep Neural Networks (DNNs) by the methods used.

**Statistical Methods** with major representatives include ARIMA [Box and Jenkins, 1968], Prophet [Taylor and Letham, 2018], and Holt Winter [Athanasopoulos and Hyndman, 2020]. However, the inherent variability and complexity of real-world time series often render these methods inadequate, as they are built on hypotheses that may not align well with the characteristics of such data. As a result, the performance of these statistical methods tends to be limited.

**RNN-based Methods** [Hochreiter and Schmidhuber, 1997, Chung et al., 2014, Rangapuram et al., 2018, Salinas et al., 2020] possess the inherent advantage of capturing semantic information in time series data through their recurrent structure, which mimics the natural information transmission process of time series. This allows them to progressively capture semantic information from local to global contexts. These methods exhibit linear time and memory complexity, enabling efficient processing of time series data with a sequence length of $L$. However, when dealing with longer sequences, the problems of gradient vanishing/exploding [Pascanu et al., 2013] and information forgetting (see Appendix B for more details) further limit it.

**CNN-based methods** [Bai et al., 2018, Franceschi et al., 2019, Sen et al., 2019, Wu et al., 2023, Wang et al., 2023] are adept at capturing local semantic information through the application of convolutional kernels, while maintaining linear time and memory complexity. However, on the one hand, most of these methods face challenges in capturing comprehensive global information due to the limited receptive field of individual convolutional layers, which make the training process more difficult and overhead [Wang et al., 2023]. On the other hand, it is difficult to directly capture long- and short-term repetitive patterns on 1D inputs. MICN [Wang et al., 2023] adopted downsampling 1D convolutions and isometric convolutions combined with a multi-branch framework, which can effectively solve the former problem, but still suffers from the latter one. TimesNet [Wu et al., 2023] transforms 1D inputs into 2D space, leveraging a parameter-efficient inception block to capture intra- and inter-period variations. However, it solves the latter problem while still facing the former one.

**Transformer-based methods** have shown advancements in time series forecasting, with two main categories. The first category uses point-wise attention mechanisms, such as Vanilla Transformer [Vaswani et al., 2017], Informer [Zhou et al., 2021], and FEDformer [Zhou et al., 2022a], while facing challenges in extracting sufficient semantic information from individual time points. The second category employs non point-wise dot product techniques, including LogTrans [Li et al., 2019], Autoformer [Wu et al., 2021], Pyraformer [Liu et al., 2021] and PatchTST [Nie et al., 2023]. They reduce computational complexities to a certain degree, yet face difficulties in incorporating the long-term correlations in time-series. For a detailed description of these methods, please refer to Appendix A. Furthermore, it is worth noting that the majority of the aforementioned methods fail to achieve lower complexity than RNN-based methods. For a comprehensive comparison of the theoretical time and memory complexity, as well as experimental results, please refer to Appendix C.

**Other DNN methods** have also demonstrated promising performance. For example, FiLM [Zhou et al., 2022b] utilizes Legendre Polynomials and Fourier projection methods to capture historical in-

formation, eliminate noise, and expedite computations using low-rank approximation. DLinear [Zeng et al., 2023] has showcased impressive outcomes by employing simple linear operations. Nevertheless, both approaches encounter difficulties in capturing various repeating patterns present in the sequence.

## 3 The WITRAN Model

The time series forecasting task involves predicting future values $Y \in \mathbb{R}^{P \times c_{\text{out}}}$ for $P$ time steps based on the historical input sequences $X = \{x_1, x_2, \ldots, x_H\} \in \mathbb{R}^{H \times c_{\text{in}}}$ of $H$ time steps, where $c_{\text{in}}$ and $c_{\text{out}}$ represent the number of input and output features respectively. In order to integrate enough historical information for analysis, it is necessary for $H$ to have a sufficient size [Liu et al., 2021, Zeng et al., 2023]. Furthermore, capturing semantic information from the historical input is crucial for accurate forecasting, which includes modeling global and local correlations, as well as discovering long- and short-term repetitive patterns. However, how to address them simultaneously is a major challenge. With these in mind, we propose WITRAN, a novel information transmission framework akin to the propagation of water waves. WITRAN captures both long- and short-term periodic semantic information, as well as global-local semantic information simultaneously during information transmission. Moreover, WITRAN reduces time complexity while maintaining linear memory complexity. The overall structure of WITRAN is depicted in Figure 3.

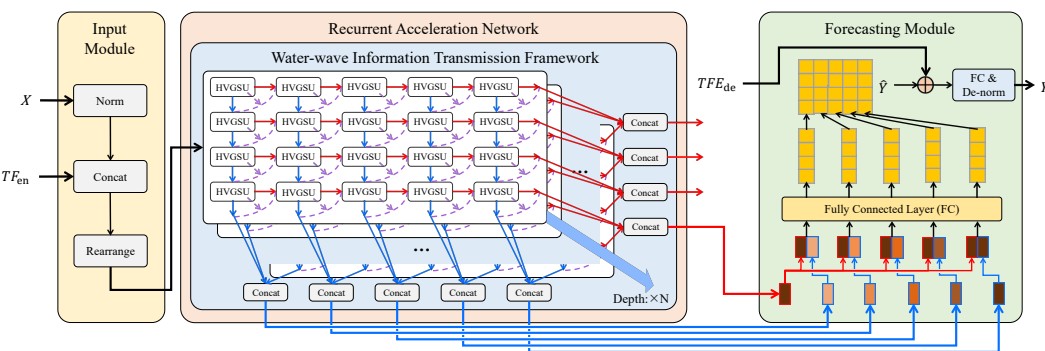

Figure 3: Overall structure of WITRAN.

### 3.1 Input Module

To facilitate the analysis of long- and short-term repetitive patterns, inspired by TimesNet [Wu et al., 2023], we first rearrange the sequence from 1D to 2D based on its natural period, as illustrated by Figure 2(a). Importantly, our approach involves analyzing the natural period of time series and setting appropriate hyperparameters to determine the input rearrangement, rather than using Fast Fourier Transform (FFT) to learn multiple adaptive periods of inputs in TimesNet. Consequently, our method is much simpler. Additionally, in order to minimize the distribution shift in datasets, we draw inspiration from NLinear [Zeng et al., 2023] and employ an adaptive learning approach to determine whether to perform simple normalization. The input module can be described as follows:

$$X_{1\text{D}} = \begin{cases} X & , norm = 0 \\ X - x_H & , norm = 1 \end{cases}$$
$$X_{2\text{D}} = \text{Rearrange}([X_{1\text{D}}, TF_{\text{en}}]), \tag{1}$$

here, $X_{1\text{D}} \in \mathbb{R}^{H \times c_{\text{in}}}$ represents the original input sequences, $x_H \in \mathbb{R}^{c_{\text{in}}}$ represents the input at the last time step of the original sequence, $TF_{\text{en}} \in \mathbb{R}^{H \times c_{\text{time}}}$ represents the temporal contextual features of original input sequence (e.g., HourOfDay, DayOfWeek, DayOfMonth and DayOfYear), where $c_{\text{time}}$ is the dimension of time features. $X_{2\text{D}} \in \mathbb{R}^{R \times C \times (c_{\text{in}} + c_{\text{time}})}$ represents the inputs after rearrangement, where $R$ denotes the total number of horizontal rows and $C$ denotes the vertical columns. $norm$ is an adaptive parameter for different tasks. $[\cdot]$ represents the concat operation and Rearrange represents the rearrange operation, with reference to Figure 2(a).

## 3.2 Horizontal Vertical Gated Selective Unit

To capture long- and short-term periodic semantic information and reserve their characteristics, we propose a novel **H**orizontal **V**ertical **G**ated **S**elective **U**nit (**HVGSU**) which consists of **G**ated **S**elective **C**ells (**GSC**s) in two directions. To capture the correlation at each time step between periodic semantic information of both directions, we design the specific operations in GSC. Furthermore, HVGSU is capable of capturing both global and local information via a recurrent structure. In this subsection, we will provide a detailed introduction to them.

**HVGSU** As depicted in Figure 3, the process of HVGSU via a recurrent structure is:

$$H_{\text{hor}}, H_{\text{ver}}, Out = \text{HVGSU}(X_{\text{2D}}), \tag{2}$$

where $H_{\text{hor}} \in \mathbb{R}^{L \times R \times d_{\text{model}}}$ and $H_{\text{ver}} \in \mathbb{R}^{L \times C \times d_{\text{model}}}$ represent the horizontal and the vertical output hidden state of HVGSU separately. $L$ is the depth of the model, and $Out \in \mathbb{R}^{R \times C \times d_{\text{model}}}$ denotes the output information of the last layer.

In greater detail, the cellular structure of HVGSU is shown in Figure 4(b), which consists of GSCs in two directions to capture the periodic semantic information of long- and short-term. The cell operations for row $r$ ($1 \leq r \leq R$) and column $c$ ($1 \leq c \leq C$) in layer $l$ ($1 \leq l \leq L$) can be formalized as:

$$
\begin{aligned}
h_{r,\,c,\,l}^{\text{hor}} &= \text{GSC}_{\text{hor}}(input_{r,\,c,\,l},\ h_{r,\,c-1,\,l}^{\text{hor}},\ h_{r-1,\,c,\,l}^{\text{ver}}) \\
h_{r,\,c,\,l}^{\text{ver}} &= \text{GSC}_{\text{ver}}(input_{r,\,c,\,l},\ h_{r-1,\,c,\,l}^{\text{ver}},\ h_{r,\,c-1,\,l}^{\text{hor}}) \\
o_{r,\,c,\,l} &= [h_{r,\,c,\,l}^{\text{hor}},\ h_{r,\,c,\,l}^{\text{ver}}],
\end{aligned} \tag{3}
$$

here, $input_{r,\,c,\,l} \in \mathbb{R}^{d_{\text{in}}}$. When $l = 1$, $input_{r,\,c,\,l} = x_{r,\,c} \in \mathbb{R}^{c_{\text{in}} + c_{\text{time}}}$ represents the input for the first layer, and when $l > 1$, $input_{r,\,c,\,l} = o_{r,\,c,\,l-1} \in \mathbb{R}^{2 \times d_{\text{model}}}$ represents the input for subsequent layers. $h_{r,\,c-1,\,l}^{\text{hor}}$ and $h_{r-1,\,c,\,l}^{\text{ver}} \in \mathbb{R}^{d_{\text{model}}}$ represent the horizontal and vertical hidden state inputs of the current cell. Note that when $r = 1$, $h_{r-1,\,c,\,l}^{\text{ver}}$ is replaced by an all-zero tensor of the same size, and the same is true for $h_{r,\,c-1,\,l}^{\text{hor}}$ when $c = 1$. $[\cdot]$ represents the concatenation operation and $o_{r,\,c,\,l} \in \mathbb{R}^{2 \times d_{\text{model}}}$ represents the output of the current cell.

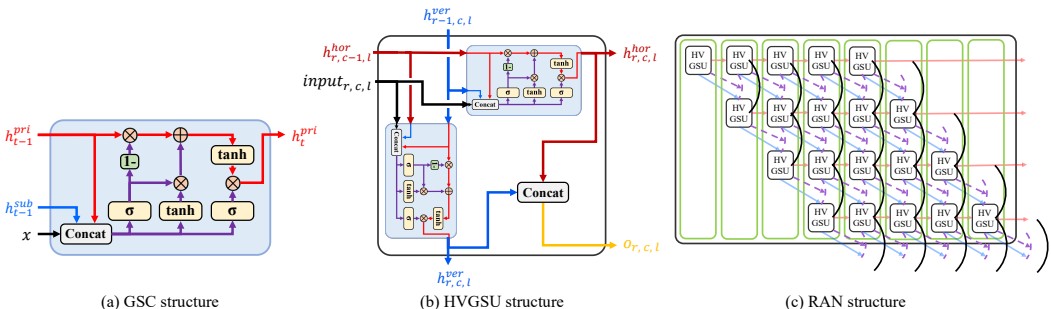

(a) GSC structure     (b) HVGSU structure     (c) RAN structure

Figure 4: The structure of our method.

**GSC** Inspired by the two popular RNN-based models, LSTM [Hochreiter and Schmidhuber, 1997] and GRU [Chung et al., 2014] (for more details, see Appendix A), we propose a Gated Selective Cell (GSC) to fuse and select information. Its structure is shown in Figure 4(a), which comprises two gates: the selection gate, and the output gate. The fused information consists of input and principal-subordinate hidden states, and the selection gate determines the retention of the original principal information and the addition of the fused information. Finally, the output gate determines the final output information of the cell. The different colored arrows in Figure 4(a) represent different semantic information transfer processes. The black arrow represents the input, the red arrows represent the process of transmitting principal hidden state information, the blue arrow represents the subordinate hidden state, and the purple arrows represent the process by which fused information of

principal-subordinate hidden states is transmitted. The formulations are given as follows:

$$S_t = \sigma(W_s[h_{t-1}^{\text{pri}}, \ h_{t-1}^{\text{sub}}, \ x] + b_s)$$
$$O_t = \sigma(W_o[h_{t-1}^{\text{pri}}, \ h_{t-1}^{\text{sub}}, \ x] + b_o)$$
$$h_f = \tanh(W_f[h_{t-1}^{\text{pri}}, \ h_{t-1}^{\text{sub}}, \ x] + b_f) \tag{4}$$
$$\widetilde{h_{t-1}^{\text{pri}}} = (1 - S_t) \odot h_{t-1}^{\text{pri}} + S_t \odot h_f$$
$$h_t^{\text{pri}} = \tanh(\widetilde{h_{t-1}^{\text{pri}}}) \odot O_t,$$

where $h_{t-1}^{\text{pri}}$ and $h_{t-1}^{\text{sub}} \in \mathbb{R}^{d_{\text{model}}}$ represent the input principal and subordinate hidden state, $x \in \mathbb{R}^{d_{\text{in}}}$ represents the input. $W_* \in \mathbb{R}^{d_{\text{model}} \times (2d_{\text{model}} + d_{\text{in}})}$ are weight matrices and $b_* \in \mathbb{R}^{d_{\text{model}}}$ are bias vectors. $S_t$ and $O_t$ denote the selection gate and the output gate, $\odot$ denotes an element-wise product, $\sigma(\cdot)$ and $\tanh(\cdot)$ denote the sigmoid and tanh activation function. $h_f$ and $\widetilde{h_{t-1}^{\text{pri}}} \in \mathbb{R}^{d_{\text{model}}}$ represent the intermediate variables of the calculation. $h_t^{\text{pri}}$ represents the output hidden state.

### 3.3 Recurrent Acceleration Network

In traditional recurrent structure, for two adjacent time steps in series, the latter one always waits for the former one until the information computation of the former one is completed. When the sequence becomes longer, this becomes slower. Fortunately, in the WIT framework we designed, some of the time step information can be computed in parallel. As shown in Figure 2(b), after a point is calculated, the right point in its horizontal direction and the point below it in its vertical direction can start calculation without waiting for each other. Therefore, we propose the Recurrent Acceleration Network (RAN) as our accelerated framework, which enables parallel computation of data points without waiting for each other, greatly improving the efficiency of information transmission in HVGSU. We place parallelizable points in a slice, and the updated information transfer process is shown in Figure 4(c). Each green box in Figure 4(c) represents a slice, and the number of green boxes is the number of times we need to recurrently compute. The meanings of the remaining markers are the same as those in Figure 2. Under the RAN framework, the recurrent length has changed from the sequence length $L = R \times C$ to $R + C - 1$, while the complexity of $R$ and $C$ is $\mathcal{O}(\sqrt{L})$. Thus, we have reduced the time complexity to $\mathcal{O}(\sqrt{L})$ via the RAN framework. It should be noted that although we parallelly compute some data points, which may increase some memory, the complexity of parallel computation, $\mathcal{O}(\sqrt{L})$, is far less than the complexity of saving the output variables, $\mathcal{O}(L)$, because we need to save the output information of each point in the sequence. Implementation source code for RAN is given in Appendix D.

### 3.4 Forecasting Module

In the forecasting module, we address the issue of error accumulation in the auto-regressive structure by drawing inspiration from Informer [Zhou et al., 2021] and Pyraformer [Liu et al., 2021], combining both horizontal and vertical hidden states, and then making predictions through a fully connected layer, as illustrated in Figure 3.

We utilize the last row of the horizontal hidden states as it contains sufficient global and latest short-term semantic information from the historical sequence. In contrast, all columns of the vertical hidden states, which capture different long-term semantic information, are all preserved. The combined operation not only maximizes the retention of the various semantic information needed for predicting the points, but also avoids excessive redundancy in order to obtain accurate predictions. This module can be formalized as follows:

$$H_{\text{rep}}^{\text{hor}} = \text{Repeat}(h_{\text{hor}})$$
$$H_{\text{h}-\text{v}} = \text{Reshape}([H_{\text{rep}}^{\text{hor}}, H_{\text{ver}}])$$
$$\widehat{Y} = \text{FC1}(H_{\text{h}-\text{v}}) \tag{5}$$
$$Y = \text{FC2}(\text{Reshape}(\widehat{Y}) + TFE_{\text{de}}),$$

where $TFE_{\text{de}} \in \mathbb{R}^{P \times d_{\text{model}}}$ represents time features encoding of the forecasting points separately. $h_{\text{hor}} \in \mathbb{R}^{L \times 1 \times d_{\text{model}}}$ represents the last row hidden state in $H_{\text{hor}}$. Repeat$(\cdot)$ is for repeat operation and

$H_{rep}^{\mathrm{hor}} \in \mathbb{R}^{L \times C \times d_{\mathrm{model}}}$ represents the hidden state after the repeat operation. $H_{\mathrm{h-v}} \in \mathbb{R}^{C \times (L*2d_{model})}$ is the vector of horizontal and vertical combination after reshape operation. FC1 and FC2 represent two fully connected layers respectively. $\widehat{Y} \in \mathbb{R}^{C \times (R_{\mathrm{de}}*d_{\mathrm{model}})}$ represents the intermediate variables of the calculation and $R_{\mathrm{de}} \times C = P$. $Y$ represents the output of this module, and it is worth noting that we need to utilize the adaptive parameter $norm$ for denormalization, when $norm = 1, Y = Y + x_H$.

## 4   Experiment

**Datasets**   To evaluate the proposed WITRAN, we conducted extensive experiments on four widely recognized real-world benchmark datasets for long-range and ultra-long-range forecasting. These datasets cover energy, traffic, and weather domains. We split all datasets into training, validation and test set in chronological order by the ratio of 6:2:2. More details about the datasets and implementation are described in Appendix E and Appendix F.

**Baselines**   In light of the underperformance produced by classical models such as ARIMA and simple RNN/CNN models, as evidenced by [Zhou et al., 2021] and [Wu et al., 2021], and the subpar performance exhibited by certain Transformer-based models like LogTrans [Li et al., 2019] and Reformer [Kitaev et al., 2020], as shown in [Wu et al., 2023] and [Wang et al., 2023], our study primarily incorporates six transformer-based models: PatchTST [Nie et al., 2023], FEDformer [Zhou et al., 2022a], Pyraformer [Liu et al., 2021], Autoformer[Zhou et al., 2022a], Informer [Zhou et al., 2021], Vanilla Transformer [Vaswani et al., 2017], and four non-transformer-based methods: MICN [Wang et al., 2023], TimesNet [Wu et al., 2023], DLinear [Zhou et al., 2022b], FiLM [Zhou et al., 2022b]. Please refer to Appendix A and Appendix H for more details of the baselines and implementation.

### 4.1   Experimental Results

For a better comparison, we adopted the experimental setup of Pyraformer [Liu et al., 2021] for long-range and ultra-long-range series forecasting. In addition, channel-independence is crucial for multivariate time series prediction [Nie et al., 2023], so it is necessary to verify the performance of models on a single channel to ensure their effectiveness across all channels in multivariate time series prediction. In this paper, experiments were conducted on a single channel. Note that in order to fairly compare the performance of each model, we set up the search space so that each model can perform optimally on each task. For further details, please refer to Appendix F.

**Long-range Forecasting Results**   We conducted five tasks on each dataset for long-range prediction, and the results are shown in Table 3. Taking the task setting 168-336 on the left side of the Table 3 as an example, it indicates that the input length is 168 and the prediction length is 336. Notably, our proposed WITRAN achieves state-of-the-art performance, surpassing the previous best method with an average MSE reduction of 5.803%. Specifically, WITRAN exhibits an average MSE reduction of 10.246% for ECL, 3.879% for traffic, 2.519% for ETTh1, 4.431% for ETTh2, and 7.939% for weather. Additionally, we note that the competition among the baselines is intense due to their best performance on each task, but none of them consistently performs well across all tasks. In contrast, WITRAN demonstrates its robust competitiveness across various tasks and datasets. In addition, we can find that in most cases, a longer input length yields greater improvement with the same prediction length. All findings above collectively highlight WITRAN's efficacy in addressing diverse time-series forecasting tasks in real-world applications. Further results and showcases are presented in Appendix H and Appendix J.

**Ultra-long-range Forecasting Results**   We also show the three tasks for ultra-long-range prediction results on each dataset in Table 4. The tasks on the left side of the table hold the same interpretation as above. Notably, WITRAN achieves an 14.275% averaged MSE reduction. More specifically, WITRAN demonstrates an average MSE reduction of 39.916% for ECL, 3.122% for traffic, 14.837% for ETTh1, 2.441% for ETTh2, and 11.062% for weather. In particular, our method showcases substantial improvements of over 10% in ECL, ETTh1, and weather, reinforcing our ability to predict ultra-long-range time series. And by comparing the results of ultra-long-range forecasting and long-range Forecasting, we can find that our method is more competitive for ultra-long-range prediction,

Table 3: **Long-range** forecasting results. A lower MSE or MAE indicates a better prediction. The best results are highlighted in bold and the suboptimal results are underlined.

| Methods | | WITRAN(Ours) | | MICN | | TimesNet | | PatchTST | | DLinear | | FiLM | | FEDformer | | Pyraformer | | Autoformer | | Informer | | Transformer | |
|---|---|---|---|---|---|---|---|---|---|---|---|---|---|---|---|---|---|---|---|---|---|---|---|
| | Metric | MSE | MAE | MSE | MAE | MSE | MAE | MSE | MAE | MSE | MAE | MSE | MAE | MSE | MAE | MSE | MAE | MSE | MAE | MSE | MAE | MSE | MAE |
| ECL | 168-168 | **0.2397** | **0.3519** | 0.3168 | 0.4067 | 0.2825 | 0.3797 | 0.2980 | 0.3832 | 0.2605 | 0.3579 | 0.2587 | 0.3557 | 0.3028 | 0.4020 | 0.2651 | 0.3802 | 0.3496 | 0.4337 | 0.3779 | 0.4594 | 0.3036 | 0.4068 |
| | 168-336 | **0.2607** | **0.3721** | 0.3002 | 0.4053 | 0.3505 | 0.4253 | 0.3840 | 0.4393 | 0.3080 | 0.3946 | 0.3062 | 0.3922 | 0.3522 | 0.4394 | 0.5392 | 0.5271 | 0.4733 | 0.5120 | 0.5037 | 0.5301 | 0.3583 | 0.4435 |
| | 336-336 | **0.2517** | **0.3627** | 0.3092 | 0.4132 | 0.3702 | 0.4307 | 0.4377 | 0.4654 | 0.2740 | 0.3720 | 0.2722 | 0.3659 | 0.3378 | 0.4303 | 0.2994 | 0.4030 | 0.5153 | 0.5304 | 0.4591 | 0.4991 | 0.5771 | 0.5643 |
| | 336-720 | **0.3084** | **0.4055** | 0.3820 | 0.4704 | 0.3879 | 0.4531 | 0.5502 | 0.5438 | 0.3208 | 0.4188 | 0.3171 | 0.4152 | 0.3813 | 0.4634 | 0.4856 | 0.5243 | 0.5045 | 0.5393 | 0.6545 | 0.5975 | 0.4368 | 0.4920 |
| | 720-720 | **0.2478** | **0.3651** | 0.3463 | 0.4381 | 0.3537 | 0.4386 | 0.5927 | 0.5742 | 0.3203 | 0.4202 | 0.3158 | 0.4154 | 0.4023 | 0.4769 | 0.3115 | 0.4218 | 0.9639 | 0.7520 | 0.4850 | 0.5238 | 0.3992 | 0.4640 |
| Traffic | 168-168 | **0.1377** | **0.2051** | 0.2428 | 0.3543 | 0.1490 | 0.2293 | 0.1622 | 0.2320 | 0.1519 | 0.2195 | 0.1501 | 0.2143 | 0.2469 | 0.3479 | 0.2979 | 0.3815 | 0.2378 | 0.3490 | 0.3363 | 0.3994 | 1.5204 | 0.9594 |
| | 168-336 | **0.1321** | **0.2059** | 0.2401 | 0.3514 | 0.1499 | 0.2356 | 0.1641 | 0.2364 | 0.1468 | 0.2210 | 0.1453 | 0.2165 | 0.2426 | 0.3449 | 0.5838 | 0.5652 | 0.2683 | 0.3803 | 0.5891 | 0.5608 | 0.6953 | 0.6015 |
| | 336-336 | **0.1306** | **0.2054** | 0.2413 | 0.3549 | 0.1446 | 0.2300 | 0.1546 | 0.2332 | 0.1325 | 0.2114 | 0.1324 | 0.2104 | 0.2339 | 0.3365 | 0.4703 | 0.4964 | 0.2460 | 0.3567 | 0.5447 | 0.5384 | 0.8482 | 0.6424 |
| | 336-720 | **0.1391** | **0.2175** | 0.2422 | 0.3513 | 0.1584 | 0.2440 | 0.1747 | 0.2536 | 0.1449 | 0.2252 | 0.1438 | 0.2229 | 0.2987 | 0.3976 | 0.5235 | 0.5292 | 0.2849 | 0.3956 | 1.2044 | 0.8254 | 0.7320 | 0.6233 |
| | 720-720 | 0.1408 | **0.2191** | 0.2552 | 0.3709 | 0.1546 | 0.2410 | 0.1543 | 0.2441 | 0.1410 | 0.2241 | **0.1383** | 0.2208 | 0.2667 | 0.3685 | 0.4811 | 0.4962 | 0.2959 | 0.4045 | 1.2954 | 0.9205 | 1.1963 | 0.8271 |
| ETTh1 | 168-168 | 0.1105 | 0.2589 | 0.1257 | 0.2803 | 0.1133 | 0.2612 | 0.1212 | 0.2704 | 0.1122 | 0.2605 | **0.1091** | **0.2558** | 0.1284 | 0.2826 | 0.1534 | 0.3287 | 0.1318 | 0.2872 | 0.1563 | 0.3299 | 0.1504 | 0.3257 |
| | 168-336 | 0.1189 | 0.2714 | 0.1422 | 0.3006 | 0.1202 | 0.2732 | 0.1287 | 0.2808 | 0.1251 | 0.2794 | **0.1187** | **0.2708** | 0.1271 | 0.2810 | 0.1665 | 0.3419 | 0.1315 | 0.2878 | 0.1663 | 0.3335 | 0.1599 | 0.3324 |
| | 336-336 | **0.1112** | **0.2638** | 0.1576 | 0.3159 | 0.1279 | 0.2846 | 0.1496 | 0.3039 | 0.1261 | 0.2803 | 0.1196 | 0.2738 | 0.1252 | 0.2794 | 0.1408 | 0.3087 | 0.1384 | 0.2959 | 0.1648 | 0.3291 | 0.1438 | 0.3121 |
| | 336-720 | **0.1494** | **0.3092** | 0.2219 | 0.3729 | 0.1501 | 0.3127 | 0.2092 | 0.3659 | 0.1942 | 0.3462 | 0.1793 | 0.3335 | 0.1534 | 0.3178 | 0.3984 | 0.5202 | 0.1928 | 0.3450 | 0.1522 | 0.3203 | 0.1644 | 0.3304 |
| | 720-720 | **0.1296** | **0.2868** | 0.2959 | 0.4402 | 0.1510 | 0.3118 | 0.2178 | 0.3694 | 0.1920 | 0.3435 | 0.1845 | 0.3379 | 0.1386 | 0.2995 | 0.1563 | 0.3253 | 0.2388 | 0.3869 | 0.1595 | 0.3259 | 0.1730 | 0.3414 |
| ETTh2 | 168-168 | **0.2389** | **0.3813** | 0.2734 | 0.4162 | 0.2655 | 0.4051 | 0.2582 | 0.3983 | 0.2556 | 0.3944 | 0.2546 | 0.3942 | 0.2844 | 0.4285 | 0.2746 | 0.4080 | 0.2903 | 0.4326 | 0.3764 | 0.4863 | 0.3043 | 0.4365 |
| | 168-336 | **0.2277** | **0.3778** | 0.3017 | 0.4429 | 0.2725 | 0.4163 | 0.3206 | 0.4515 | 0.2891 | 0.4256 | 0.2894 | 0.4263 | 0.2961 | 0.4355 | 0.2392 | 0.3834 | 0.4471 | 0.4964 | 0.3364 | 0.4583 | 0.3662 | 0.4671 |
| | 336-336 | **0.2432** | **0.3922** | 0.3472 | 0.4796 | 0.3184 | 0.4431 | 0.3559 | 0.4779 | 0.2950 | 0.4329 | 0.2951 | 0.4347 | 0.2884 | 0.4314 | 0.2610 | 0.4010 | 0.2805 | 0.4255 | 0.3709 | 0.4785 | 0.3218 | 0.4412 |
| | 336-720 | 0.2373 | **0.3888** | 0.4248 | 0.5268 | 0.2858 | 0.4253 | 0.4936 | 0.5592 | 0.4125 | 0.5136 | 0.4158 | 0.5162 | 0.3425 | 0.4656 | **0.2341** | 0.3818 | 0.3372 | 0.4625 | 0.3572 | 0.4675 | 0.3582 | 0.4629 |
| | 720-720 | **0.2635** | **0.4018** | 0.3549 | 0.4805 | 0.2936 | 0.4238 | 0.5243 | 0.5745 | 0.3495 | 0.4749 | 0.4045 | 0.5105 | 0.3275 | 0.4534 | 0.2795 | 0.4151 | 0.4668 | 0.5477 | 0.3585 | 0.4699 | 0.3087 | 0.4320 |
| Weather | 168-168 | **0.2050** | **0.3338** | 0.2231 | 0.3489 | 0.2420 | 0.3608 | 0.2469 | 0.3597 | 0.2421 | 0.3578 | 0.2426 | 0.3544 | 0.2583 | 0.3774 | 0.2144 | 0.3451 | 0.2670 | 0.3813 | 0.2639 | 0.3926 | 0.2200 | 0.3438 |
| | 168-336 | **0.2197** | **0.3470** | 0.2663 | 0.3837 | 0.2821 | 0.3885 | 0.3040 | 0.4049 | 0.2918 | 0.3975 | 0.2981 | 0.3988 | 0.2909 | 0.4030 | 0.2594 | 0.3833 | 0.2990 | 0.4096 | 0.2798 | 0.4061 | 0.2230 | 0.3488 |
| | 336-336 | **0.2163** | **0.3482** | 0.2701 | 0.3804 | 0.2684 | 0.3752 | 0.3149 | 0.4145 | 0.2905 | 0.3969 | 0.2943 | 0.3969 | 0.2791 | 0.3984 | 0.2310 | 0.3591 | 0.3066 | 0.4162 | 0.2898 | 0.4129 | 0.2308 | 0.3556 |
| | 336-720 | **0.2054** | **0.3424** | 0.3086 | 0.4138 | 0.2930 | 0.4045 | 0.4358 | 0.4937 | 0.3897 | 0.4739 | 0.4096 | 0.4767 | 0.2648 | 0.3915 | 0.3241 | 0.4300 | 0.3468 | 0.4592 | 0.2483 | 0.3778 | 0.2334 | 0.3570 |
| | 720-720 | **0.2008** | **0.3417** | 0.2828 | 0.3969 | 0.2967 | 0.4070 | 0.5701 | 0.5491 | 0.3724 | 0.4614 | 0.3999 | 0.4661 | 0.2416 | 0.3728 | 0.2378 | 0.3684 | 0.4309 | 0.5085 | 0.3545 | 0.4569 | 0.2463 | 0.3722 |

Table 4: **Ultra-long-range** forecasting results. A lower MSE or MAE indicates a better prediction. The best results are highlighted in bold and the suboptimal results are underlined.

| Methods | | WITRAN(Ours) | | MICN | | TimesNet | | PatchTST | | DLinear | | FiLM | | FEDformer | | Pyraformer | | Autoformer | | Informer | | Transformer | |
|---|---|---|---|---|---|---|---|---|---|---|---|---|---|---|---|---|---|---|---|---|---|---|---|
| | Metric | MSE | MAE | MSE | MAE | MSE | MAE | MSE | MAE | MSE | MAE | MSE | MAE | MSE | MAE | MSE | MAE | MSE | MAE | MSE | MAE | MSE | MAE |
| ECL | 720-1440 | 0.2499 | **0.3727** | 1.0460 | 0.7765 | 0.6119 | 0.5962 | 0.8243 | 0.6704 | 0.4923 | 0.5473 | 0.4730 | 0.5336 | 0.4833 | 0.5393 | 0.3250 | 0.4332 | 0.4957 | 0.9533 | 0.5064 | 0.5317 | 0.4030 | 0.4797 |
| | 1440-1440 | **0.2408** | **0.3680** | 2.2862 | 1.2207 | 0.5720 | 0.5712 | 0.9053 | 0.7328 | 0.5146 | 0.5615 | 0.4849 | 0.5429 | 0.5142 | 0.5571 | 0.4895 | 0.5280 | 1.7873 | 1.0283 | 0.7247 | 0.6292 | 0.5531 | 0.5524 |
| | 1440-2880 | **0.3359** | **0.4383** | 2.8936 | 1.3717 | 0.7683 | 0.6846 | 1.1282 | 0.8087 | 0.8355 | 0.7193 | 0.6847 | 0.6493 | 0.9018 | 1.5276 | 0.4320 | 0.5161 | 1.2867 | 0.8878 | 0.6152 | 0.5953 | 0.5243 | 0.5460 |
| Traffic | 720-1440 | 0.1672 | 0.2449 | 0.2876 | 0.3916 | 0.1882 | 0.2656 | 0.1904 | 0.2685 | 0.1639 | **0.2412** | **0.1638** | 0.2448 | 0.2753 | 0.3650 | 0.4463 | 0.4609 | 0.3104 | 0.4095 | 0.7614 | 0.6469 | 0.9876 | 0.7445 |
| | 1440-1440 | **0.1543** | **0.2325** | 0.2950 | 0.3923 | 0.1598 | 0.2388 | 0.1817 | 0.2764 | 0.1599 | 0.2411 | 0.1602 | 0.2437 | 0.2848 | 0.3681 | 0.4710 | 0.4916 | 0.2970 | 0.3999 | 0.7375 | 0.6414 | 0.7430 | 0.6492 |
| | 1440-2880 | **0.1425** | **0.2333** | 0.2823 | 0.3874 | 0.1560 | 0.2409 | 0.2029 | 0.3100 | 0.1550 | 0.2472 | 0.1744 | 0.2693 | 0.2952 | 0.3844 | 0.5165 | 0.5305 | 0.3035 | 0.3982 | 0.9849 | 0.7618 | 0.6000 | 0.5877 |
| ETTh1 | 720-1440 | **0.1331** | **0.2943** | 0.4640 | 0.5836 | 0.1391 | 0.3049 | 0.3708 | 0.4906 | 0.2952 | 0.4370 | 0.2949 | 0.4388 | 0.1768 | 0.3409 | 0.1666 | 0.3315 | 0.3298 | 0.4741 | 0.1378 | 0.3051 | 0.1905 | 0.3555 |
| | 1440-1440 | **0.1304** | **0.2902** | 0.5650 | 0.6293 | 0.1801 | 0.3372 | 0.4475 | 0.5329 | 0.2200 | 0.3714 | 0.2294 | 0.3759 | 0.3574 | 0.4878 | 0.3487 | 0.4866 | 0.4531 | 0.5507 | 0.1430 | 0.3156 | 0.1972 | 0.3630 |
| | 1440-2880 | **0.1850** | **0.3452** | 0.7591 | 0.7215 | 0.2732 | 0.4094 | 0.9617 | 0.8271 | 0.3773 | 0.4794 | 0.6834 | 0.7096 | 0.4269 | 0.5252 | 0.5857 | 0.6760 | 1.3566 | 0.9235 | 0.3177 | 0.4733 | 0.3495 | 0.4911 |
| ETTh2 | 720-1440 | **0.2915** | **0.4289** | 0.4922 | 0.5649 | 0.4186 | 0.5092 | 0.9401 | 0.7680 | 0.5037 | 0.5645 | 0.7166 | 0.6628 | 0.3731 | 0.4827 | 0.2952 | 0.4336 | 0.5633 | 0.5996 | 0.4025 | 0.4991 | 0.3712 | 0.4805 |
| | 1440-1440 | **0.2815** | **0.4220** | 0.5030 | 0.5644 | 0.4409 | 0.5218 | 0.7860 | 0.6704 | 0.5176 | 0.5734 | 0.7446 | 0.6590 | 0.3906 | 0.4951 | 0.2946 | 0.4316 | 0.8029 | 0.7140 | 0.3484 | 0.4786 | 0.3797 | 0.4818 |
| | 1440-2880 | **0.3280** | 0.4585 | 0.5549 | 0.5886 | 1.5304 | 0.9026 | 2.0561 | 1.1595 | 0.5053 | 0.5584 | 3.2835 | 1.6030 | 1.7167 | 0.9698 | 0.3345 | **0.4544** | 4.1031 | 1.7198 | 0.3335 | **0.4482** | 0.3737 | 0.4787 |
| Weather | 720-1440 | **0.1872** | **0.3312** | 0.3999 | 0.4848 | 0.2407 | 0.3694 | 0.5453 | 0.5631 | 0.4406 | 0.5264 | 0.6360 | 0.5997 | 0.2352 | 0.3733 | 0.6810 | 0.6352 | 0.8599 | 0.7064 | 0.2466 | 0.3849 | 0.2188 | 0.3512 |
| | 1440-1440 | **0.1907** | **0.3366** | 0.2873 | 0.4201 | 0.2869 | 0.4033 | 0.5371 | 0.5559 | 0.3147 | 0.4417 | 0.6002 | 0.5880 | 0.2226 | 0.3609 | 0.2401 | 0.3777 | 0.9766 | 0.7739 | 0.2556 | 0.3969 | 0.2610 | 0.3823 |
| | 1440-2880 | **0.1769** | **0.3257** | 0.3570 | 0.4810 | 0.2199 | 0.3563 | 0.9061 | 0.7220 | 0.3197 | 0.4533 | 1.2605 | 0.8805 | 0.2138 | 0.3599 | 0.1852 | 0.3332 | 1.7465 | 1.0962 | 0.2126 | 0.3600 | 0.1993 | 0.3436 |

underscoring WITRAN's capability to capture semantic information in ultra-long historical input. More results and showcases are provided in Appendix H and Appendix J.

## 4.2 Time and Memory Consumption

To assess the efficiency of our proposed WITRAN, we compared its time and memory consumption with selected baselines that demonstrate superior performance. To fully evaluate the actual complexity of our model and other models for the long range forecasting and ultra-long range forecasting tasks, we set up two cases: 1) constant input length and varying prediction length, 2) constant prediction length and varying input length. In both cases, we fixed the input length and output length at 720. Figure 5 illustrates the comparison, revealing that WITRAN achieves the lowest actual time complexity and memory complexity. This section is described in more detail in Appendix I.

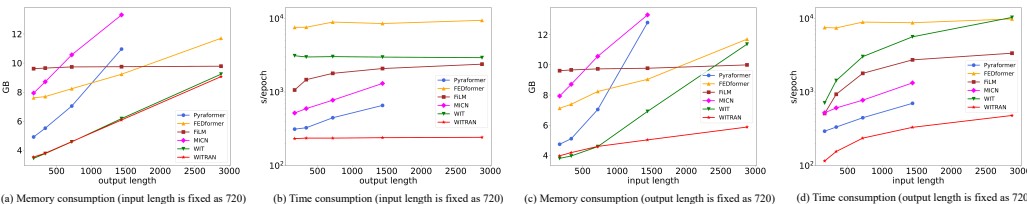

Figure 5: Time and memory consumption. WIT is the framework that does not involve the RAN.

## 4.3 Ablation Study

The impact of RAN can be clearly observed in Figure 5. In addition, we conducted ablation studies to measure the impact of HVGSU and GSC in WIT, and the combined operation of the last row of horizontally hidden states and each column of vertically hidden states. Detailed results and discussions

can be found in Appendix G. Here, we provide a concise summary of the key findings: 1) The fusion and selection design of GSC enables both long- and short-term semantic information to be captured for each historical input data points. 2) Setting up independent cells in both directions enables to extract semantic information in the long- and short-term respectively. 3) The specially designed combined operation can make fuller use of the captured local and global semantic information while ensuring that the information is not redundant. 4) RAN offers advantages in terms of speed and space complexity. The utilization of RAN eliminates the need to store excessive intermediate variables, as shown in the previous section.

## 4.4 Robustness Analysis

We have followed MICN [Wang et al., 2023] and introduced a simple white noise injection to demonstrate the robustness of our model. Specifically, we randomly select a proportion $\varepsilon$ of data from the original input sequence and apply random perturbations within the range $[-2X_i, 2X_i]$ to the selected data, where $X_i$ denotes the original data. After the noise injection, the data is then used for training, and the MSE and MAE metrics are recorded in Table 5.

Table 5: **Robustness experiments** of forecasting results. Different $\varepsilon$ indicates different proportions of noise injection. And **WITRAN** is used as the base model.

| | Tasks | 168-168 | | 168-336 | | 336-336 | | 336-720 | | 720-720 | | 720-1440 | | 1440-1440 | | 1440-2880 | |
|---|---|---|---|---|---|---|---|---|---|---|---|---|---|---|---|---|---|
| | Metric | MSE | MAE | MSE | MAE | MSE | MAE | MSE | MAE | MSE | MAE | MSE | MAE | MSE | MAE | MSE | MAE |
| ECL | $\varepsilon=0\%$ | 0.2397 | 0.3519 | 0.2607 | 0.3721 | 0.2517 | 0.3627 | 0.3084 | 0.4055 | 0.2478 | 0.3651 | 0.2499 | 0.3727 | 0.2408 | 0.3680 | 0.3359 | 0.4383 |
| | $\varepsilon=1\%$ | 0.2420 | 0.3534 | 0.2630 | 0.3738 | 0.2516 | 0.3626 | 0.3078 | 0.4060 | 0.2345 | 0.3960 | 0.2502 | 0.3739 | 0.2395 | 0.3672 | 0.3372 | 0.4476 |
| | $\varepsilon=5\%$ | 0.2463 | 0.3573 | 0.2692 | 0.3788 | 0.2567 | 0.3652 | 0.3039 | 0.4046 | 0.2525 | 0.3703 | 0.2565 | 0.3786 | 0.2410 | 0.3687 | 0.3315 | 0.4408 |
| | $\varepsilon=10\%$ | 0.2543 | 0.3621 | 0.2726 | 0.3811 | 0.2702 | 0.3736 | 0.3098 | 0.4061 | 0.2569 | 0.3756 | 0.2674 | 0.3880 | 0.2444 | 0.3716 | 0.3366 | 0.4445 |
| Traffic | $\varepsilon=0\%$ | 0.1377 | 0.2051 | 0.1321 | 0.2059 | 0.1306 | 0.2054 | 0.1391 | 0.2175 | 0.1408 | 0.2191 | 0.1672 | 0.2449 | 0.1543 | 0.2325 | 0.1425 | 0.2333 |
| | $\varepsilon=1\%$ | 0.1376 | 0.2063 | 0.1329 | 0.2083 | 0.1316 | 0.2074 | 0.1423 | 0.2218 | 0.1409 | 0.2218 | 0.1676 | 0.2466 | 0.1551 | 0.2372 | 0.1436 | 0.2360 |
| | $\varepsilon=5\%$ | 0.1370 | 0.2115 | 0.1362 | 0.2148 | 0.1323 | 0.2106 | 0.1432 | 0.2260 | 0.1416 | 0.2247 | 0.1699 | 0.2519 | 0.1572 | 0.2373 | 0.1519 | 0.2480 |
| | $\varepsilon=10\%$ | 0.1409 | 0.2164 | 0.1355 | 0.2193 | 0.1372 | 0.2200 | 0.1467 | 0.2325 | 0.1450 | 0.2291 | 0.1652 | 0.2475 | 0.1561 | 0.2383 | 0.1575 | 0.2580 |
| ETTh$_1$ | $\varepsilon=0\%$ | 0.1105 | 0.2589 | 0.1189 | 0.2714 | 0.1112 | 0.2638 | 0.1494 | 0.3092 | 0.1296 | 0.2868 | 0.1331 | 0.2943 | 0.1304 | 0.2902 | 0.1850 | 0.3452 |
| | $\varepsilon=1\%$ | 0.1112 | 0.2596 | 0.1208 | 0.2726 | 0.1111 | 0.2637 | 0.1463 | 0.3035 | 0.1304 | 0.2885 | 0.1367 | 0.2978 | 0.1319 | 0.2907 | 0.1834 | 0.3484 |
| | $\varepsilon=5\%$ | 0.1135 | 0.2622 | 0.1221 | 0.2751 | 0.1199 | 0.2689 | 0.1527 | 0.3124 | 0.1304 | 0.2888 | 0.1336 | 0.2952 | 0.1359 | 0.2955 | 0.1801 | 0.3435 |
| | $\varepsilon=10\%$ | 0.1137 | 0.2628 | 0.1227 | 0.2757 | 0.1196 | 0.2688 | 0.1559 | 0.3148 | 0.1358 | 0.2952 | 0.1374 | 0.2994 | 0.1405 | 0.3015 | 0.1842 | 0.3479 |
| ETTh$_2$ | $\varepsilon=0\%$ | 0.2389 | 0.3813 | 0.2277 | 0.3778 | 0.2432 | 0.3922 | 0.2373 | 0.3888 | 0.2635 | 0.4018 | 0.2915 | 0.4289 | 0.2815 | 0.4220 | 0.3280 | 0.4585 |
| | $\varepsilon=1\%$ | 0.2535 | 0.3904 | 0.2284 | 0.3777 | 0.2459 | 0.3942 | 0.2390 | 0.3902 | 0.2629 | 0.3902 | 0.3049 | 0.4358 | 0.2936 | 0.4297 | 0.3353 | 0.4642 |
| | $\varepsilon=5\%$ | 0.2364 | 0.3799 | 0.2379 | 0.3834 | 0.2563 | 0.4025 | 0.2501 | 0.3967 | 0.2661 | 0.4056 | 0.3207 | 0.4482 | 0.2935 | 0.4308 | 0.3307 | 0.4603 |
| | $\varepsilon=10\%$ | 0.2603 | 0.3959 | 0.2475 | 0.3902 | 0.2613 | 0.4054 | 0.2581 | 0.4022 | 0.2674 | 0.4053 | 0.3193 | 0.4481 | 0.2847 | 0.4237 | 0.3317 | 0.4612 |
| Weather | $\varepsilon=0\%$ | 0.2050 | 0.3338 | 0.2197 | 0.3470 | 0.2163 | 0.3482 | 0.2054 | 0.3424 | 0.2008 | 0.3417 | 0.1872 | 0.3312 | 0.1907 | 0.3366 | 0.1769 | 0.3257 |
| | $\varepsilon=1\%$ | 0.2050 | 0.3343 | 0.2154 | 0.3470 | 0.2214 | 0.3522 | 0.2055 | 0.3419 | 0.2005 | 0.3419 | 0.1872 | 0.3313 | 0.1903 | 0.3361 | 0.1828 | 0.3327 |
| | $\varepsilon=5\%$ | 0.2057 | 0.3362 | 0.2241 | 0.3517 | 0.2268 | 0.3557 | 0.2058 | 0.3426 | 0.2008 | 0.3421 | 0.1867 | 0.3305 | 0.1897 | 0.3353 | 0.1831 | 0.3357 |
| | $\varepsilon=10\%$ | 0.2059 | 0.3369 | 0.2220 | 0.3484 | 0.2308 | 0.3595 | 0.2059 | 0.3438 | 0.2007 | 0.3418 | 0.1854 | 0.3290 | 0.1900 | 0.3370 | 0.1828 | 0.3336 |

It can be found that as the perturbation proportion increases, there is a slight increase in the MSE and MAE metrics in terms of forecasting. It indicates that WITRAN demonstrates good robustness when dealing with less noisy data (up to 10%), and it possesses a significant advantage in effectively handling various abnormal data fluctuations.

## 5 Conclusions

In this paper, we propose WITRAN, a novel Water-wave Information Transmission framework and a universal acceleration framework. WITRAN effectively captures both long- and short-term repetitive patterns and global and local correlations with $\mathcal{O}(\sqrt{L})$ time complexity and $\mathcal{O}(L)$ memory complexity. The experimental results demonstrate the remarkable performance and efficiency of WITRAN. However, this recurrent structure is still not optimally efficient for Python-based implementations because of the information waiting between slices. Therefore, we plan to explore the integration of WITRAN into an interface using C++, similar to the implementation of nn.GRU/nn.LSTM in PyTorch in the future, to further improve its efficiency.

## Acknowledgments

This work was supported by the Fundamental Research Funds for the Central Universities under Grant No. 2022YJS142. Furthermore, we would like to express our gratitude to Shuo Wang and Shuohao Lin for their valuable discussions and assistance.

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

# A   A Brief Overview of RNN-Based and Transformer-Based Models

In this section, we will provide a brief overview of representative variants of RNN-based methods (Appendix A.1) and Transformer-based methods (Appendix A.2). Detailed descriptions of other methods can be found in Section 2.

## A.1   RNN-based methods

LSTM [Hochreiter and Schmidhuber, 1997] and GRU [Chung et al., 2014] are two prominent variants of RNN-based models. They have gained popularity in diverse domains, such as natural language processing, speech recognition, and video analysis due to their ability to capture important information. This subsection, we will provide a detailed introduction to LSTM and GRU.

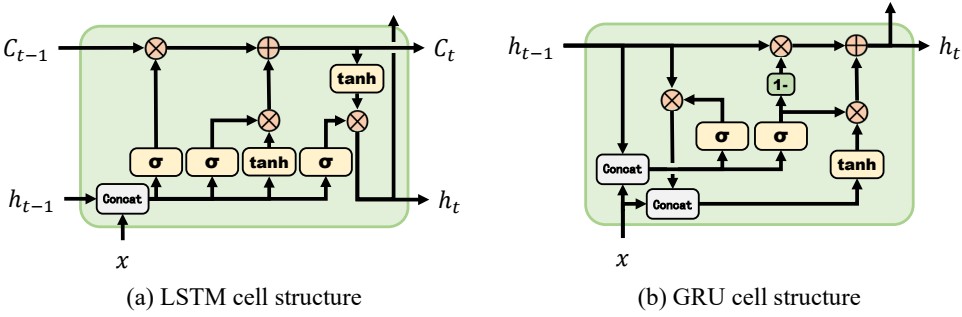

(a) LSTM cell structure          (b) GRU cell structure

Figure 6: The structure of LSTM and GRU.

**LSTM**   The structure of LSTM, shown in Figure 6(a), comprises three gates: the input gate, the forget gate and the output gate. In addition, LSTM has two vectors: the hidden state and the cell state, for recursively processing the information flow between different time steps. LSTM can be formalized as follows:

$$
\begin{aligned}
f_t^{\text{lstm}} &= \sigma(W_f^{\text{lstm}}[h_{t-1},\, x] + b_f^{\text{lstm}}) \\
i_t^{\text{lstm}} &= \sigma(W_i^{\text{lstm}}[h_{t-1},\, x] + b_i^{\text{lstm}}) \\
\widetilde{C_t} &= \tanh(W_c^{\text{lstm}}[h_{t-1},\, x] + b_c^{\text{lstm}}) \\
o_t^{\text{lstm}} &= \sigma(W_o^{\text{lstm}}[h_{t-1},\, x] + b_o^{\text{lstm}}) \\
C_t &= f_t^{\text{lstm}} \odot C_{t-1} + i_t^{\text{lstm}} \odot \widetilde{C_t} \\
h_t &= o_t^{\text{lstm}} \odot \tanh(C_t),
\end{aligned}
\tag{6}
$$

where $f_t^{\text{lstm}}$, $i_t^{\text{lstm}}$ and $o_t^{\text{lstm}}$ represent the input gate, the forget gate, and the output gate. $x \in \mathbb{R}^{d_{\text{in}}}$ represents the input, and $C_{t-1}$, $h_{t-1} \in \mathbb{R}^{d_{\text{model}}}$ represents the input cell state and the input hidden state. $W_*^{\text{lstm}} \in \mathbb{R}^{d_{\text{model}} \times (d_{\text{model}}+d_{\text{in}})}$ are weight matrices and $b_*^{\text{lstm}} \in \mathbb{R}^{d_{\text{model}}}$ are bias vectors. $C_t$ and $h_t$ denote the output cell state and the output hidden state; $\widetilde{C_t}$ represents the intermediate variables of the calculation; $\odot$ denotes an element-wise product; $\sigma(\cdot)$ and $\tanh(\cdot)$ denote the sigmoid and tanh activation function.

**GRU**   The structure of GRU, depicted in Figure 6(b), consists of two gates: the reset gate and the update gate. Besides, GRU has one vector: the hidden state. It can be mathematically formulated as follows:

$$
\begin{aligned}
r_t^{\text{gru}} &= \sigma(W_r^{\text{gru}}[h_{t-1},\, x] + b_r^{\text{gru}}) \\
u_t^{\text{gru}} &= \sigma(W_u^{\text{gru}}[h_{t-1},\, x] + b_u^{\text{gru}}) \\
\widetilde{h_t^{\text{gru}}} &= \tanh(W_h^{\text{gru}}[r_t^{\text{gru}} \odot h_{t-1},\, x] + b_h^{\text{gru}}) \\
h_t &= (1 - u_t^{\text{gru}}) \odot h_{t-1} + u_t^{\text{gru}} \odot \widetilde{h_t^{\text{gru}}},
\end{aligned}
\tag{7}
$$

where $r_t^{\mathrm{gru}}$ and $u_t^{\mathrm{gru}}$ represent the reset gate and the update gate. $x \in \mathbb{R}^{d_{\mathrm{in}}}$ represents the input, and $h_{t-1} \in \mathbb{R}^{d_{\mathrm{model}}}$ represent the input hidden state. $W_*^{\mathrm{gru}} \in \mathbb{R}^{d_{\mathrm{model}} \times (d_{\mathrm{model}} + d_{\mathrm{in}})}$ are weight matrices and $b_*^{\mathrm{gru}} \in \mathbb{R}^{d_{\mathrm{model}}}$ are bias vectors. $h_t$ denotes the output hidden state; $\widetilde{h_t}^{\mathrm{gru}}$ represents the intermediate variables of the calculation; $\odot$ denotes an element-wise product; $\sigma(\cdot)$ and $\tanh(\cdot)$ denote the sigmoid and tanh activation function.

## A.2 Transformer-based methods

In recent years, variants of Transformer have made significant progress in time series forecasting and can be divided into two categories.

The first category captures the correlation between different time steps through point-wise attention mechanisms. Vanilla Transformer [Vaswani et al., 2017] is able to effectively extract correlations between any two-time steps through its self-attention mechanism, but its complexity is too high, reaching $\mathcal{O}(L^2)$. Informer [Zhou et al., 2021] proposes a ProbSparse self-attention with distilling techniques, reducing the complexity to $\mathcal{O}(L \log L)$. FEDformer [Zhou et al., 2022a] takes a different approach by applying attention modules in the frequency domain through Fourier or wavelet transformations, achieving a complexity of $\mathcal{O}(L)$. Despite the performance and efficiency improvements achieved by these representative methods, they face difficulties in extracting sufficient semantic information from a single time point, which has been raised in previous works [Nie et al., 2023, Wu et al., 2023].

The second category adopts non-dot-product techniques to extract correlation information. Log-Trans [Li et al., 2019] effectively captures local information by leveraging LogSparse and convolutional self-attention layers. However, it overlooks the long-term repetitive pattern of points among subseries and exhibits a high complexity of $\mathcal{O}(L \log L)$. Drawing from the traditional ideas of time-series analysis, Autoformer [Wu et al., 2021] is able to capture long-range trends and short-range changes in sequences through decomposition and auto-correlation mechanisms. However, its complexity remains high at $\mathcal{O}(L \log L)$. Pyraformer [Liu et al., 2021] initializes coarser-scale node information in the pyramidal graph by capturing local and global correlations through convolution. It then employs pyramidal attention to effectively capture long- and short-term repetitive patterns, leading to significant performance improvements while reducing complexity to $\mathcal{O}(L)$. However, Pyraformer still suffers from the limitation of the receptive field of the convolution kernel as discussed earlier. PatchTST [Nie et al., 2023] captures local semantic information through patch, and further reduces the complexity to $\mathcal{O}((L/S)^2)$, where $S$ represents the stride length. Nevertheless, it does not consider the long-term repeated pattern among subseries.

Based on the above analysis, we find that most Transformer-based methods struggle to simultaneously effectively model local and global correlations (global-local semantic information) and capture long- and short-term repetitive patterns (periodic semantic information) in sequences. Furthermore, it is worth noting that the majority of the aforementioned methods fail to achieve lower complexity than RNN-based methods. For a comprehensive comparison of the theoretical time and memory complexity, as well as experimental results, please refer to Appendix C.

## B  Inspiration and Our Approach

RNNs, such as LSTM and GRU, have a significant advantage in capturing semantic information due to their recurrent structure, as discussed in Appendix A.1. Yet RNNs still have some drawbacks. In addition to the well-known issues of gradient vanishing/exploding [Pascanu et al., 2013]. This paper focuses on analyzing the information forgetting problem (Appendix B.2) and the difficulty in parallelizing RNN (Appendix C). In this section, we will analyze RNNs using LSTM and GRU as representatives, and introduce the inspiration we draw from during the process of designing our model.

### B.1  LSTM VS GRU

LSTM incorporates three gates (input gate, output gate, and forget gate), while GRU utilizes two gates (reset gate and update gate). The complex gating mechanism of LSTM enables it to capture and learn intricate patterns, making it effective for processing longer sequences. On the other hand, GRU

performs well on shorter sequences and requires less computational and memory resources compared to LSTM. It should be noted that, as shown in Equation 7, the computation of $\widetilde{h}_t$ in GRU is dependent on the completion of $r_t$ computation. This undoubtedly slows down the computation that involves the input information $x$. Therefore, in terms of actual running time, GRU may not necessarily have an advantage over LSTM, as detailed in Appendix C.2.

## B.2  The information forgetting problem of RNNs

When Recurrent Neural Networks (RNNs) process sufficiently long sequences, the information contained in the earlier parts of the sequence may be forgotten within the recurrent structure as it passes through multiple cells. To verify this phenomenon, we conducted a concrete experiment. We conducted a benchmark experiment using the ECL dataset, with a historical sequence length of 720 and a prediction horizon of 720, and the description of the ECL dataset can be found in Appendix E.

To eliminate the influence caused by the split of training, validation and test sets, we partitioned the dataset with a fixed input sequence length of 720 and a prediction horizon of 720. Furthermore, we conducted experiments with varying input sequence lengths using LSTM and GRU models. For the experiments, we used the last $m = \{24, 48, 72, \ldots, 696, 720\}$ points near the prediction time as input, and the results of these experiments are shown in Figure 7.

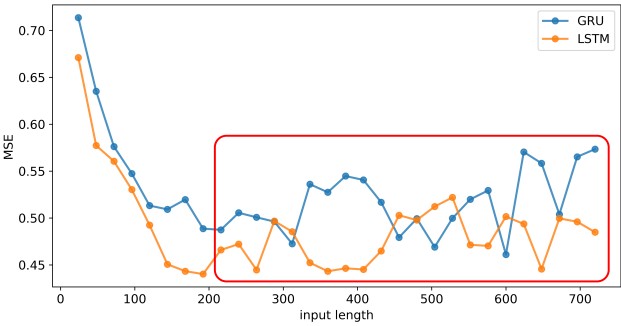

Figure 7: The issue of information forgetting in LSTM and GRU.

Observing Figure 7, on the one hand, we can see that when the input sequence length is relatively short (not within the red box), the prediction performance of LSTM and GRU gradually improves as sequence length increases. This indicates that longer input sequences can provide more contextual information, which can enhance the model's prediction performance and is consistent with previous research findings [Liu et al., 2021, Zeng et al., 2023]. On the other hand, the part within the red box indicates that when longer historical sequences are fed into LSTM and GRU, their prediction performance does not continue to improve. This suggests that introducing earlier information does not provide significant benefits for the models, which in turn confirms the problem of information forgetting in RNNs.

## B.3  Inspiration and WITRAN

Our proposed Water-wave Information Transmission (WIT) framework utilizes input rearrangement to capture both long- and short-term repeating patterns in the sequence more directly and efficiently, while avoiding the processing of excessively long sequences, as described in Section 3.1. This successfully addresses the limitations of RNNs, such as the problems of exploding and vanishing gradients, as well as information forgetting. Furthermore, through the design of Recurrent Acceleration Network (RAN), we enable parallel computation of the entire sequence, as described in Section 3.3. For a comprehensive understanding of RAN, we provide detailed code and analysis of RAN in Appendix D. In this subsection, we focus on introducing the inspiration behind the important components of WIT.

The specific structure of WIT has been detailed in Section 3.2. The inspiration behind the design of Horizontal Vertical Gated Selective Unit (HVGSU) and Gated Selective Cell (GSC) comes from the advantages of LSTM and GRU. HVGSU serves two major purposes: 1) capturing both long- and short-term periodic semantic information in the sequence, and 2) modeling these two types

of periodic semantic information simultaneously for each time step. To achieve these objectives, two separate GSCs are established in the HVGSU, each responsible for the fusion and selection of long-term and short-term information in both directions.

For each GSC cell, to selectively incorporate fused information into the periodic semantic information in each direction, we draw inspiration from the reset gate of GRU and design a selection gate to control the retention of original information and the addition of new information. Furthermore, we are inspired by the output gate of LSTM and design our output gate to output the updated information. To reduce computational costs while maintaining sufficient learning capacity, we draw inspiration from GRU and only utilize these two gates, resulting in smaller computational and memory overheads. However, we address the problem of waiting for information to be computed between gate calculations in GRU by parallelizing the gate calculation vectors with the input $x$, which greatly accelerated the computation speed of the cell. In summary, we analyze and draw inspiration from the strengths of LSTM and GRU, and incorporated them into the design of our GSC.

### B.4 WITRAN VS other RNN-based models

There are also several other RNN models, such as DilatedRNN [Chang et al., 2017], SlicedRNN [Yu and Liu, 2018], and RIMP-LSTM [Shen et al., 2018], which also have certain advantages in handling sequences. To further compare the differences between **WITRAN** and these methods, we provide the comparison in Table 6:

Table 6: Advantages of **WITRAN** compared to **other RNN methods**.

| Advantages | DilatedRNN | SlicedRNN | RIMP-LSTM | WITRAN (ours) |
|---|---|---|---|---|
| Efficiently (1 layer) model global correlations | ✓ | ✓ | ✓(1) | ✓(1) |
| Special design to capture long-term repetitive patterns | ✓ | ✗ | ✗ | ✓ |
| Using 1 layer to capture long- and short-term repetitive patterns simultaneously | ✗ | ✗ | ✗ | ✓ |
| Well solve the gradient vanishing/exploding problem of RNN | ✓ | ✓ | ✓ | ✓ |

**(1) Efficiently (1 layer) model global correlations** (a) When the dilations of DilatedRNN does not include the value 1, multiple layers need to be constructed to extract global correlations. (b) SlicedRNN improves efficiency to some extent by parallel processing of minimum subsequences, but it still requires the introduction of multiple layers to capture the global correlations of the sequence.

**(2) Special design to capture long-term repetitive patterns** (a) SlicedRNN is unable to capture long-term repetitive patterns among elements of sub-sequences. (b) Although RIMP-LSTM incorporates Residual Paths and Residual Sum Unit designs, it still cannot effectively extract long-term repetitive patterns.

**(3) Using 1 layer to capture long- and short-term repetitive patterns simultaneously** (a) DilatedRNN can capture long- and short-term repetitive patterns, but it requires the use of multiple layers to achieve this. (b) SlicedRNN and RIMP-LSTM are not particularly adept at handling long-term repetitive patterns, as mentioned in **(2)**.

**(4) Well solve the gradient vanishing/exploding problem of RNN** (a) In DilatedRNN [Chang et al., 2017], it states and provides formal proof that reducing the length of information paths between time steps can prevent the issues of gradient vanishing/exploding. DilatedRNN, SlicedRNN, and WITRAN tackle this problem by reducing the length of information transmission paths. (b) RIMP-LSTM addresses this issue by the designs of Residual Paths and Residual Sum Units.

## C    Time and Memory Consumption of RNNs and Transformer

In this section, we derive the complexities of LSTM, GRU and Vanilla Transformer respectively, and analyze their actual running speeds and memory usage through experiments.

### C.1    The theoretical computational complexity

The structure diagrams of LSTM and GRU are shown in Figure 6, while the structure of Vanilla Transformer can be found in paper [Vaswani et al., 2017]. Assuming the sequence length is $L$, the

input dimension is $d_{\text{in}}$ and the hidden layer dimension is $d_{\text{model}}$, and considering the case where each model has only one layer, the derivation process is as follows:

**GRU**  GRU includes two gates and three matrix multiplication operations, as mentioned in Appendix A.1. Each matrix multiplication operation has a complexity of $d_{\text{model}} \times (d_{\text{model}} + d_{\text{in}})$, although there are also $\odot$ operations in GRU, and the complexity of element-wise multiplication is $d_{\text{model}}$, which is much smaller than the complexity of matrix multiplication operation. Therefore, the complexity of one GRU unit can be simplified as $3 \times d_{\text{model}} \times (d_{\text{model}} + d_{\text{in}})$. When GRU processes the entire sequence, it needs to perform $L$ sequential computations, therefore, the overall computational complexity can be summarized as: $3 \times L \times d_{\text{model}} \times (d_{\text{model}} + d_{\text{in}})$.

**LSTM**  LSTM includes three gates and four matrix multiplication operations, as mentioned in Appendix A.1. Similar to GRU, the computational complexity of LSTM can also be summarized as: $4 \times L \times d_{\text{model}} \times (d_{\text{model}} + d_{\text{in}})$.

**Transformer**  Taking the single-head attention mechanism in Transformer as an example, in the calculation of the attention mechanism, the each linear transformations of $Query(Q)$, $Key(K)$ and $Value(V)$ require matrix multiplication operation of size $d_{\text{model}} \times d_{\text{model}}$. In the calculation of the attention weights, the $Q$ and transpose of $K$ are multiplied together, and since the size of $Q$ and $K$ are $L \times d_{\text{model}}$, the complexity of computing the attention weights is $L \times d_{\text{model}} \times L$, and the size of attention weights is $L \times L$. Similarly, when we multiply the attention weights with $V$ in the computation, the complexity is $L \times L \times d_{\text{model}}$. The output from the attention matrix multiplication is further linearly transformed, and thus requires an additional $d_{\text{model}} \times d_{\text{model}}$ complexity. To summarize, the overall complexity of the single-headed attention computation in Transformer can be expressed as: 1) Linear transformations of $Q$, $K$ and $V$: $3 \times L \times d_{\text{model}} \times d_{\text{model}}$. 2) Calculation of attention weights: $L \times d_{\text{model}} \times L$. 3) Matrix multiplication of attention weights and $V$: $L \times L \times d_{\text{model}}$. 4) Linear transformation of the output: $L \times d_{\text{model}} \times d_{\text{model}}$. The total complexity of the single-headed attention computation in Transformer is $3 \times L \times d_{\text{model}} \times d_{\text{model}} + L \times d_{\text{model}} \times L + L \times L \times d_{\text{model}} + L \times d_{\text{model}} \times d_{\text{model}} = 4 \times L \times d_{\text{model}}^2 + 2 \times L^2 \times d_{\text{model}}$. It should be noted that the complexity improvements in Transformer-based methods only apply to the $L^2$ regularization term, as the linear transformations are necessary for any Transformer-based model. In addition, assuming the dimension of the Feed-Forward Networks (FFN) in Transformer is $d_{\text{ff}}$, the first layer of the fully connected network has a complexity of $L \times d_{\text{model}} \times d_{\text{ff}}$, while the second layer has a complexity of $L \times d_{\text{ff}} \times d_{\text{ff}}$, and the total complexity of FFN is $L \times d_{\text{model}} \times d_{\text{ff}} + L \times d_{\text{ff}}^2$. It should be noted that the input and output transformations in Transformer still require a certain level of complexity. On the hand, as it needs to transform the input data from dimension $d_{\text{in}}$ to dimension $d_{\text{model}}$, the input transformation in Transformer requires at least a $L \times d_{\text{in}} \times d_{\text{model}}$ complexity. On the other hand, as it needs to transform the input data from dimension $d_{\text{ff}}$ to dimension $d_{\text{model}}$, the output transformation requires a $L \times d_{\text{ff}} \times d_{\text{model}}$ complexity. Therefore, the overall complexity of Transformer can be summarized as: $L \times d_{\text{in}} \times d_{\text{model}} + 4 \times L \times d_{\text{model}}^2 + 2 \times L^2 \times d_{\text{model}} + L \times d_{\text{model}} \times d_{\text{ff}} + L \times d_{\text{ff}}^2 + L \times d_{\text{ff}} \times d_{\text{model}}$. It should be noted that in most cases for Transformer, $d_{\text{ff}}$ is larger than $d_{\text{model}}$.

**Summary**  In time series prediction tasks, the input dimension $d_{\text{in}}$ is usually much smaller than the model dimension $d_{\text{model}}$. Based on the analysis conducted earlier, we can observe that the computational complexity of the encoder part of most Transformer-based methods is generally higher than that of LSTM and GRU. This is because Transformer includes many complex operations such as matrix multiplication and linear transformation. Although there have been improvements made to the attention mechanism, these modifications only affect a small portion of the overall complexity which remains high. Therefore, compared to LSTM and GRU, the computational complexity of Transformer is higher.

## C.2   The practical time and memory consumption

In practical situations, LSTM can perform computations in parallel without the need to wait for information from all four matrix operations. On the other hand, in GRU, the first two matrix operations can be parallelized, but the third matrix operation has a dependency on the completion of the previous operations, as mentioned in Appendix A.1. Therefore, the actual runtime of GRU may be longer than

---

**Algorithm 1** GRU based on Python

---

```python
import torch
import torch.nn as nn
import torch.nn.functional as F

class Manual_GRU(nn.Module):
    def __init__(self, input_size, hidden_size, dropout):
        super(Manual_GRU, self).__init__()
        self.hidden_size = hidden_size
        self.dropout = dropout
        self.gates = nn.Linear(input_size+hidden_size, hidden_size*2)
        self.hidden_transform = nn.Linear(input_size+hidden_size, hidden_size)
        self.sigmoid = nn.Sigmoid()
        self.tanh = nn.Tanh()
        for param in self.parameters():
            if param.dim() > 1:
                nn.init.xavier_uniform_(param)

    def forward(self, x):
        batch_size = x.size(0)
        seq_len = x.size(1)
        h = torch.zeros(batch_size, self.hidden_size).to(x.device)
        y_list = []
        for i in range(seq_len):
            update_gate, reset_gate = self.gates(torch.cat([x[:, i, :], h], dim=−1)).chunk(2, −1)
            update_gate, reset_gate = (self.sigmoid(gate) for gate in (update_gate, reset_gate))
            candidate_hidden = self.tanh(self.hidden_transform(torch.cat([x[:, i, :], reset_gate * h], dim=−1)
                ))
            h = (1−update_gate) * h + update_gate * candidate_hidden
            y_list.append(h)
        output = F.dropout(torch.stack(y_list, dim=1), self.dropout, self.training)
        return output, h
```

---

that of LSTM. To verify this, we compared the processing time and memory usage of a single-layer LSTM and a single-layer GRU on sequences of the same length using Python implementation, as shown in Figure 8.

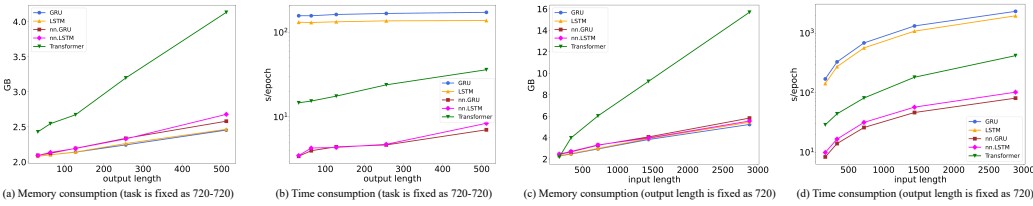

Figure 8: Time and memory consumption of RNNs and Transformer.

Transformer has the advantage of parallelizing the computation of correlations for a sequence of length L, which greatly improves its speed compared to GRU/LSTM. It is important to note that this comparison is not entirely fair, as Transformer can parallelize at the sequence level, whereas RNNs can parallelize at the batch level. Each approach has its own strengths and is suited for different scenarios.

To ensure a fair comparison of runtime and memory usage, it is important to consider the conditions under which the models are compared. Following the experimental setup in FiLM [Zhou et al., 2022b], we can ensure that the hidden state dimension and batch size of the models being compared are the same. This allows us to evaluate whether the models can be trained on devices with low memory capacity. For time evaluation, we need to ensure that the hidden state dimension is the same and the memory utilization is similar, so that memory resources are not wasted. In terms of time comparison between Transformer and GRU/LSTM, although GRU/LSTM are slower than

Transformer under the same batch conditions in Python implementation, they can still speed up their parallel processing by increasing batch size while ensuring similar memory usage to Transformer. At the same time, because the computation process of GRU and LSTM is much simpler than that of Transformer, they can be more easily implemented in C++ and integrated into PyTorch's library as nn.GRU/nn.LSTM, which is faster. For comparison, we compared the memory and time costs of these five methods, as shown in Figure 8. The Python implementation of GRU and LSTM code is shown in Algorithm 1 and Algorithm 2.

---

**Algorithm 2** LSTM based on Python

---

```
import torch
import torch.nn as nn
import torch.nn.functional as F

class Manual_LSTM(nn.Module):
    def __init__(self, input_size, hidden_size, dropout):
        super(Manual_LSTM, self).__init__()
        self.hidden_size = hidden_size
        self.dropout = dropout
        self.gates = nn.Linear(input_size+hidden_size, hidden_size * 4)
        self.sigmoid = nn.Sigmoid()
        self.tanh = nn.Tanh()
        for param in self.parameters():
            if param.dim() > 1:
                nn.init.xavier_uniform_(param)

    def forward(self, x):
        batch_size = x.size(0)
        seq_len = x.size(1)
        h, c = (torch.zeros(batch_size, self.hidden_size).to(x.device) for _ in range(2))
        y_list = []
        for i in range(seq_len):
            forget_gate, input_gate, output_gate, candidate_cell = self.gates(torch.cat([x[:, i, :], h], dim=−1))
                .chunk(4, −1)
            forget_gate, input_gate, output_gate = (self.sigmoid(g)
                for g in (forget_gate, input_gate, output_gate))
            c = forget_gate * c + input_gate * self.tanh(candidate_cell)
            h = output_gate * self.tanh(c)
            y_list.append(h)
        output = F.dropout(torch.stack(y_list, dim=1), self.dropout, self.training)
        return output, h, c
```

---

# D   The Design Process and Implementation Code of Recurrent Acceleration Network (RAN)

In this section, we will provide a detailed explanation of the implementation approach (Appendix D.1) and specific code of RAN (Appendix D.2).

## D.1   Implementation approach of RAN

In Section 3.3, we introduced the Recurrent Acceleration Network (RAN) as part of our overall approach. However, we encountered a challenge when the number of rows processed by RAN is smaller than the number of columns. In such cases, if the number of slices processed exceeds the number of rows, the intermediate variables of the slices become fully occupied. To ensure the continuity of operations, it becomes necessary to continuously shift and reset these variables. However, we found that this reset operation was time-consuming and hindered the expected acceleration benefits of RAN. To overcome this issue, we made a simple adjustment by setting the size of the slice to the maximum value of the rows and columns. While this may introduce additional space complexity, it maintains the length of the slice and eliminates the need for a reset operation. As a result, the

processing time is significantly reduced. The comparison diagram between RAN-min and RAN-max is shown in Figure 9.

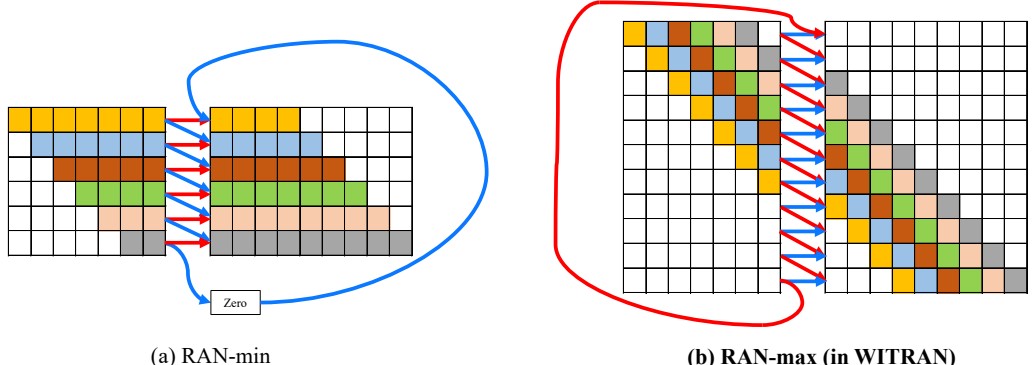

(a) RAN-min                    (b) RAN-max (in WITRAN)

Figure 9: The figure shows two designs of RANs side by side. The left one represents the case where the number of slices is the minimum value between the number of rows and columns, while the right one represents the case where the number of slices is the maximum value between the number of rows and columns.

## D.2 Specific code of RAN

The specific code of RAN can be found in Algorithm 3. Due to its long content, we place it at the end of the Appendix.

## E Dataset Details

In this section, we will provide a detailed introduction of the datasets used in this paper. (1) *Electricity*[2] (*ECL*) contains the hourly electricity consumption of 321 customers from 2012 to 2014. (2) *ETT* [Zhou et al., 2021] contains the load and oil temperature, which was recorded every 15 minutes between July 2016 and July 2018 from electricity transformers. (3) *Traffic*[3] contains the hourly data the road occupancy rates measured by different sensors on San Francisco Bay area freeways, collected from California Department of Transportation. (4) *Weather*[4] contains 21 meteorological indicators (such as air temperature, humidity, etc.) and was recorded every 10 minutes for 2020 whole year. They are all split into the training set, validation set and test set by the ratio of 6:2:2 during modeling.

Due to the different original acquisition granularities of each dataset, in order to ensure that they contain the same semantic information on the same task, in this paper, our experiments are conducted by aggregating them into one hour. Table 7 summarizes all the features of the four benchmark datasets. The target value of ETT is 'oil temperature (OT)', the target value of ECL is 'MT_320', the target value of Traffic is 'Node_862' and the target value of Weather is 'wet_bulb'.

Table 7: The details of datasets.

| Dataset | Sample Length | Dimension | Usage Frequency | Recorded Frequency |
|---------|---------------|-----------|-----------------|--------------------|
| ETT     | 17420         | 8         | 1h              | 15min              |
| ECL     | 26304         | 322       | 1h              | 1h                 |
| Traffic | 17544         | 863       | 1h              | 1h                 |
| Weather | 35064         | 22        | 1h              | 10min              |

---

[2] https://archive.ics.uci.edu/ml/datasets/ElectricityLoadDiagrams20112014
[3] http://pems.dot.ca.gov
[4] https://www.bgc-jena.mpg.de/wetter/

## F   Experiment Setup Details

Our method and all the baseline methods are trained with the L2 loss, using the ADAM [Kingma and Ba, 2014] optimizer with an initial learning rate of 10-3. Batch size is set to 32. The training process is early stopped after 5 epochs if there is no loss degradation on the validation set and the max epochs is 25. We save all the models with the lowest loss on the validation set for the final testing.

To ensure a fair comparison of each model's performance, we set the same search space for the common parameters included in each model. This ensures that each model can achieve its best performance under the same conditions for comparison, with the seed set to 2023.

Regarding the private parameters of each model, we set the corresponding search space according to the specific descriptions given in their respective papers. In addition, it should be emphasized that the performance of Pyraformer shown in this paper is the best between the two prediction designs of the model, the performance of FEDformer shown here is the best between FEDformer-w and FEDformer-f, the performance of DLinear shown here is the best among DLinear, NLinear and Linear, and the performance of MICN is the best between MICN-regre and MICN-mean.

The mean square error (MSE) and mean absolute error (MAE) are used as metrics. All experiments are repeated 5 times and the mean of the metrics is reported as the final results in Table 3 and Table 4.

All the deep learning networks are implemented using PyTorch [Paszke et al., 2019] and conducted on NVIDIA RTX A4000 16GB GPUs.

The search space for the common parameters is as follows: (1) $d_{\text{model}}$ is set to $\{32, 64, 128, 256, 512, 1024\}$. (2) $e_{\text{layer}}$ and $d_{\text{layer}}$ are set to $\{1, 2, 3\}$. (3) $n_{\text{head}}$ is set to $\{1, 2, 4, 8\}$ and $d_{\text{ff}}$ is four times as large as $d_{\text{model}}$. Regarding the private parameters of our model, $norm$ is set to $\{0, 1\}$ and $C$ represents the period is set to $\{12, 24, 48\}$ and it is required that the length $H$ of the input sequence is divisible by $C$. Furthermore, it is important to note that adjustments should be made to $C$ for different tasks to ensure its ability to adequately represent the period.

## G   Ablation Study

In Section 4.3, we presented the results of our ablation experiments to demonstrate the effectiveness of each component in the Water-wave Information Transmission (WIT) framework. We design five variant methods and evaluate their performance on four datasets. The experimental results are summarized in Table 8, and we will provide a detailed analysis of these results in this section. It is important to note that the role of the Recurrent Acceleration Network (RAN) is primarily to improve computational efficiency, as demonstrated in Section 4.2, and it does not significantly impact the model's accuracy. Therefore, our focus in this section will be on discussing the design and role of the WIT framework.

### G.1   Impact of forecasting module

Our proposed forecasting module is also a part of the model design, and its goal is to effectively capture periodic semantic information and global-local semantic information obtained through the Water-wave Information Transmission (WIT) framework. Our specific design has been described in detail in Section 3.4. To verify that we can fully utilize the relevant information by combining operations for pairwise prediction, we changed the combination prediction method of this module to a simple way of first concatenating the horizontal hidden state and the vertical hidden state, and then using the fully connected prediction module for prediction. We named this variant WITRAN-FC.

The comparison of results between WITRAN and WITRAN-FC in Table 8 shows that when we remove the combining operations for pairwise prediction and replace it with ordinary concatenation and fully connected operations, even though the same information is used, WITRAN-FC has difficulty discovering the most relevant long-term periodic semantic information for each prediction point. This demonstrates the effectiveness of the combining operations for pairwise prediction in the forecasting module.

Table 8: Results of the **ablation study** on **long-range** and **ultra-long-range** forecasting tasks. A lower MSE or MAE indicates a better prediction. The best results are highlighted in bold.

| Methods | | **WITRAN** | | WITRAN-FC | | WITRAN-2DLSTM | | WITRAN-2DGRU | | WITRAN-LSTM | | WITRAN-GRU | |
|---|---|---|---|---|---|---|---|---|---|---|---|---|---|
| Metric | | MSE | MAE | MSE | MAE | MSE | MAE | MSE | MAE | MSE | MAE | MSE | MAE |
| ECL | 168-168 | **0.2397** | **0.3519** | 0.3005 | 0.4029 | 0.2662 | 0.3736 | 0.2578 | 0.3677 | 0.2749 | 0.3800 | 0.2536 | 0.3640 |
| | 168-336 | **0.2607** | **0.3721** | 0.3425 | 0.4355 | 0.3027 | 0.3996 | 0.2916 | 0.3924 | 0.3164 | 0.4087 | 0.2872 | 0.3863 |
| | 336-336 | **0.2517** | **0.3627** | 0.3242 | 0.4232 | 0.2916 | 0.3896 | 0.2826 | 0.3851 | 0.3151 | 0.4066 | 0.2811 | 0.3805 |
| | 336-720 | **0.3084** | **0.4055** | 0.3566 | 0.4523 | 0.3279 | 0.4262 | 0.3172 | 0.4187 | 0.3435 | 0.4350 | 0.3123 | 0.4153 |
| | 720-720 | **0.2478** | **0.3651** | 0.4544 | 0.4972 | 0.3120 | 0.4184 | 0.2948 | 0.4060 | 0.3362 | 0.4325 | 0.2912 | 0.4009 |
| | 720-1440 | **0.2499** | **0.3727** | 0.4732 | 0.5277 | 0.3426 | 0.4505 | 0.3337 | 0.4440 | 0.3517 | 0.4540 | 0.3296 | 0.4414 |
| | 1440-1440 | **0.2408** | **0.3680** | 0.6433 | 0.6172 | 0.3954 | 0.4876 | 0.3822 | 0.4819 | 0.4164 | 0.5006 | 0.3864 | 0.4843 |
| | 1440-2880 | **0.3359** | **0.4383** | 0.7406 | 0.6906 | 0.5936 | 0.6215 | 0.5680 | 0.6077 | 0.5583 | 0.6005 | 0.5859 | 0.6173 |
| Traffic | 168-168 | **0.1377** | **0.2051** | 0.1541 | 0.2382 | 0.1918 | 0.2811 | 0.1824 | 0.2699 | 0.2065 | 0.2935 | 0.1846 | 0.2666 |
| | 168-336 | **0.1321** | **0.2059** | 0.1572 | 0.2503 | 0.2070 | 0.2975 | 0.1945 | 0.2837 | 0.2201 | 0.3058 | 0.2059 | 0.2887 |
| | 336-336 | **0.1306** | **0.2054** | 0.1549 | 0.2455 | 0.1923 | 0.2846 | 0.1808 | 0.2732 | 0.2029 | 0.2935 | 0.1840 | 0.2715 |
| | 336-720 | **0.1391** | **0.2175** | 0.1725 | 0.2611 | 0.2087 | 0.3000 | 0.2042 | 0.2958 | 0.2197 | 0.3080 | 0.2072 | 0.2940 |
| | 720-720 | **0.1408** | **0.2191** | 0.1722 | 0.2622 | 0.2157 | 0.3048 | 0.2097 | 0.2997 | 0.2242 | 0.3123 | 0.2150 | 0.3023 |
| | 720-1440 | **0.1672** | **0.2449** | 0.2178 | 0.2979 | 0.2404 | 0.3178 | 0.2450 | 0.3245 | 0.2586 | 0.3346 | 0.2501 | 0.3264 |
| | 1440-1440 | **0.1543** | **0.2325** | 0.2150 | 0.2954 | 0.2343 | 0.3148 | 0.2340 | 0.3151 | 0.2406 | 0.3199 | 0.2469 | 0.3249 |
| | 1440-2880 | **0.1425** | **0.2333** | 0.2072 | 0.3082 | 0.2060 | 0.3081 | 0.2134 | 0.3182 | 0.2113 | 0.3116 | 0.2129 | 0.3162 |
| ETTh$_1$ | 168-168 | **0.1105** | **0.2589** | 0.1143 | 0.2649 | 0.1132 | 0.2624 | 0.1120 | 0.2622 | 0.1141 | 0.2642 | 0.1141 | 0.2643 |
| | 168-336 | **0.1189** | **0.2714** | 0.1211 | 0.2753 | 0.1160 | 0.2694 | 0.1172 | 0.2710 | 0.1191 | 0.2733 | 0.1173 | 0.2708 |
| | 336-336 | **0.1112** | **0.2638** | 0.1167 | 0.2715 | 0.1125 | 0.2668 | 0.1125 | 0.2668 | 0.1147 | 0.2696 | 0.1138 | 0.2676 |
| | 336-720 | **0.1494** | **0.3092** | 0.1347 | 0.2933 | 0.1317 | 0.2905 | 0.1314 | 0.2902 | 0.1310 | 0.2902 | 0.1317 | 0.2906 |
| | 720-720 | **0.1296** | **0.2868** | 0.1515 | 0.3109 | 0.1438 | 0.3037 | 0.1436 | 0.3034 | 0.1450 | 0.3048 | 0.1513 | 0.3111 |
| | 720-1440 | **0.1331** | **0.2943** | 0.1807 | 0.3378 | 0.1759 | 0.3361 | 0.1764 | 0.3375 | 0.1766 | 0.3361 | 0.1791 | 0.3391 |
| | 1440-1440 | **0.1304** | **0.2902** | 0.2627 | 0.4096 | 0.1861 | 0.3439 | 0.2063 | 0.3616 | 0.1969 | 0.3562 | 0.2123 | 0.3656 |
| | 1440-2880 | **0.1850** | **0.3452** | 0.2655 | 0.4134 | 0.2485 | 0.3998 | 0.2463 | 0.3925 | 0.2444 | 0.3923 | 0.2486 | 0.3962 |
| ETTh$_2$ | 168-168 | **0.2389** | **0.3813** | 0.2507 | 0.3859 | 0.2577 | 0.4012 | 0.2567 | 0.3998 | 0.2593 | 0.4013 | 0.2556 | 0.3980 |
| | 168-336 | **0.2277** | **0.3778** | 0.2377 | 0.3896 | 0.2791 | 0.4193 | 0.2791 | 0.4193 | 0.2818 | 0.4211 | 0.2690 | 0.4108 |
| | 336-336 | **0.2432** | **0.3922** | 0.2928 | 0.4273 | 0.2614 | 0.4085 | 0.2621 | 0.4084 | 0.2680 | 0.4130 | 0.2604 | 0.4058 |
| | 336-720 | **0.2373** | **0.3888** | 0.2948 | 0.4355 | 0.3051 | 0.4434 | 0.2815 | 0.4253 | 0.3113 | 0.4458 | 0.2798 | 0.4239 |
| | 720-720 | **0.2635** | **0.4018** | 0.3243 | 0.4548 | 0.2990 | 0.4369 | 0.2848 | 0.4283 | 0.3096 | 0.4428 | 0.2832 | 0.4262 |
| | 720-1440 | **0.2915** | **0.4289** | 0.3212 | 0.4598 | 0.3308 | 0.4659 | 0.3396 | 0.4723 | 0.3241 | 0.4617 | 0.3446 | 0.4759 |
| | 1440-1440 | **0.2815** | **0.4220** | 0.4162 | 0.5145 | 0.3950 | 0.4962 | 0.4214 | 0.5144 | 0.4199 | 0.5132 | 0.4243 | 0.5193 |
| | 1440-2880 | **0.3280** | **0.4585** | 0.5532 | 0.5959 | 0.6846 | 0.6639 | 0.6554 | 0.6530 | 0.6581 | 0.6531 | 0.7473 | 0.7006 |
| Weather | 168-168 | **0.2050** | 0.3338 | 0.2066 | **0.3324** | 0.2180 | 0.3419 | 0.2199 | 0.3437 | 0.2194 | 0.3435 | 0.2211 | 0.3462 |
| | 168-336 | **0.2197** | **0.3470** | 0.2200 | 0.3585 | 0.2580 | 0.3789 | 0.2723 | 0.3871 | 0.2596 | 0.3794 | 0.2668 | 0.3843 |
| | 336-336 | **0.2163** | **0.3482** | 0.2259 | 0.3586 | 0.2472 | 0.3730 | 0.2613 | 0.3816 | 0.2462 | 0.3726 | 0.2501 | 0.3732 |
| | 336-720 | **0.2054** | **0.3424** | 0.2125 | 0.3497 | 0.2497 | 0.3840 | 0.2775 | 0.4004 | 0.2473 | 0.3796 | 0.2859 | 0.4099 |
| | 720-720 | **0.2008** | **0.3417** | 0.2104 | 0.3458 | 0.2564 | 0.3843 | 0.2861 | 0.4030 | 0.2633 | 0.3873 | 0.2772 | 0.3990 |
| | 720-1440 | **0.1872** | **0.3312** | 0.1984 | 0.3397 | 0.2301 | 0.3676 | 0.2853 | 0.4156 | 0.2376 | 0.3751 | 0.2737 | 0.4063 |
| | 1440-1440 | **0.1907** | **0.3366** | 0.2084 | 0.3507 | 0.2419 | 0.3832 | 0.2671 | 0.4070 | 0.2454 | 0.3847 | 0.2631 | 0.4013 |
| | 1440-2880 | **0.1769** | **0.3257** | 0.2148 | 0.3570 | 0.2308 | 0.3735 | 0.2460 | 0.3887 | 0.2317 | 0.3736 | 0.2305 | 0.3751 |

## G.2 Impact of information fusion and selection

As information fusion and selection are completed together in the Gated Selective Cell (GSC) we designed, we need to verify the effectiveness of GSC, that is, the effectiveness of information fusion and selection. Here, we replaced the cells in the two directions of Horizontal Vertical Gated Selective Unit (HVGSU) with LSTM and GRU respectively, and named them WITRAN-2DLSTM and WITRAN-2DGRU. It should be noted that since the information between the LSTM/GRU cell in the two directions is not fused at this time, the information in the last row of the horizontal direction cannot contain global information, so we cannot use the proposed forecasting module for application, but only use the FC method mentioned in the previous subsection for prediction. It should be noted that the comparison between WITRAN-2DLSTM/WITRAN-2DGRU and WITRAN-FC reflects the impact of information redundancy on experimental performance, while the comparison between WITRAN-2DLSTM/WITRAN-2DGRU and WITRAN reflects the impact of information fusion and selection on experimental performance.

The comparison of results between WITRAN-2DLSTM/WITRAN-2DGRU and WITRAN-FC in Table 8 demonstrates that too much information redundancy may be detrimental to the model's prediction performance, because for WITRAN-FC, the horizontal hidden states it receives contain more information as they go further down, and they also contain all the information in the horizontal direction above them. The same applies to the vertical direction, where the further to the right the hidden state is, the more information it contains. Furthermore, the information captured by WITRAN-2DLSTM and WITRAN-2DGRU is independent among different horizontal rows and among different vertical columns, and the semantic information between horizontal and vertical

hidden state are different. Therefore, WITRAN-2DLSTM and WITRAN-2DGRU are not heavily influenced by redundant information. In addition, the comparison of results between WITRAN and WITRAN-2DLSTM/WITRAN-2DGRU shows that information fusion and selection can effectively extract relevant information, which demonstrates the necessity of the GSC design in WIT.

### G.3 Impact of bi-granular information transmission

In the WITRAN we designed, information can be propagated in two directions, and because the time granularity of sequence propagation in these two directions is different, their periodic semantic information is different. In order to verify the effectiveness of the two independent GSCs set up in HVGSU to capture long- and short-term periodic semantic information, we replaced the cells in the two directions of WITRAN-2DLSTM/WITRAN-2DGRU with a single LSTM/GRU cell, and uniformly used the hidden state information of one point to transmit information in both directions. It should be noted that at this time, the information propagated horizontally and vertically by the model is completely the same. Sure, we can name the current models as WITRAN-LSTM/WITRAN-GRU.

The comparison of results between WITRAN-2DLSTM/WITRAN-2DGRU and WITRAN-LSTM/WITRAN-GRU in Table 8 demonstrates that failing to distinguish between long- and short-term periodic semantic information can have a significant negative impact on the performance of prediction tasks.

### G.4 Summary

In this section, we focused on the importance of each building module of WIT. During the experiments, we gradually removed the building blocks we designed, and it was clear that the performance decreased significantly. This fully demonstrates the rationality and effectiveness of the WIT framework we designed. Specifically, the summary of WIT is as follows: (1) The fusion and selection design of GSC enables both long- and short-term semantic information to be captured for each historical input data point. (2) Setting up independent cells in both directions enables to extract semantic information in the long- and short-term respectively. (3) The specially designed combined operation is able to make fuller use of the captured local and global semantic information while ensuring that the information is not redundant.

## H  Experiment Error Bars

We saved all models and used the model with the lowest validation loss for the final testing. We repeated this process 5 times and calculated the error bars for all models to compare their robustness on long-range and ultra-long-range tasks, as shown in Table 9. It can be seen from Table 9 that the overall performance of the proposed WITRAN is better than other state-of-the-art baseline models, indicating the effectiveness of our approach.

## I  Model Analysis

In this section, we will analyze the parameter sensitivity (Appendix I.1) of our proposed model and provide a detailed explanation (Appendix I.2) of the time and memory consumption in Section 4.2.

### I.1 Parameter sensitivity

Regarding the private parameters of our model, $norm$ is set to $\{0, 1\}$ and $C$ represents the period is set to $\{12, 24, 48\}$ and it is required that the length $H$ of the input sequence is divisible by $C$. In this subsection, we will explain the choices of these parameters and their impact on the model's predictions.

**The impact of** $norm$    For different prediction tasks on different datasets, we determined the value of $norm$ through validation set. To verify that the value of $norm$ conforms to the distribution of the datasets, we conducted experiments on the distribution of the training and validation sets for different prediction tasks, the results and the value of $norm$ are shown in Table 10. We noticed that although there may be a significant difference in the mean values of the data between the training and validation

Table 9: MSE and MAE with error bars (Mean and STD) for WITRAN and all the baseline methods for long-range and ultra-long-range forecasting. All experiments are repeated 5 times.

| | Datasets | ECL | | Traffic | | ETTh1 | | ETTh2 | | Weather | |
|---|---|---|---|---|---|---|---|---|---|---|---|
| | Metric | MSE | MAE | MSE | MAE | MSE | MAE | MSE | MAE | MSE | MAE |
| **WITRAN(ours)** | 168-168 | 0.2397±0.00859 | 0.3519±0.00601 | 0.1377±0.00231 | 0.2051±0.00300 | 0.1105±0.00082 | 0.2589±0.00128 | 0.2389±0.00615 | 0.3813±0.00566 | 0.2050±0.00428 | 0.3338±0.00483 |
| | 168-336 | 0.2607±0.00926 | 0.3721±0.00783 | 0.1321±0.00327 | 0.2059±0.00359 | 0.1189±0.00325 | 0.2714±0.00439 | 0.2277±0.00805 | 0.3778±0.00772 | 0.2197±0.00629 | 0.3470±0.00328 |
| | 336-336 | 0.2517±0.00946 | 0.3627±0.00824 | 0.1306±0.00080 | 0.2054±0.00161 | 0.1112±0.00161 | 0.2638±0.00221 | 0.2432±0.00303 | 0.3922±0.00470 | 0.2163±0.00520 | 0.3482±0.00910 |
| | 336-720 | 0.3084±0.01255 | 0.4055±0.00807 | 0.1391±0.00019 | 0.2175±0.00202 | 0.1494±0.01004 | 0.3092±0.01220 | 0.2373±0.00815 | 0.3888±0.00558 | 0.2054±0.00322 | 0.3424±0.00328 |
| | 720-720 | 0.2478±0.02129 | 0.3651±0.01580 | 0.1408±0.00109 | 0.2191±0.00018 | 0.1296±0.00220 | 0.2868±0.00265 | 0.2635±0.01346 | 0.4018±0.00935 | 0.2008±0.00374 | 0.3417±0.00159 |
| | 720-1440 | 0.2499±0.00779 | 0.3727±0.00097 | 0.1672±0.00389 | 0.2449±0.00746 | 0.1331±0.00273 | 0.2943±0.00342 | 0.2915±0.02237 | 0.4289±0.01468 | 0.1872±0.00281 | 0.3312±0.00442 |
| | 1440-1440 | 0.2408±0.01817 | 0.3680±0.01533 | 0.1543±0.00570 | 0.2325±0.00944 | 0.1304±0.00425 | 0.2902±0.00503 | 0.2815±0.01771 | 0.4220±0.01177 | 0.1907±0.00622 | 0.3366±0.00648 |
| | 1440-2880 | 0.3359±0.01107 | 0.4383±0.00809 | 0.1425±0.00461 | 0.2333±0.00745 | 0.1850±0.01106 | 0.3452±0.01388 | 0.3280±0.04586 | 0.4585±0.03506 | 0.1769±0.00270 | 0.3257±0.00212 |
| **MICN** | 168-168 | 0.3168±0.01684 | 0.4067±0.01116 | 0.2428±0.00416 | 0.3543±0.00511 | 0.1257±0.01332 | 0.2803±0.01514 | 0.2734±0.01795 | 0.4162±0.01662 | 0.2231±0.00259 | 0.3489±0.00404 |
| | 168-336 | 0.3002±0.00979 | 0.4053±0.00853 | 0.2401±0.00537 | 0.3514±0.00657 | 0.1422±0.02929 | 0.3006±0.03410 | 0.3017±0.04440 | 0.4429±0.03426 | 0.2663±0.00330 | 0.3837±0.00602 |
| | 336-336 | 0.3092±0.00621 | 0.4132±0.00590 | 0.2413±0.00863 | 0.3549±0.00901 | 0.1576±0.03357 | 0.3159±0.03352 | 0.3472±0.05168 | 0.4796±0.03738 | 0.2701±0.00708 | 0.3804±0.00650 |
| | 336-720 | 0.3820±0.01996 | 0.4704±0.01565 | 0.2422±0.00545 | 0.3513±0.00292 | 0.2219±0.05068 | 0.3729±0.04821 | 0.4248±0.04274 | 0.5268±0.02923 | 0.3086±0.00548 | 0.4138±0.00974 |
| | 720-720 | 0.3463±0.01609 | 0.4381±0.00853 | 0.2552±0.01039 | 0.3709±0.01432 | 0.2959±0.07568 | 0.4402±0.06838 | 0.3549±0.03558 | 0.4805±0.02590 | 0.2828±0.00451 | 0.3969±0.00339 |
| | 720-1440 | 1.0460±0.18834 | 0.7765±0.07715 | 0.2876±0.00409 | 0.3916±0.00354 | 0.4640±0.04845 | 0.5836±0.03802 | 0.4922±0.05114 | 0.5649±0.03187 | 0.3999±0.03417 | 0.4848±0.02302 |
| | 1440-1440 | 2.2862±0.36532 | 1.2207±0.13085 | 0.2950±0.00611 | 0.3923±0.00630 | 0.5650±0.12064 | 0.6293±0.07367 | 0.5030±0.04996 | 0.5644±0.02600 | 0.2873±0.01038 | 0.4201±0.00799 |
| | 1440-2880 | 2.8936±1.28817 | 1.3717±0.37787 | 0.2823±0.01489 | 0.3874±0.01046 | 0.7591±0.49493 | 0.7215±0.30100 | 0.5549±0.08677 | 0.5886±0.05173 | 0.3570±0.03915 | 0.4810±0.02942 |
| **TimesNet** | 168-168 | 0.2825±0.00960 | 0.3797±0.00798 | 0.1490±0.00412 | 0.2293±0.00686 | 0.1133±0.00253 | 0.2612±0.00377 | 0.2655±0.01248 | 0.4051±0.01202 | 0.2420±0.00614 | 0.3608±0.00466 |
| | 168-336 | 0.3505±0.00624 | 0.4253±0.00428 | 0.1499±0.00136 | 0.2356±0.00298 | 0.1202±0.00237 | 0.2732±0.00359 | 0.2725±0.00610 | 0.4163±0.00564 | 0.2821±0.01949 | 0.3885±0.01692 |
| | 336-336 | 0.3702±0.05183 | 0.4307±0.02897 | 0.1446±0.00408 | 0.2300±0.00499 | 0.1279±0.01137 | 0.2846±0.01151 | 0.3184±0.07475 | 0.4431±0.04618 | 0.2684±0.01958 | 0.3752±0.01685 |
| | 336-720 | 0.3879±0.02491 | 0.4531±0.01278 | 0.1584±0.00141 | 0.2440±0.00275 | 0.1501±0.00999 | 0.3127±0.01039 | 0.2858±0.00846 | 0.4253±0.00492 | 0.2930±0.01619 | 0.4045±0.01427 |
| | 720-720 | 0.3537±0.02613 | 0.4386±0.01932 | 0.1546±0.00393 | 0.2410±0.00160 | 0.1510±0.01739 | 0.3118±0.01866 | 0.2936±0.02218 | 0.4238±0.01457 | 0.2967±0.02166 | 0.4070±0.01439 |
| | 720-1440 | 0.6119±0.12940 | 0.5962±0.06975 | 0.1882±0.05920 | 0.2656±0.05923 | 0.1391±0.00371 | 0.3049±0.00499 | 0.4186±0.05841 | 0.5092±0.03150 | 0.2407±0.00761 | 0.3694±0.00643 |
| | 1440-1440 | 0.5720±0.14337 | 0.5712±0.07619 | 0.1598±0.00164 | 0.2388±0.01276 | 0.1681±0.01165 | 0.3372±0.01145 | 0.4409±0.07863 | 0.5218±0.04688 | 0.2869±0.01684 | 0.4033±0.01504 |
| | 1440-2880 | 0.7683±0.06278 | 0.6846±0.02955 | 0.1560±0.00700 | 0.2409±0.00845 | 0.2732±0.02157 | 0.4094±0.01088 | 1.5304±0.84959 | 0.9026±0.32013 | 0.2199±0.02223 | 0.3563±0.01606 |
| **PatchTST** | 168-168 | 0.2980±0.00339 | 0.3832±0.00341 | 0.1622±0.01189 | 0.2320±0.00878 | 0.1212±0.00480 | 0.2704±0.00375 | 0.2582±0.00722 | 0.3983±0.00598 | 0.2469±0.00797 | 0.3597±0.00569 |
| | 168-336 | 0.3840±0.01674 | 0.4393±0.01441 | 0.1641±0.01070 | 0.2364±0.00646 | 0.1287±0.00488 | 0.2808±0.00483 | 0.3206±0.01683 | 0.4515±0.01199 | 0.3040±0.00457 | 0.4049±0.00439 |
| | 336-336 | 0.4377±0.04995 | 0.4654±0.02878 | 0.1546±0.00478 | 0.2332±0.00351 | 0.1496±0.01338 | 0.3039±0.01289 | 0.3559±0.03962 | 0.4779±0.02546 | 0.3149±0.01735 | 0.4145±0.01288 |
| | 336-720 | 0.5502±0.05453 | 0.5438±0.02876 | 0.1747±0.01107 | 0.2536±0.00686 | 0.2092±0.01996 | 0.3659±0.01425 | 0.4936±0.03701 | 0.5592±0.02042 | 0.4358±0.01861 | 0.4937±0.01341 |
| | 720-720 | 0.5927±0.10039 | 0.5742±0.04460 | 0.1543±0.00528 | 0.2441±0.00599 | 0.2178±0.02162 | 0.3694±0.01528 | 0.5243±0.04377 | 0.5745±0.02768 | 0.5701±0.10231 | 0.5491±0.03951 |
| | 720-1440 | 0.8243±0.06568 | 0.6704±0.02131 | 0.1904±0.01783 | 0.2685±0.01319 | 0.3708±0.08923 | 0.4906±0.06185 | 0.9401±0.19202 | 0.7680±0.07649 | 0.5453±0.04814 | 0.5631±0.02233 |
| | 1440-1440 | 0.9053±0.27275 | 0.7328±0.11057 | 0.1817±0.00859 | 0.2764±0.00909 | 0.4475±0.10894 | 0.5329±0.06986 | 0.7860±0.01219 | 0.6704±0.00613 | 0.5371±0.06814 | 0.5559±0.03300 |
| | 1440-2880 | 1.1282±0.05767 | 0.8087±0.02547 | 0.2029±0.01118 | 0.3100±0.01305 | 0.9617±0.30428 | 0.8271±0.12771 | 2.0561±0.15722 | 1.1595±0.04848 | 0.9061±0.14939 | 0.7220±0.05635 |
| **Dlinear** | 168-168 | 0.2605±0.00138 | 0.3579±0.00117 | 0.1519±0.00017 | 0.2195±0.00023 | 0.1122±0.00081 | 0.2605±0.00106 | 0.2556±0.00080 | 0.3944±0.00091 | 0.2421±0.00302 | 0.3578±0.00240 |
| | 168-336 | 0.3080±0.00167 | 0.3946±0.00149 | 0.1468±0.00012 | 0.2210±0.00017 | 0.1251±0.00086 | 0.2794±0.00101 | 0.2891±0.00143 | 0.4256±0.00082 | 0.2918±0.00149 | 0.3975±0.00112 |
| | 336-336 | 0.2740±0.00539 | 0.3720±0.00491 | 0.1235±0.00035 | 0.2114±0.00053 | 0.1261±0.00103 | 0.2803±0.00130 | 0.2950±0.00116 | 0.4329±0.00099 | 0.2905±0.00198 | 0.3969±0.00170 |
| | 336-720 | 0.3208±0.00328 | 0.4188±0.00280 | 0.1449±0.00126 | 0.2252±0.00135 | 0.1942±0.00074 | 0.3462±0.00067 | 0.4125±0.00304 | 0.5136±0.00225 | 0.3897±0.00344 | 0.4739±0.00166 |
| | 720-720 | 0.3203±0.00199 | 0.4202±0.00150 | 0.1410±0.00065 | 0.2241±0.00069 | 0.1920±0.00181 | 0.3435±0.00205 | 0.3495±0.01694 | 0.4749±0.01135 | 0.3724±0.00649 | 0.4614±0.00467 |
| | 720-1440 | 0.4923±0.01121 | 0.5473±0.00517 | 0.1639±0.00075 | 0.2412±0.00083 | 0.2952±0.00412 | 0.4370±0.00330 | 0.5037±0.03426 | 0.5645±0.01609 | 0.4406±0.00678 | 0.5264±0.00268 |
| | 1440-1440 | 0.5146±0.01460 | 0.5615±0.00841 | 0.1599±0.00409 | 0.2411±0.00560 | 0.2200±0.00889 | 0.3714±0.00775 | 0.5176±0.02561 | 0.5734±0.01383 | 0.3147±0.01282 | 0.4417±0.00889 |
| | 1440-2880 | 0.8355±0.01493 | 0.7193±0.00730 | 0.1550±0.00478 | 0.2472±0.00679 | 0.3773±0.02685 | 0.4794±0.01928 | 0.5053±0.04308 | 0.5584±0.02510 | 0.3197±0.00551 | 0.4533±0.00462 |
| **FiLM** | 168-168 | 0.2587±0.00032 | 0.3557±0.00029 | 0.1501±0.00067 | 0.2143±0.00139 | 0.1091±0.00033 | 0.2558±0.00035 | 0.2546±0.00141 | 0.3942±0.00161 | 0.2426±0.00037 | 0.3544±0.00036 |
| | 168-336 | 0.3062±0.00030 | 0.3922±0.00014 | 0.1453±0.00033 | 0.2165±0.00071 | 0.1187±0.00058 | 0.2708±0.00068 | 0.2894±0.00718 | 0.4263±0.00566 | 0.2981±0.00036 | 0.3988±0.00033 |
| | 336-336 | 0.2722±0.00027 | 0.3659±0.00034 | 0.1324±0.00046 | 0.2104±0.00641 | 0.1196±0.00045 | 0.2738±0.00069 | 0.2951±0.00226 | 0.4347±0.00188 | 0.2943±0.00107 | 0.3969±0.00112 |
| | 336-720 | 0.3171±0.00083 | 0.4152±0.00072 | 0.1438±0.00028 | 0.2229±0.00039 | 0.1793±0.00112 | 0.3335±0.00091 | 0.4158±0.01033 | 0.5162±0.00636 | 0.4096±0.00062 | 0.4767±0.00069 |
| | 720-720 | 0.3158±0.00086 | 0.4154±0.00061 | 0.1383±0.00039 | 0.2208±0.00053 | 0.1845±0.00103 | 0.3379±0.00086 | 0.4045±0.00302 | 0.5105±0.00184 | 0.3999±0.00326 | 0.4661±0.00329 |
| | 720-1440 | 0.4730±0.00086 | 0.5336±0.00046 | 0.1638±0.00089 | 0.2448±0.00100 | 0.2949±0.00327 | 0.4388±0.00289 | 0.7166±0.00912 | 0.6628±0.00477 | 0.6360±0.00325 | 0.5997±0.00232 |
| | 1440-1440 | 0.4849±0.00169 | 0.5429±0.00048 | 0.1602±0.00327 | 0.2437±0.00456 | 0.2294±0.01020 | 0.3759±0.00827 | 0.7446±0.01672 | 0.6590±0.00733 | 0.6002±0.00341 | 0.5880±0.00276 |
| | 1440-2880 | 0.6847±0.00543 | 0.6493±0.00310 | 0.1744±0.00361 | 0.2693±0.00556 | 0.6834±0.01115 | 0.7096±0.00431 | 2.3835±0.10801 | 1.6030±0.02952 | 1.2605±0.00135 | 0.8805±0.00301 |
| **FEDformer** | 168-168 | 0.3028±0.01797 | 0.4020±0.01216 | 0.2469±0.02237 | 0.3479±0.02483 | 0.1284±0.00272 | 0.2826±0.00467 | 0.2844±0.01173 | 0.4285±0.01111 | 0.2583±0.00281 | 0.3774±0.00393 |
| | 168-336 | 0.3522±0.01174 | 0.4394±0.00829 | 0.2426±0.00987 | 0.3449±0.01478 | 0.1271±0.00393 | 0.2810±0.00486 | 0.2961±0.00488 | 0.4355±0.00418 | 0.2909±0.00348 | 0.4030±0.00359 |
| | 336-336 | 0.3378±0.00695 | 0.4303±0.00400 | 0.2339±0.01067 | 0.3365±0.01400 | 0.1252±0.00689 | 0.2794±0.00863 | 0.2884±0.01345 | 0.4314±0.01098 | 0.2791±0.00466 | 0.3984±0.00539 |
| | 336-720 | 0.3813±0.00988 | 0.4634±0.00748 | 0.2987±0.03172 | 0.3976±0.02903 | 0.1534±0.01574 | 0.3178±0.01628 | 0.3425±0.02883 | 0.4656±0.02074 | 0.2648±0.00820 | 0.3915±0.01046 |
| | 720-720 | 0.4023±0.03710 | 0.4769±0.02158 | 0.2667±0.02331 | 0.3685±0.02884 | 0.1386±0.00869 | 0.2995±0.01203 | 0.3275±0.01948 | 0.4534±0.01363 | 0.2416±0.00381 | 0.3728±0.00513 |
| | 720-1440 | 0.4833±0.02229 | 0.5393±0.01414 | 0.2753±0.00622 | 0.3650±0.00502 | 0.1768±0.03601 | 0.3409±0.03216 | 0.3731±0.04086 | 0.4827±0.02391 | 0.2352±0.03670 | 0.3733±0.03472 |
| | 1440-1440 | 0.5142±0.03440 | 0.5571±0.01701 | 0.2848±0.02063 | 0.3681±0.01120 | 0.3574±0.14083 | 0.4878±0.11818 | 0.3906±0.04614 | 0.4951±0.02924 | 0.2226±0.01285 | 0.3609±0.01098 |
| | 1440-2880 | 3.9018±4.00564 | 1.5276±0.92236 | 0.2952±0.01400 | 0.3844±0.01182 | 0.4269±0.29204 | 0.5252±0.20346 | 1.7167±1.76025 | 0.9698±0.60094 | 0.2138±0.00950 | 0.3599±0.01022 |
| **Pyraformer** | 168-168 | 0.2651±0.01387 | 0.3802±0.01069 | 0.2979±0.01338 | 0.3815±0.01238 | 0.1534±0.01808 | 0.3287±0.01959 | 0.2746±0.01024 | 0.4080±0.00685 | 0.2144±0.02003 | 0.3451±0.02343 |
| | 168-336 | 0.5392±0.26585 | 0.5271±0.10622 | 0.5838±0.02841 | 0.5652±0.01840 | 0.1665±0.00781 | 0.3419±0.01014 | 0.2392±0.02900 | 0.3834±0.02057 | 0.2594±0.04642 | 0.3833±0.04056 |
| | 336-336 | 0.2994±0.01279 | 0.4030±0.01117 | 0.4703±0.06731 | 0.4964±0.05265 | 0.1408±0.01032 | 0.3087±0.01180 | 0.2610±0.03612 | 0.4010±0.02955 | 0.2310±0.02787 | 0.3591±0.02687 |
| | 336-720 | 0.4856±0.06663 | 0.5243±0.03642 | 0.5235±0.10386 | 0.5292±0.08052 | 0.3984±0.23533 | 0.5202±0.16798 | 0.2341±0.00798 | 0.3818±0.00520 | 0.3241±0.06639 | 0.4300±0.04847 |
| | 720-720 | 0.3115±0.02065 | 0.4218±0.01610 | 0.4831±0.05721 | 0.4962±0.04154 | 0.1563±0.01867 | 0.3253±0.01669 | 0.2795±0.04548 | 0.4151±0.03349 | 0.2378±0.00668 | 0.3684±0.05805 |
| | 720-1440 | 0.3250±0.01497 | 0.4332±0.01241 | 0.4463±0.04505 | 0.4609±0.03322 | 0.1666±0.03171 | 0.3315±0.02987 | 0.2952±0.0432 | 0.4336±0.02947 | 0.6810±0.30460 | 0.6352±0.18544 |
| | 1440-1440 | 0.4895±0.04971 | 0.5280±0.03132 | 0.4710±0.01399 | 0.4794±0.00751 | 0.3487±0.10751 | 0.4866±0.08011 | 0.2946±0.02847 | 0.4316±0.01817 | 0.2401±0.04767 | 0.3777±0.04109 |
| | 1440-2880 | 0.4320±0.01920 | 0.5161±0.01283 | 0.5165±0.02708 | 0.5305±0.01786 | 0.5857±0.12854 | 0.6760±0.09034 | 0.3345±0.01392 | 0.4544±0.01035 | 0.1852±0.00663 | 0.3332±0.00954 |
| **Autoformer** | 168-168 | 0.3496±0.01079 | 0.4337±0.01014 | 0.2378±0.01295 | 0.3490±0.01273 | 0.1438±0.00332 | 0.2872±0.00515 | 0.2903±0.01461 | 0.4326±0.01218 | 0.2670±0.00243 | 0.3813±0.00317 |
| | 168-336 | 0.4733±0.05218 | 0.5120±0.03357 | 0.2683±0.01140 | 0.3803±0.01102 | 0.1315±0.00670 | 0.2878±0.00682 | 0.4447±0.30613 | 0.4964±0.12815 | 0.2990±0.00389 | 0.4096±0.00475 |
| | 336-336 | 0.5153±0.12322 | 0.5304±0.07432 | 0.2460±0.01621 | 0.3567±0.02022 | 0.1384±0.01765 | 0.2959±0.01631 | 0.2805±0.01655 | 0.4255±0.01391 | 0.3066±0.01615 | 0.4162±0.01489 |
| | 336-720 | 0.5045±0.02785 | 0.5393±0.01945 | 0.2849±0.03404 | 0.3956±0.03523 | 0.1928±0.03373 | 0.3450±0.03092 | 0.3372±0.01763 | 0.4625±0.01084 | 0.3468±0.01639 | 0.4592±0.01022 |
| | 720-720 | 0.9639±0.26411 | 0.7520±0.10858 | 0.2959±0.02180 | 0.4045±0.01861 | 0.2388±0.04468 | 0.3869±0.03952 | 0.4668±0.10896 | 0.5477±0.06613 | 0.4309±0.04345 | 0.5085±0.02459 |
| | 720-1440 | 1.4957±0.49225 | 0.9533±0.17135 | 0.3104±0.03063 | 0.4095±0.03276 | 0.3298±0.03720 | 0.4741±0.02983 | 0.5633±0.09564 | 0.5996±0.04671 | 0.8599±0.43154 | 0.7064±0.19138 |
| | 1440-1440 | 1.7873±1.12827 | 1.0283±0.36295 | 0.2970±0.02378 | 0.3999±0.02575 | 0.4531±0.12054 | 0.5507±0.09597 | 0.8029±0.34886 | 0.7140±0.15332 | 0.9766±0.38596 | 0.7739±0.10604 |
| | 1440-2880 | 1.2867±0.32261 | 0.8878±0.12519 | 0.3035±0.02038 | 0.3982±0.02444 | 1.3566±1.10518 | 0.9235±0.4592 | 4.1031±2.27321 | 1.7198±0.61623 | 1.7465±0.51402 | 1.0962±0.18964 |
| **Informer** | 168-168 | 0.3779±0.02565 | 0.4594±0.01559 | 0.3363±0.03784 | 0.3994±0.02525 | 0.1563±0.01273 | 0.3299±0.01299 | 0.3764±0.01781 | 0.4863±0.00921 | 0.2639±0.02557 | 0.3926±0.02566 |
| | 168-336 | 0.5037±0.08433 | 0.5301±0.04791 | 0.5891±0.11257 | 0.5608±0.05492 | 0.1663±0.03122 | 0.3335±0.03102 | 0.3364±0.02216 | 0.4583±0.01619 | 0.2798±0.02688 | 0.4061±0.02242 |
| | 336-336 | 0.4591±0.03057 | 0.4991±0.01943 | 0.5447±0.03647 | 0.5384±0.01976 | 0.1648±0.04980 | 0.3291±0.04794 | 0.3709±0.02624 | 0.4785±0.01520 | 0.2898±0.01358 | 0.4129±0.01501 |
| | 336-720 | 0.6545±0.12350 | 0.5975±0.05506 | 1.2044±0.28757 | 0.8254±0.10672 | 0.1522±0.01115 | 0.3203±0.01333 | 0.3572±0.01765 | 0.4675±0.00646 | 0.2483±0.00848 | 0.3778±0.00462 |
| | 720-720 | 0.4850±0.01891 | 0.5238±0.00888 | 1.2954±0.57549 | 0.9205±0.19654 | 0.1595±0.01501 | 0.3259±0.01221 | 0.3585±0.01299 | 0.4699±0.00700 | 0.3545±0.13325 | 0.4569±0.08858 |
| | 720-1440 | 0.5064±0.04967 | 0.5317±0.02899 | 0.7614±0.17528 | 0.6469±0.09409 | 0.1378±0.00853 | 0.3051±0.00856 | 0.4025±0.04622 | 0.4991±0.02220 | 0.2466±0.00877 | 0.3849±0.01187 |
| | 1440-1440 | 0.7247±0.23436 | 0.6292±0.10907 | 0.7375±0.04517 | 0.6414±0.02969 | 0.1430±0.00927 | 0.3156±0.00902 | 0.3484±0.01264 | 0.4786±0.00480 | 0.2556±0.00559 | 0.3969±0.00630 |
| | 1440-2880 | 0.6152±0.25666 | 0.5953±0.13194 | 0.9849±0.23417 | 0.7618±0.10144 | 0.3177±0.10572 | 0.4733±0.08814 | 0.3335±0.00947 | 0.4482±0.00366 | 0.2126±0.01052 | 0.3600±0.01317 |
| **Transformer** | 168-168 | 0.3036±0.01240 | 0.4068±0.01224 | 1.5204±0.76091 | 0.9594±0.35768 | 0.1504±0.01406 | 0.3257±0.01774 | 0.3043±0.02208 | 0.4365±0.01397 | 0.2200±0.00977 | 0.3438±0.00915 |
| | 168-336 | 0.3583±0.01874 | 0.4435±0.01628 | 0.6953±0.28867 | 0.6015±0.15680 | 0.1599±0.01629 | 0.3324±0.01735 | 0.3662±0.02673 | 0.4671±0.01450 | 0.2230±0.00914 | 0.3488±0.01073 |
| | 336-336 | 0.5771±0.27509 | 0.5643±0.15434 | 0.8482±0.53327 | 0.6424±0.21839 | 0.1438±0.00533 | 0.3121±0.00606 | 0.3218±0.03005 | 0.4412±0.02016 | 0.2308±0.00771 | 0.3556±0.01044 |
| | 336-720 | 0.4368±0.05110 | 0.4920±0.03146 | 0.7320±0.22311 | 0.6233±0.14229 | 0.1644±0.02587 | 0.3304±0.02911 | 0.3582±0.14008 | 0.4629±0.09578 | 0.2334±0.01492 | 0.3570±0.01743 |
| | 720-720 | 0.3992±0.04794 | 0.4640±0.02729 | 1.1963±0.74703 | 0.8271±0.35161 | 0.1730±0.00988 | 0.3414±0.01075 | 0.3087±0.02578 | 0.4320±0.01658 | 0.2463±0.00604 | 0.3722±0.01029 |
| | 720-1440 | 0.4030±0.03837 | 0.4797±0.02257 | 0.9876±0.19880 | 0.7445±0.10064 | 0.1905±0.03701 | 0.3555±0.03645 | 0.3712±0.01640 | 0.4805±0.00732 | 0.2188±0.01761 | 0.3512±0.01479 |
| | 1440-1440 | 0.5531±0.05384 | 0.5524±0.02729 | 0.7430±0.05136 | 0.6492±0.02996 | 0.1972±0.02466 | 0.3630±0.02502 | 0.3797±0.02330 | 0.4818±0.01429 | 0.2610±0.01388 | 0.3823±0.01305 |
| | 1440-2880 | 0.5243±0.06259 | 0.5460±0.03610 | 0.6000±0.07314 | 0.5877±0.05861 | 0.3495±0.08217 | 0.4911±0.06523 | 0.3737±0.05141 | 0.4787±0.03206 | 0.1993±0.00884 | 0.3436±0.00934 |

sets, when their variances are not significantly different, it indicates that the data fluctuations in the two sets are similar. In this case, the difference in the data distribution is not significant, and selecting $norm$ as 0 is reasonable. However, when the variance difference between the training and validation sets is relatively large (approximately twice or half), there is a significant difference in the data distribution between the two sets, and $norm$ should be 1 to re-normalize the input of the model. It should be noted that in the Weather dataset, there are negative values present, which makes its mean close to 0, resulting in a large difference between its variance and mean. However, this is reasonable. For the Traffic dataset, there are no negative values present, so even if the variances of its training and validation sets are similar, its significant fluctuations can be observed when combined with its mean. This was also demonstrated by setting the $norm$ value to 1 during the model training process.

Table 10: The distribution of data in the training and validation sets (Mean and STD) and the value of $norm$.

| Datasets | ECL | | | Traffic | | | ETTh1 | | | ETTh2 | | | Weather | | |
|---|---|---|---|---|---|---|---|---|---|---|---|---|---|---|---|
| Tasks | training set | validation set | norm | training set | validation set | norm | training set | validation set | norm | training set | validation set | norm | training set | validation set | norm |
| 168-168 | 3425.733±564.8776 | 3036.397±388.2128 | | 0.029±0.0170 | 0.034±0.0201 | | 16.880±8.2921 | 6.667±4.1794 | | 28.959±12.0653 | 18.680±9.0427 | | 0.500±6.6321 | 1.143±7.7659 | |
| 168-336 | 3427.480±566.9556 | 3036.291±388.0110 | | 0.029±0.0170 | 0.035±0.0202 | | 16.606±8.0735 | 6.258±3.8462 | | 28.767±12.0604 | 17.922±8.5800 | | 0.536±6.6444 | 1.017±7.8162 | |
| 336-336 | 3428.455±569.1108 | 3036.291±388.0110 | 0 | 0.029±0.0170 | 0.035±0.0202 | 1 | 16.207±7.5364 | 6.258±3.8462 | 1 | 28.434±11.8740 | 17.922±8.5800 | 0 | 0.585±6.6409 | 1.017±7.8162 | 0 |
| 336-720 | 3434.150±573.2660 | 3037.919±387.2758 | | 0.029±0.0170 | 0.035±0.0203 | | 15.446±6.6217 | 5.583±3.4658 | | 27.774±11.6120 | 16.360±7.7594 | | 0.700±6.6410 | 0.721±7.9234 | |
| 720-720 | 3437.773±578.4705 | 3037.919±387.2758 | | 0.029±0.0170 | 0.035±0.0203 | | 14.832±5.9927 | 5.583±3.4658 | | 27.111±11.3299 | 16.360±7.7594 | | 0.825±6.6270 | 0.721±7.9234 | |
| 720-1440 | 3439.817±586.5029 | 3046.877±397.7761 | | 0.029±0.0170 | 0.035±0.0204 | | 14.044±5.5077 | 4.273±2.7600 | | 26.403±11.4257 | 13.646±6.5725 | | 0.978±6.6791 | -0.547±7.5174 | |
| 1440-1440 | 3452.135±594.6857 | 3046.877±397.7761 | 0 | 0.029±0.0169 | 0.035±0.0204 | 1 | 13.722±5.5456 | 4.273±2.7600 | 1 | 26.355±11.8918 | 13.646±6.5725 | 1 | 1.029±6.7679 | -0.547±7.5174 | 0 |
| 1440-2880 | 3458.328±610.2118 | 3093.128±446.4128 | | 0.029±0.0171 | 0.036±0.0208 | | 14.195±5.5780 | 2.623±2.5005 | | 28.303±12.1275 | 9.130±5.5363 | | 0.725±6.8812 | -3.859±5.5129 | |

**The impact of** $C$  The choice of $C$ for the model represents the determination of the sequence period. Due to the different original acquisition granularities of each dataset, in order to ensure that they contain the same semantic information on the same task, in this paper, our experiments are conducted by aggregating them into one hour. To facilitate the discovery of long-term repetitive patterns in the sequence, we set the $C$ to 12, 24, 48, and the specific experimental results are shown in Table 11. From Table 11, we can clearly see that for sequences with time steps in hours, using a period of 24 hours (1 day) for division yields better results. This is because in WIT, information is transmitted horizontally at the granularity of hours, while information is transmitted vertically at the granularity of days. This approach can more fully capture the long- and short-term repetitive patterns and global/local correlations hidden in the time series data.

Table 11: Parameter Sensitivity of $C$ on ECL and Traffic datasets.

| Settings for $C$ | | 12 | | 24 | | 48 | |
|---|---|---|---|---|---|---|---|
| Datasets | Tasks | MSE | MAE | MSE | MAE | MSE | MAE |
| ECL | 168-168 | 0.2461 | 0.3648 | 0.2397 | 0.3519 | - | - |
| | 168-336 | 0.3166 | 0.4230 | 0.2607 | 0.3721 | - | - |
| | 336-336 | 0.3217 | 0.4099 | 0.2517 | 0.3627 | 0.2634 | 0.3847 |
| | 336-720 | 0.3397 | 0.4310 | 0.3084 | 0.4055 | 0.3233 | 0.4130 |
| | 720-720 | 0.2802 | 0.3951 | 0.2478 | 0.3651 | 0.3175 | 0.4063 |
| | 720-1440 | 0.2907 | 0.4051 | 0.2499 | 0.3727 | 0.2985 | 0.3946 |
| | 1440-1440 | 0.3007 | 0.4110 | 0.2408 | 0.3680 | 0.2748 | 0.3768 |
| | 1440-2880 | 0.4301 | 0.5150 | 0.3359 | 0.4383 | 0.3464 | 0.4347 |
| Traffic | 168-168 | 0.1598 | 0.2408 | 0.1377 | 0.2051 | - | - |
| | 168-336 | 0.2272 | 0.2783 | 0.1321 | 0.2059 | - | - |
| | 336-336 | 0.2131 | 0.2967 | 0.1306 | 0.2054 | 0.1412 | 0.2278 |
| | 336-720 | 0.2022 | 0.2804 | 0.1391 | 0.2175 | 0.1510 | 0.2371 |
| | 720-720 | 0.2028 | 0.2846 | 0.1408 | 0.2191 | 0.1505 | 0.2326 |
| | 720-1440 | 0.2710 | 0.3311 | 0.1672 | 0.2449 | 0.1813 | 0.2605 |
| | 1440-1440 | 0.2676 | 0.3301 | 0.1543 | 0.2325 | 0.1881 | 0.2652 |
| | 1440-2880 | 0.2407 | 0.3443 | 0.1425 | 0.2333 | 0.1692 | 0.2765 |

## I.2  A detailed explanation of time and memory consumption

The efficiency of the model is crucial, as even if the model's efficiency is high, it is limited if it does not have good predictive performance. Therefore, in this paper, we only compared our proposed method with methods that have good or the latest results. It should be noted that, for example, Transformer [Vaswani et al., 2017], Informer [Zhou et al., 2021] and Autoformer [Wu et al., 2021] have been shown in previous works to have higher complexity than FiLM [Zhou et al., 2022b], so we only compared our proposed method with FiLM, rather than comparing with Transformer, Informer and Autoformer again.

**Memory Usage**  As shown in Figure 5(a) and (c), WITRAN has good memory usage with the prolonging the input length and output length. For a fair comparison, we fix the experimental settings of all methods, where we fix the input length as 720 and prolong the output length. Moreover, we fix the output length as 720 and prolong the input length. And for each model, to achieve good performance, it is important to have a sufficiently large search space to select the most suitable parameters for the model, in order to achieve the best possible results. Therefore, we set all public parameters to their upper limits to test the memory usage of each model. To allow as many models as possible to be trained, we uniformly set the batch size to 8 for testing in this section. It should be noted that if a point is not shown on the graph, it indicates that the model encountered an "out of memory" (OOM) error at that particular configuration of input sequence length and output length.

As for TimesNet, due to its large memory usage, it cannot be trained even with a batch size of 1. Therefore, we did not include TimesNet in the Figure 5. From Figure 5(a), we can observe that when the input sequence is fixed, the memory usage of WITRAN and WIT is not significantly different since they both have the same prediction module. Additionally, we found that as the output length increases, the memory usage of WITRAN and WIT also increases linearly, but our proposed method still has the lowest memory usage. From Figure 5(c), we can observe that the memory usage of WITRAN increases very slowly with the increase in sequence length, and the memory usage of WITRAN is even lower than that of WIT. This is because WITRAN does not need to retain a large number of intermediate storage variables in the computation of each slice, while WIT needs to store the intermediate variables for the entire sequence. Through comparison with other methods, we can clearly see the advantages of WITRAN on memory consumption.

**Training Speed**    As shown in Figure 5(b) and (d), WITRAN has a faster training speed than others with the prolonging the input length and output length. To ensure a fair comparison of the performance of each model, we fixed the experimental settings, which were the same as the settings used for testing memory usage. The experiment is performed on ECL dataset. Due to our extremely low space complexity, we followed the experimental settings used in FiLM [Zhou et al., 2022b] for comparing the training Speed. To fully utilize the GPU memory, our model can increase the batch size from 8 to 32 under the experimental settings mentioned above. Similarly, FiLM can also increase the batch size = 32. However, we still have the fastest training speed, and due to the role of RAN mainly being on the input sequence, our advantage can be clearly seen from Figure 5(d), which is related to the square root of the sequence length. Through comparison with other methods, we can clearly see the advantages of WITRAN on time consumption.

# J   Showcases of Main Results

As shown in Figure 10 to Figure 49, we plot the forecasting results from the test set of all datasets for comparison. For the long-range forecasting task, we chose two suboptimal models, FiLM and Pyraformer. As for the ultra-long-range forecasting task, we also chose two suboptimal models, Pyraformer and TimesNet. Our model WITRAN gives the best performance among different models. Moreover, WITRAN is significantly better at modeling global and local correlations, as well as discovering long- and short-term repetitive patterns in the time series than other models.

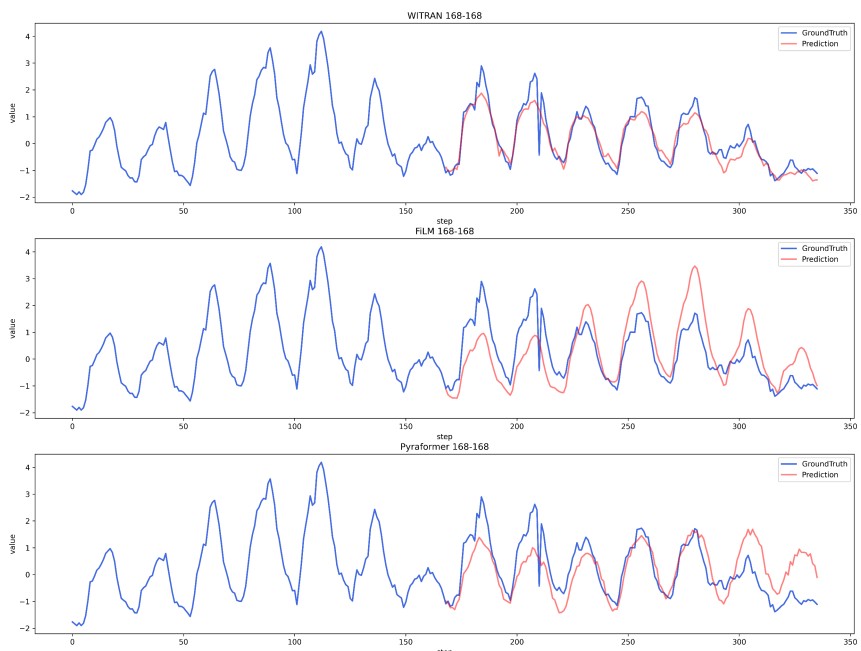

Figure 10: Forecasting cases comparison of WITRAN, FiLM and Pyraformer on the 168-168 task of the ECL dataset.

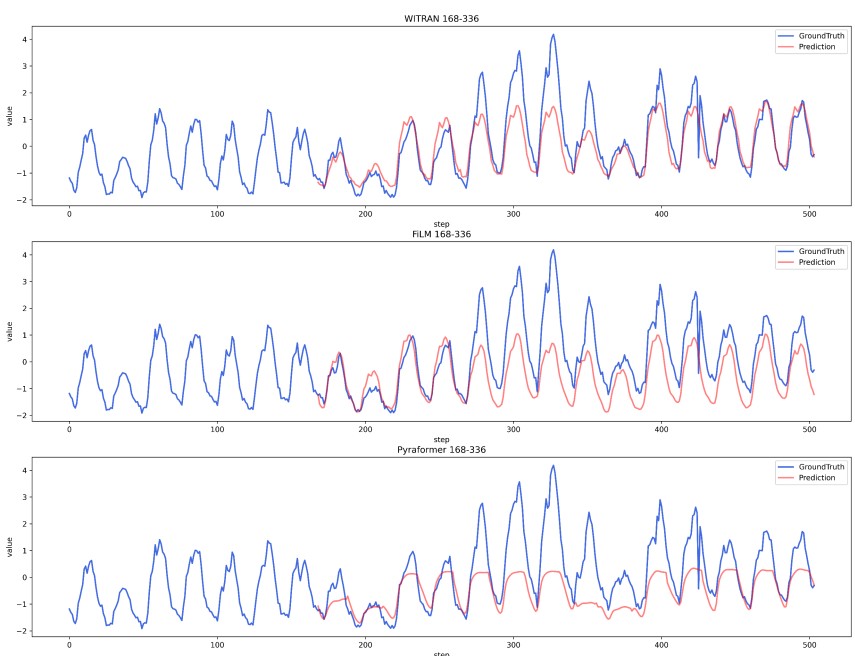

Figure 11: Forecasting cases comparison of WITRAN, FiLM and Pyraformer on the 168-336 task of the ECL dataset.

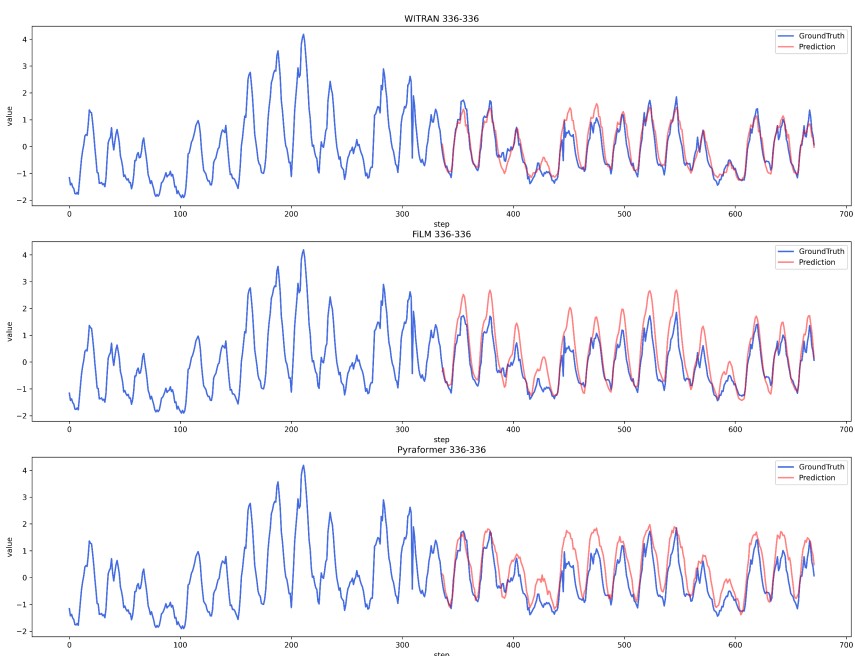

Figure 12: Forecasting cases comparison of WITRAN, FiLM and Pyraformer on the 336-336 task of the ECL dataset.

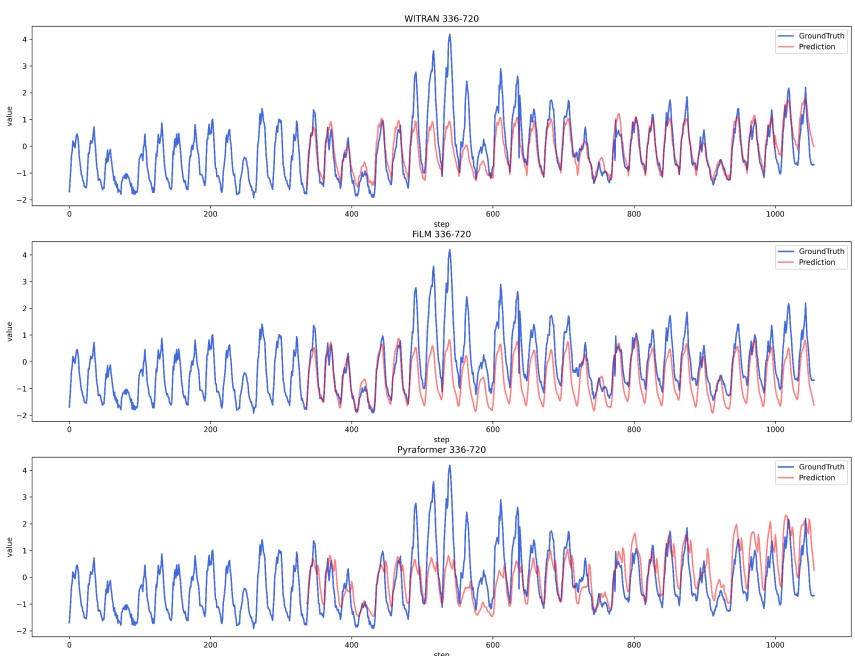

Figure 13: Forecasting cases comparison of WITRAN, FiLM and Pyraformer on the 336-720 task of the ECL dataset.

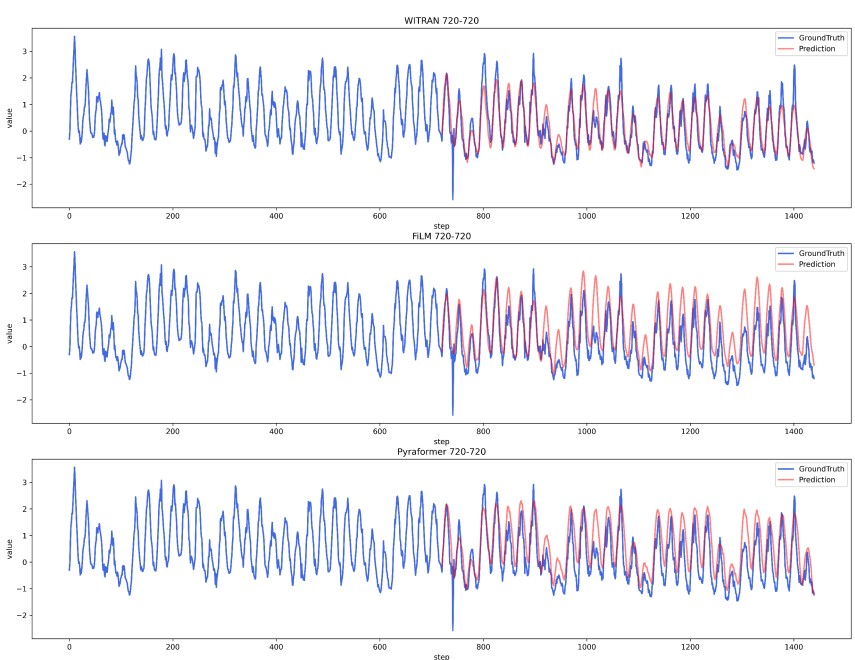

Figure 14: Forecasting cases comparison of WITRAN, FiLM and Pyraformer on the 720-720 task of the ECL dataset.

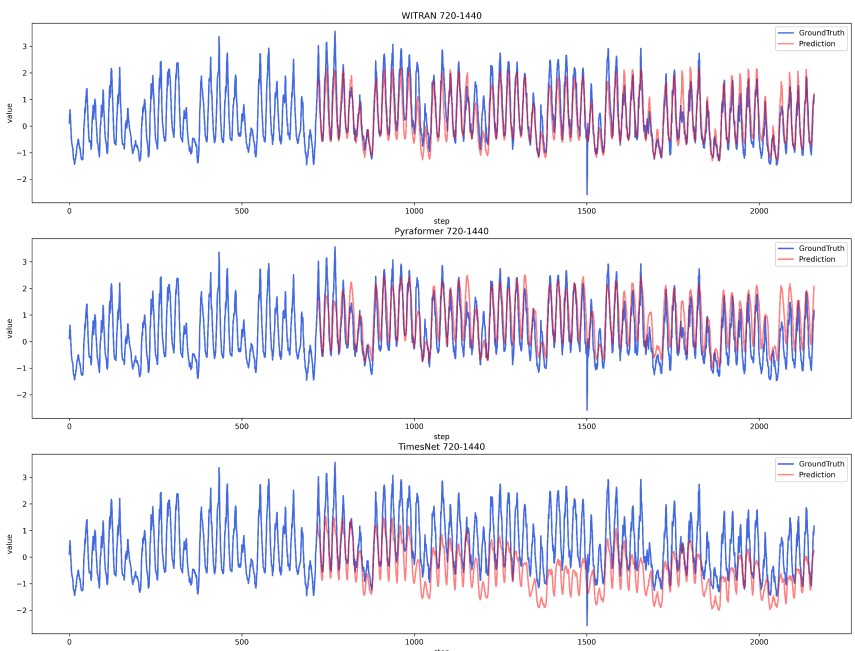

Figure 15: Forecasting cases comparison of WITRAN, Pyraformer and TimesNet on the 720-1440 task of the ECL dataset.

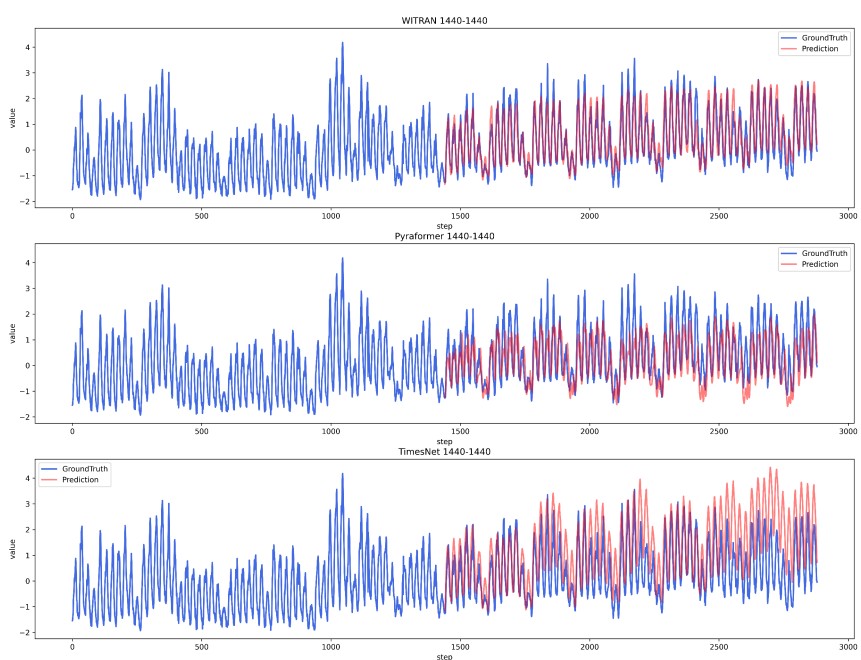

Figure 16: Forecasting cases comparison of WITRAN, Pyraformer and TimesNet on the 1440-1440 task of the ECL dataset.

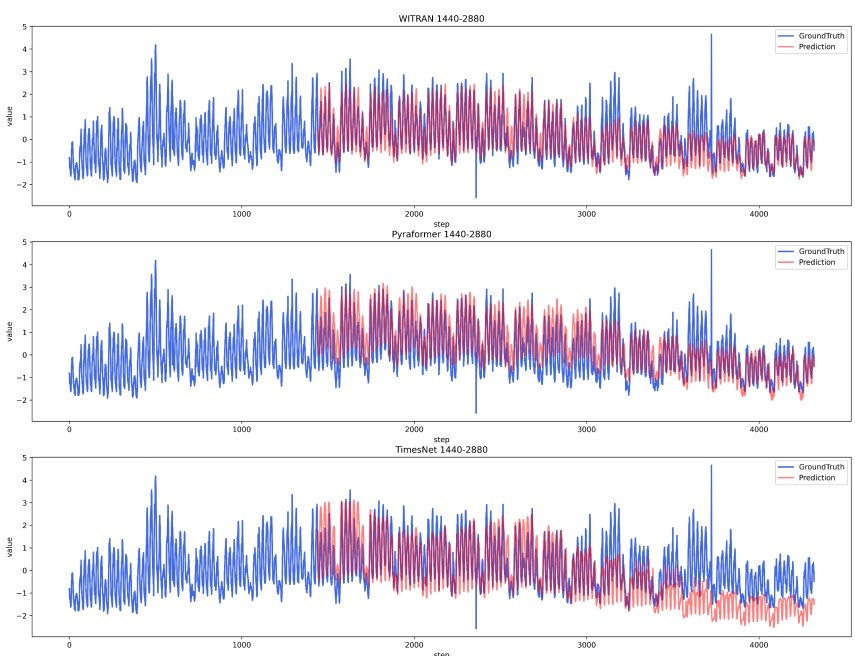

Figure 17: Forecasting cases comparison of WITRAN, Pyraformer and TimesNet on the 1440-2880 task of the ECL dataset.

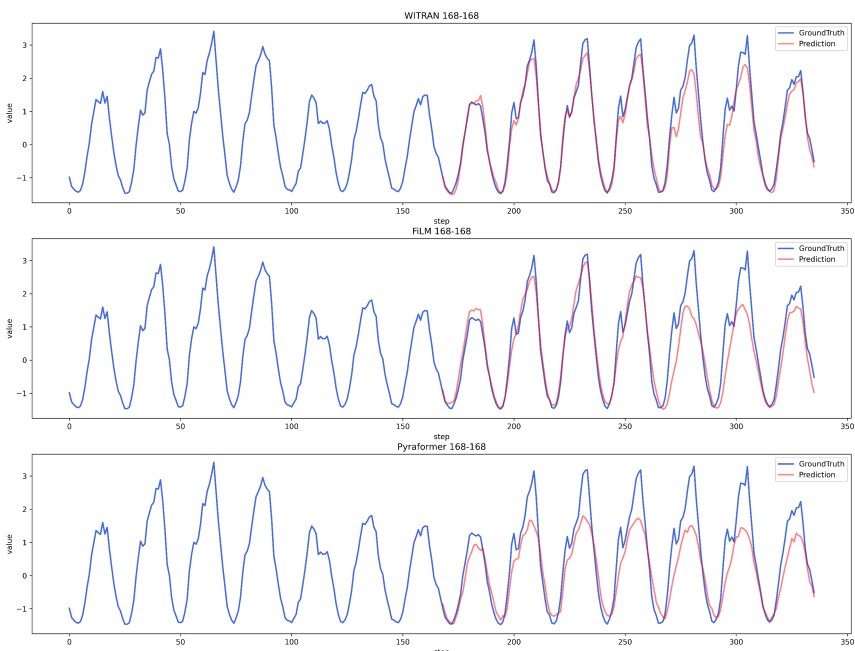

Figure 18: Forecasting cases comparison of WITRAN, FiLM and Pyraformer on the 168-168 task of the Traffic dataset.

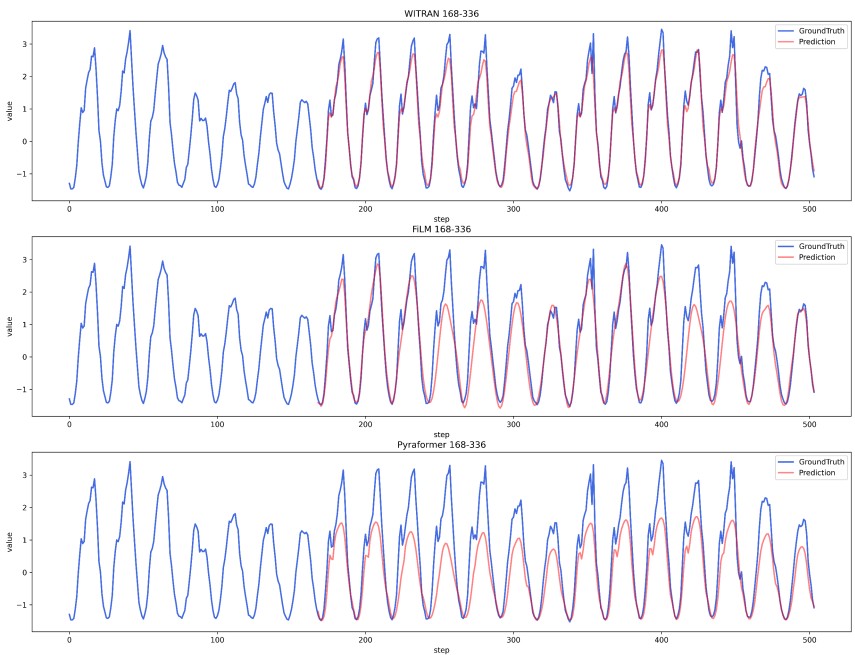

Figure 19: Forecasting cases comparison of WITRAN, FiLM and Pyraformer on the 168-336 task of the Traffic dataset.

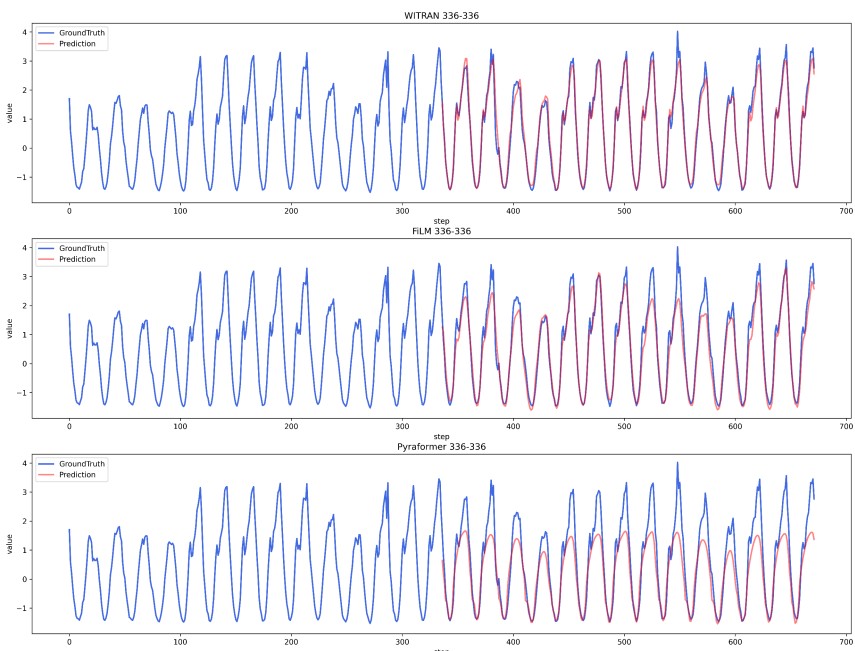

Figure 20: Forecasting cases comparison of WITRAN, FiLM and Pyraformer on the 336-336 task of the Traffic dataset.

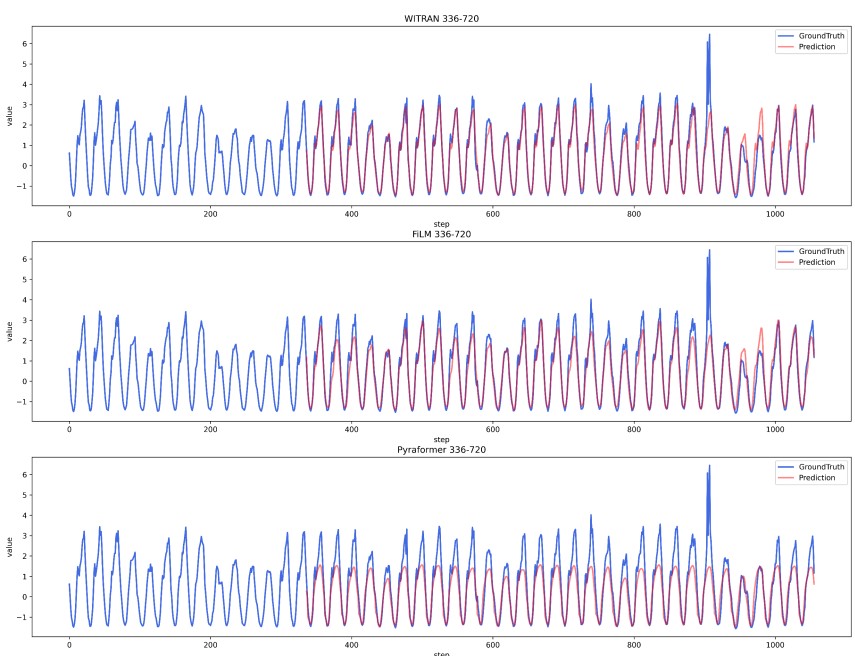

Figure 21: Forecasting cases comparison of WITRAN, FiLM and Pyraformer on the 336-720 task of the Traffic dataset.

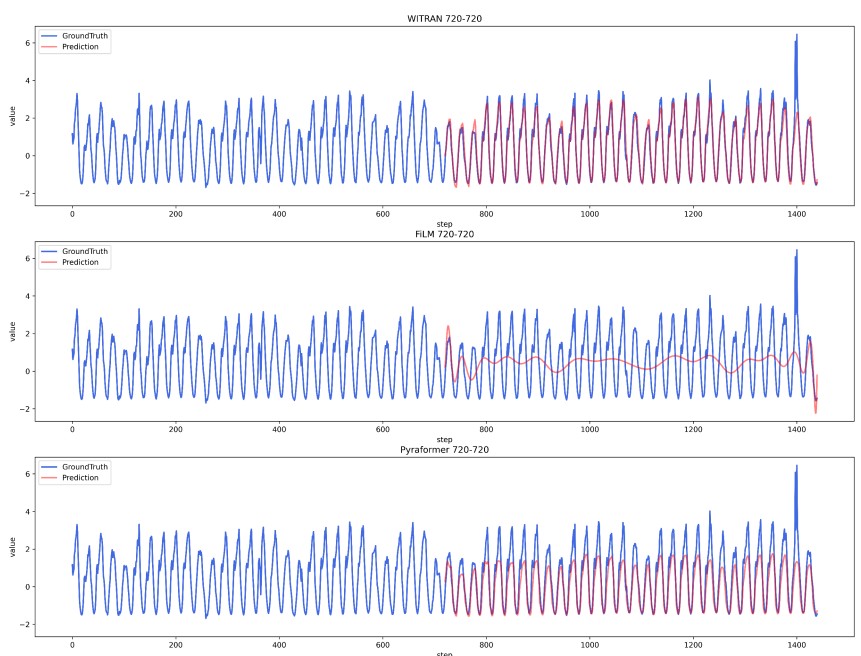

Figure 22: Forecasting cases comparison of WITRAN, FiLM and Pyraformer on the 720-720 task of the Traffic dataset.

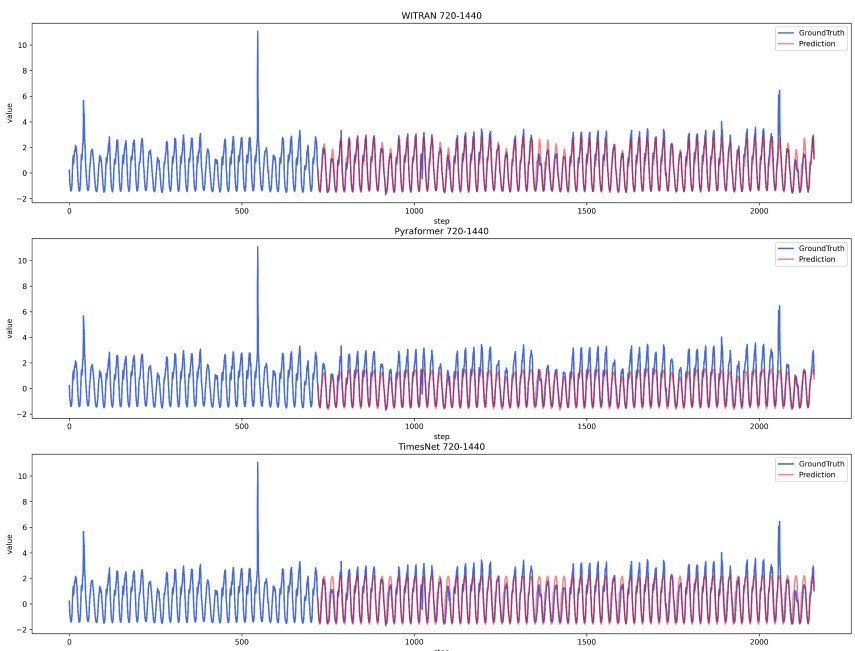

Figure 23: Forecasting cases comparison of WITRAN, Pyraformer and TimesNet on the 720-1440 task of the Traffic dataset.

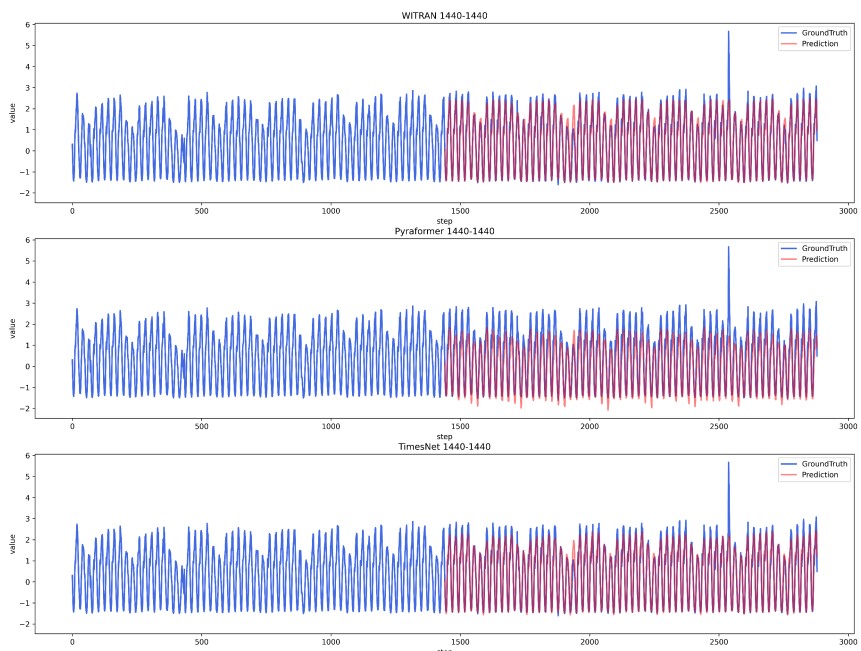

Figure 24: Forecasting cases comparison of WITRAN, Pyraformer and TimesNet on the 1440-1440 task of the Traffic dataset.

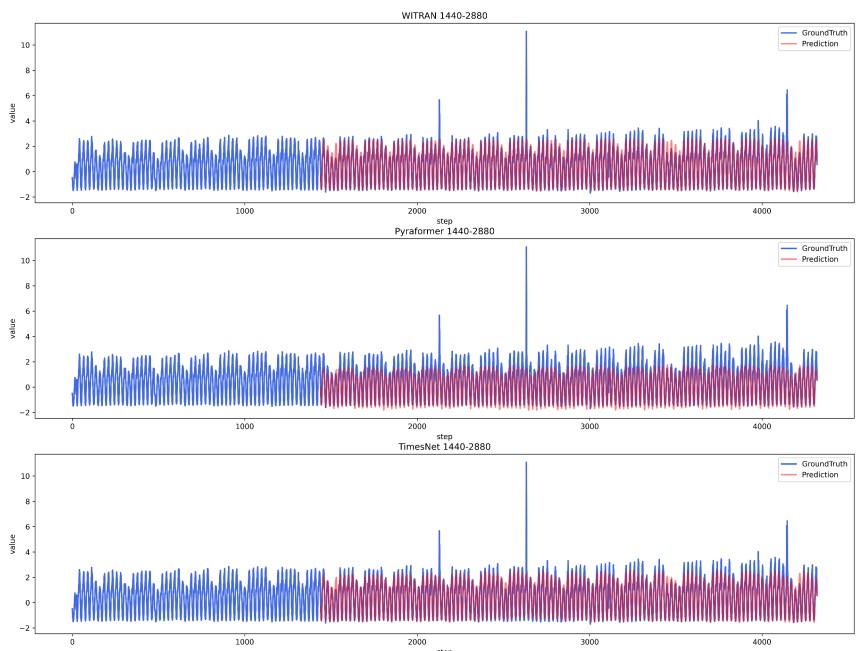

Figure 25: Forecasting cases comparison of WITRAN, Pyraformer and TimesNet on the 1440-2880 task of the Traffic dataset.

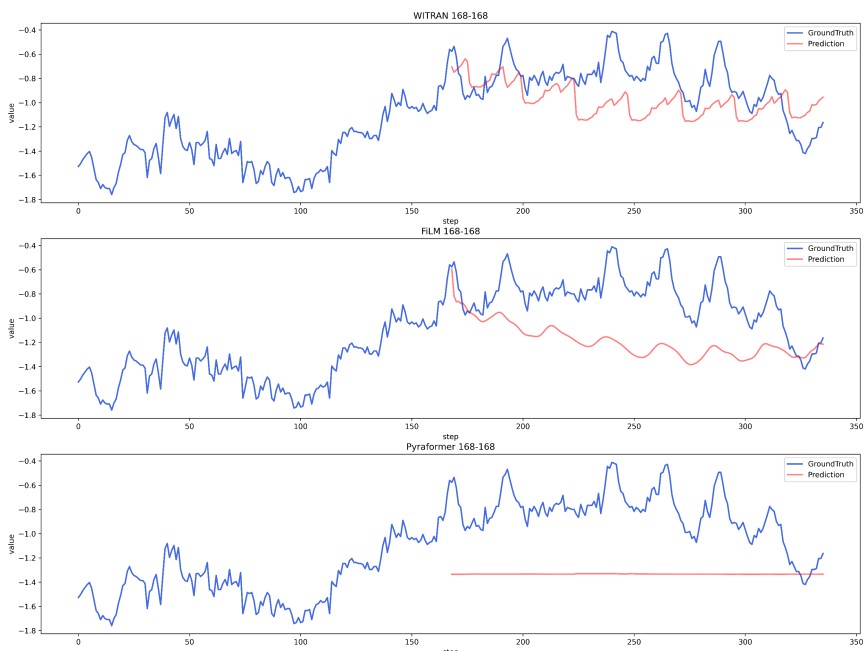

Figure 26: Forecasting cases comparison of WITRAN, FiLM and Pyraformer on the 168-168 task of the ETTh1 dataset.

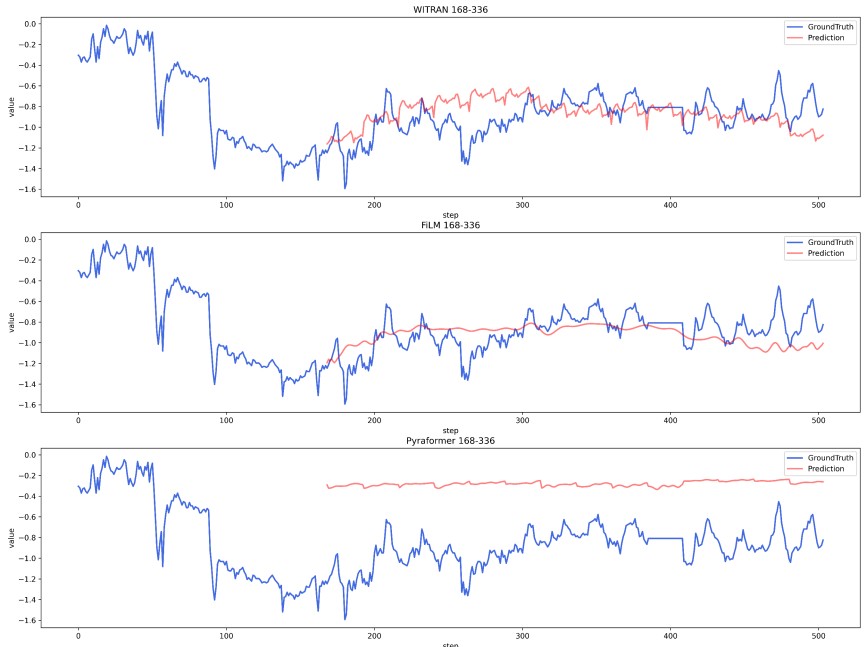

Figure 27: Forecasting cases comparison of WITRAN, FiLM and Pyraformer on the 168-336 task of the ETTh1 dataset.

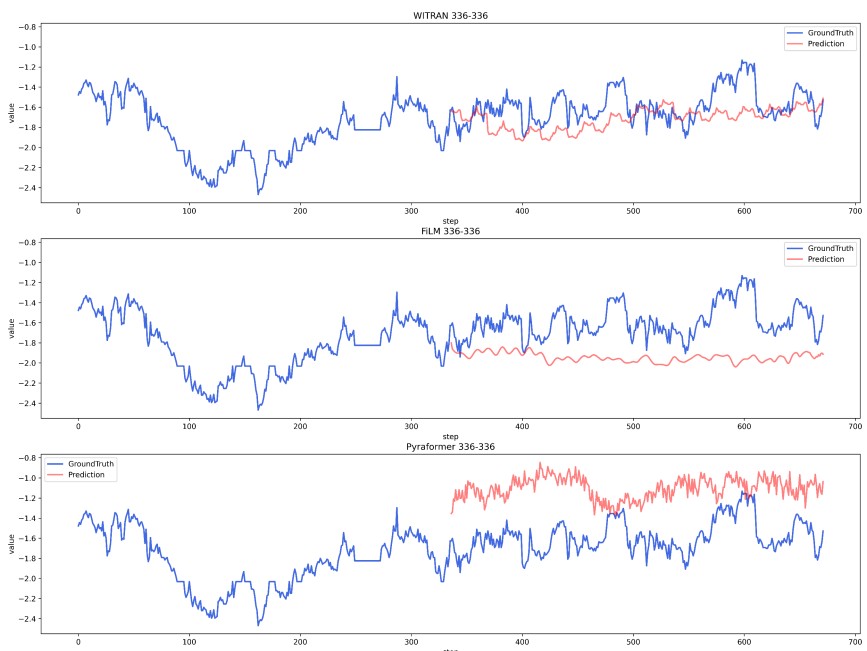

Figure 28: Forecasting cases comparison of WITRAN, FiLM and Pyraformer on the 336-336 task of the ETTh1 dataset.

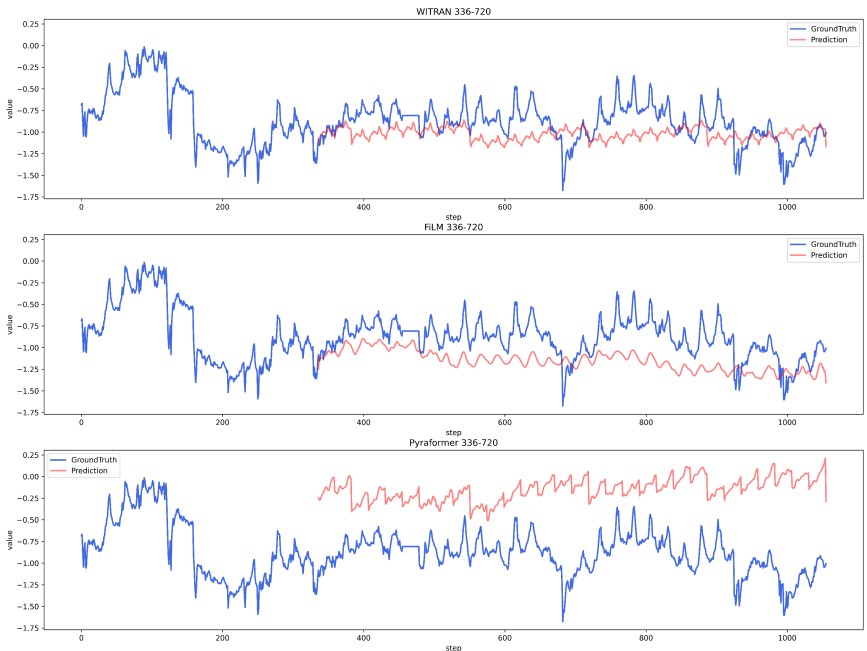

Figure 29: Forecasting cases comparison of WITRAN, FiLM and Pyraformer on the 336-720 task of the ETTh1 dataset.

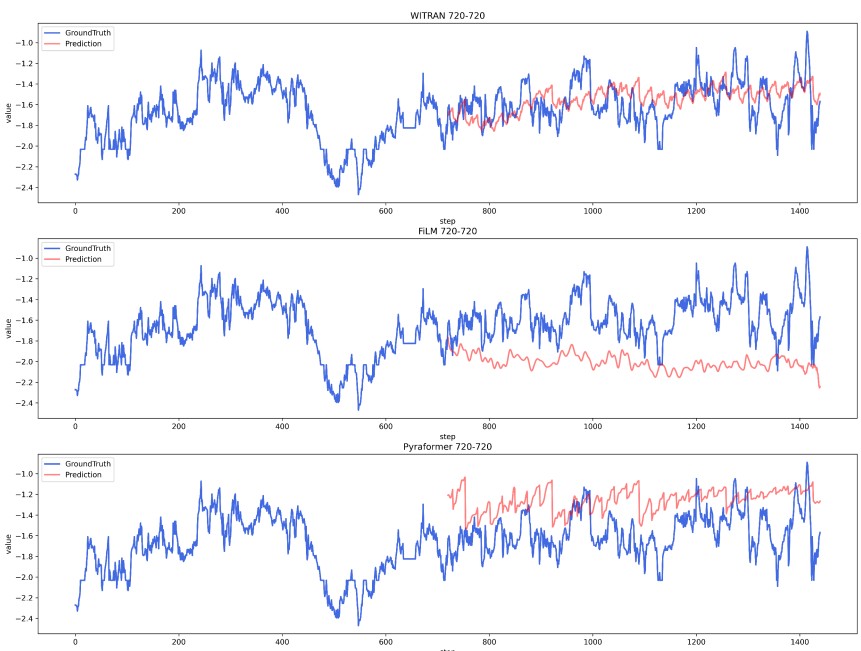

Figure 30: Forecasting cases comparison of WITRAN, FiLM and Pyraformer on the 720-720 task of the ETTh1 dataset.

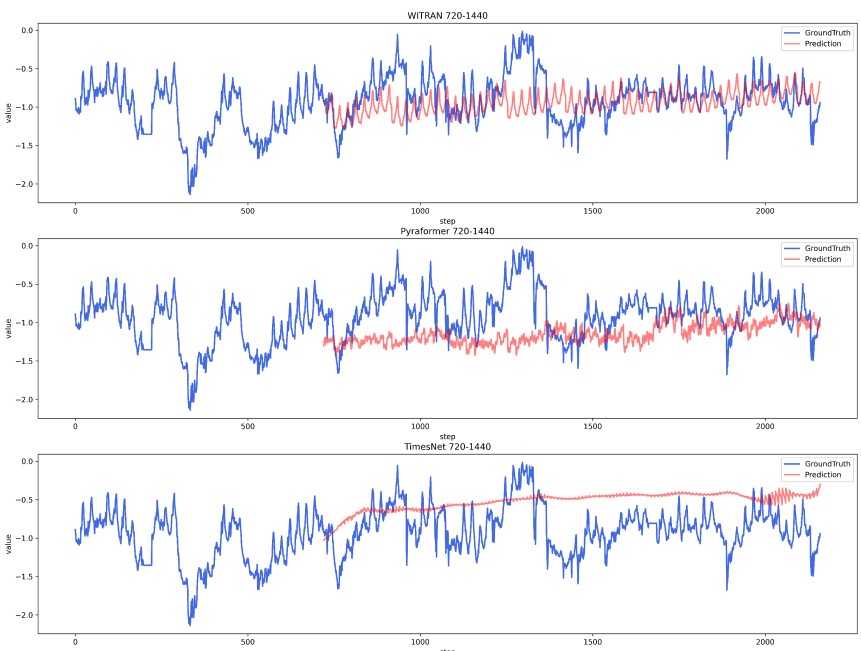

Figure 31: Forecasting cases comparison of WITRAN, Pyraformer and TimesNet on the 720-1440 task of the ETTh1 dataset.

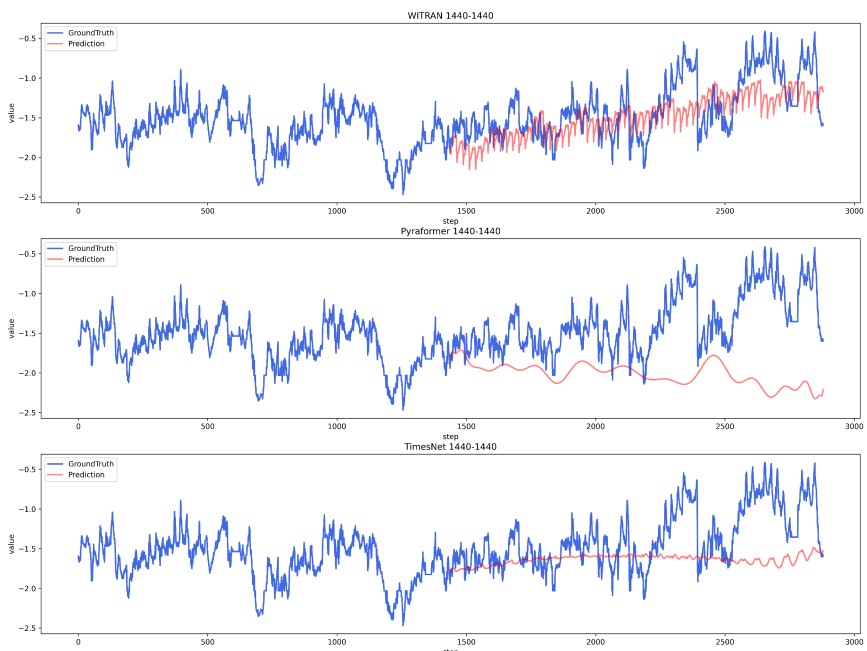

Figure 32: Forecasting cases comparison of WITRAN, Pyraformer and TimesNet on the 1440-1440 task of the ETTh1 dataset.

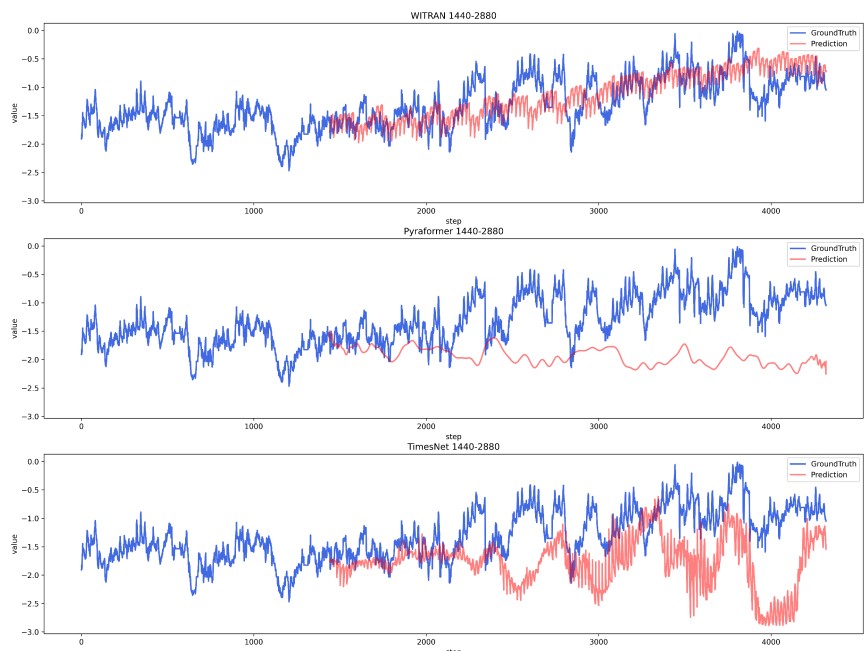

Figure 33: Forecasting cases comparison of WITRAN, Pyraformer and TimesNet on the 1440-2880 task of the ETTh1 dataset.

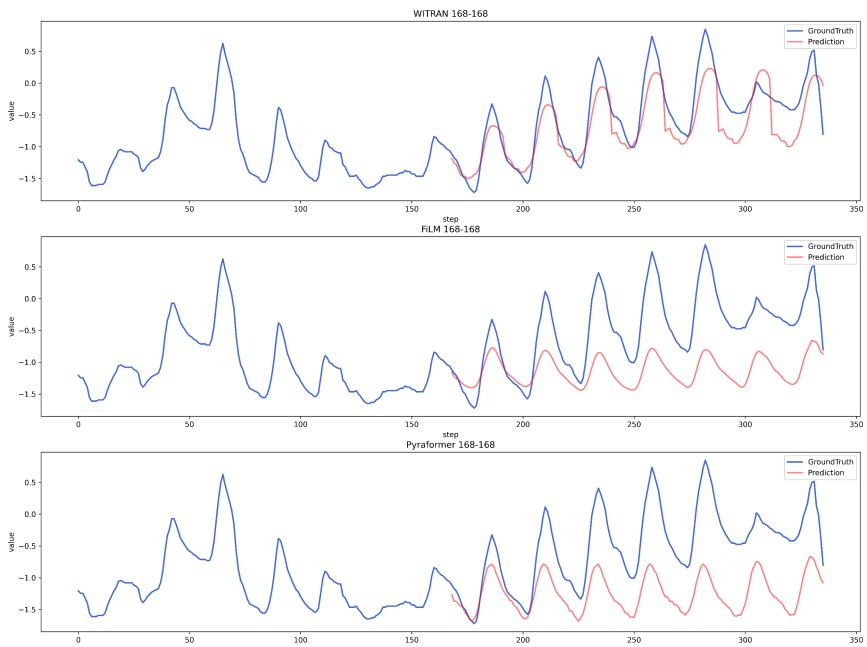

Figure 34: Forecasting cases comparison of WITRAN, FiLM and Pyraformer on the 168-168 task of the ETTh2 dataset.

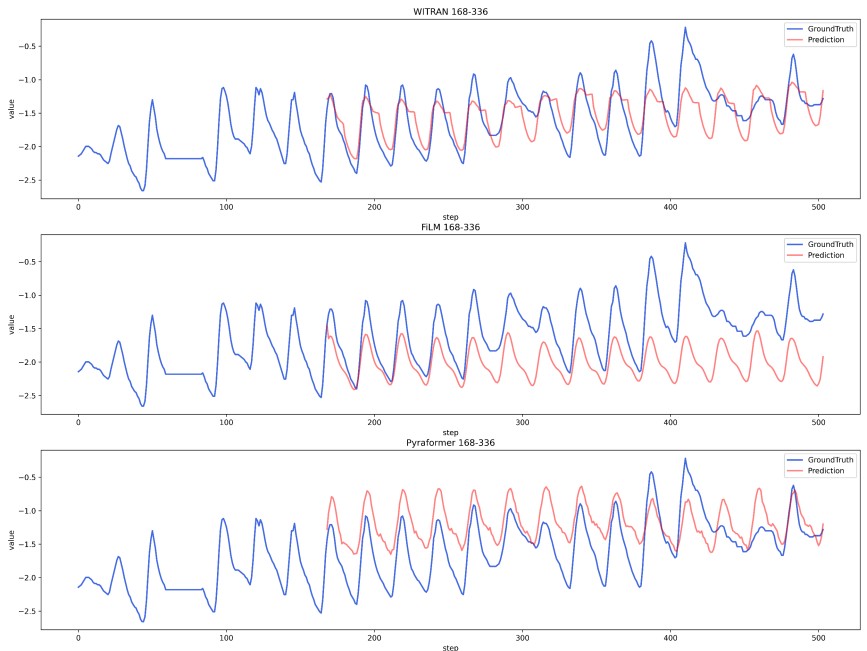

Figure 35: Forecasting cases comparison of WITRAN, FiLM and Pyraformer on the 168-336 task of the ETTh2 dataset.

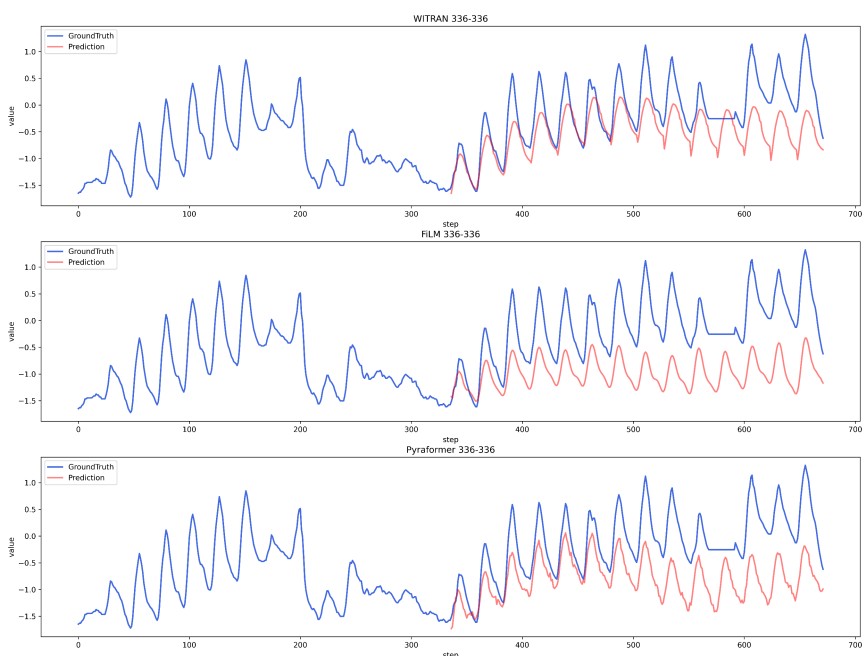

Figure 36: Forecasting cases comparison of WITRAN, FiLM and Pyraformer on the 336-336 task of the ETTh2 dataset.

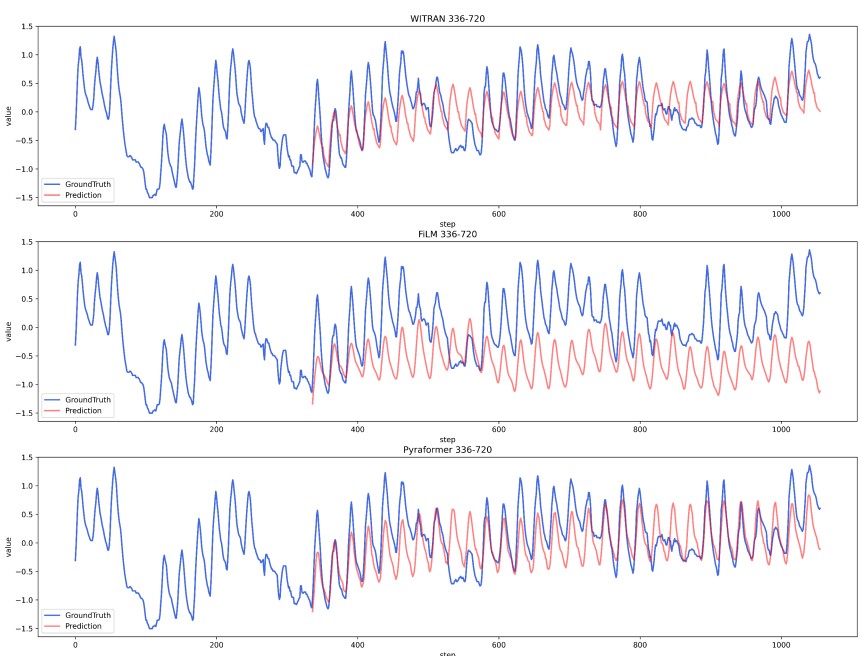

Figure 37: Forecasting cases comparison of WITRAN, FiLM and Pyraformer on the 336-720 task of the ETTh2 dataset.

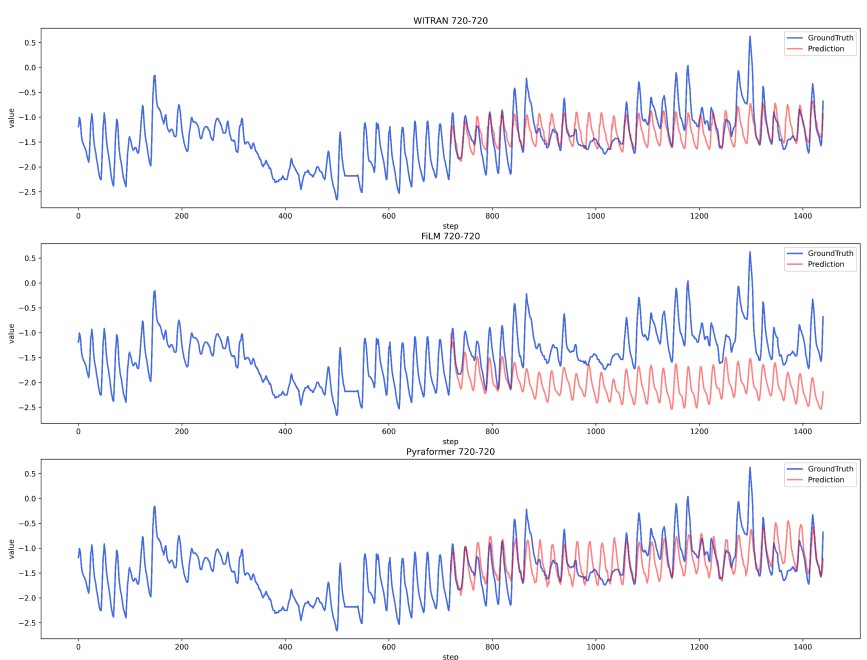

Figure 38: Forecasting cases comparison of WITRAN, FiLM and Pyraformer on the 720-720 task of the ETTh2 dataset.

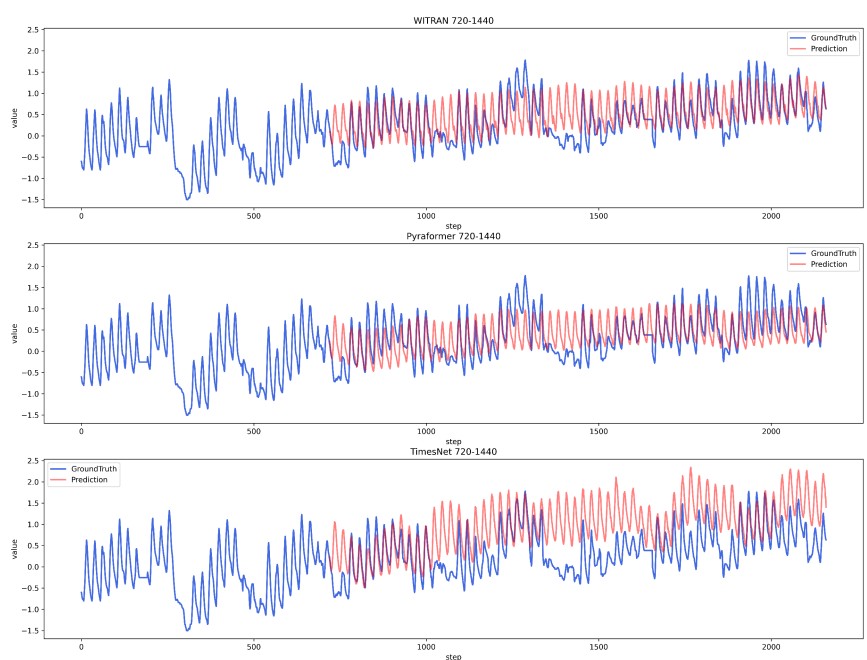

Figure 39: Forecasting cases comparison of WITRAN, Pyraformer and TimesNet on the 720-1440 task of the ETTh2 dataset.

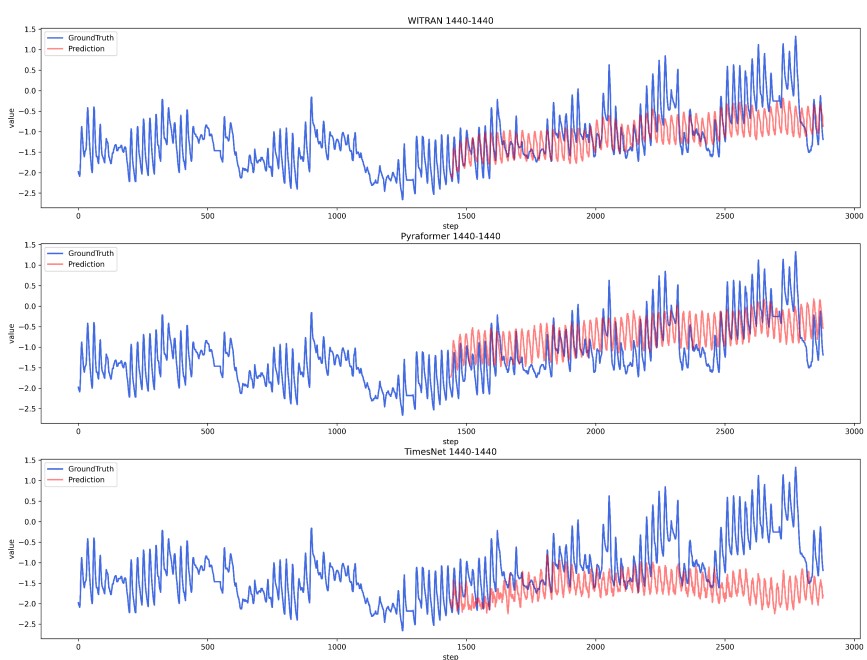

Figure 40: Forecasting cases comparison of WITRAN, Pyraformer and TimesNet on the 1440-1440 task of the ETTh2 dataset.

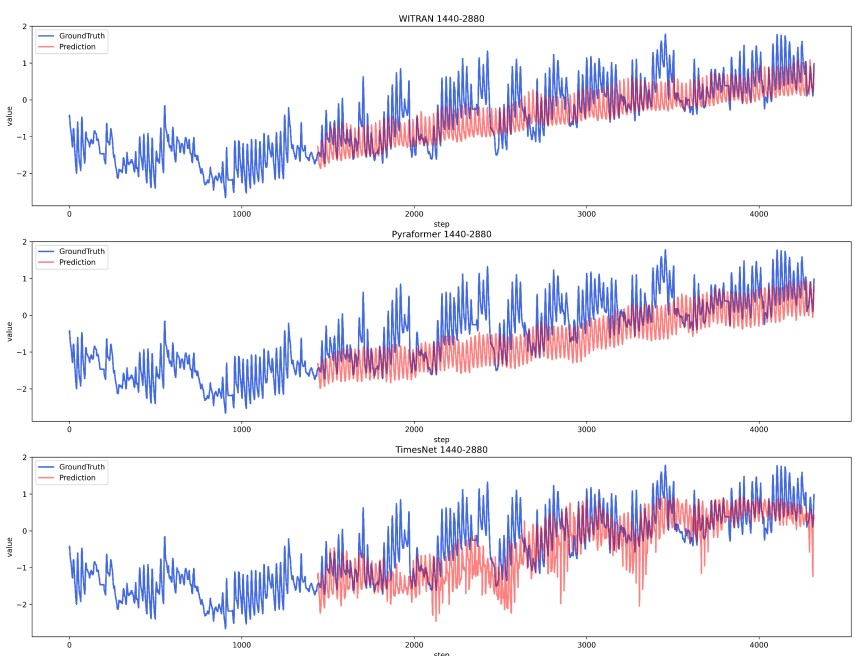

Figure 41: Forecasting cases comparison of WITRAN, Pyraformer and TimesNet on the 1440-2880 task of the ETTh2 dataset.

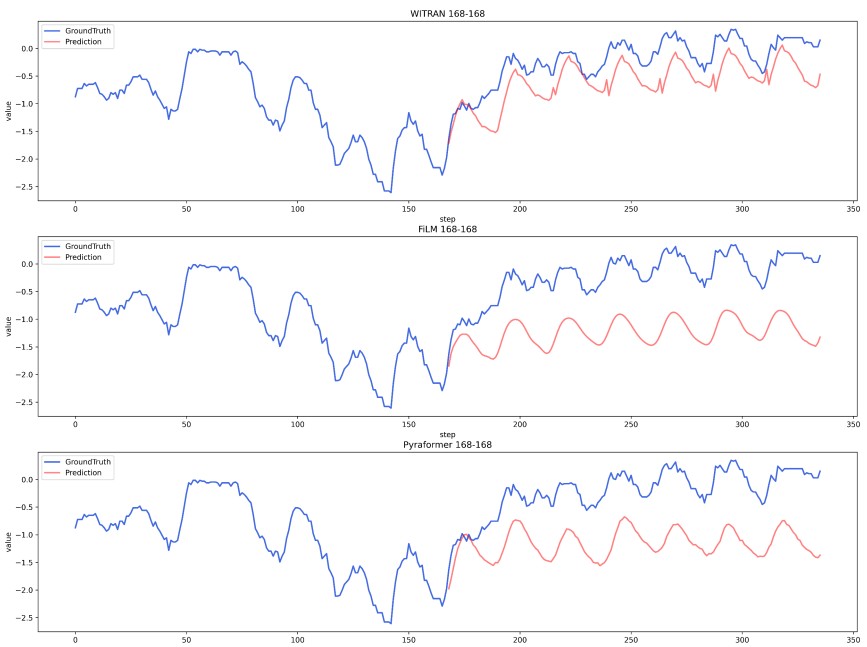

Figure 42: Forecasting cases comparison of WITRAN, FiLM and Pyraformer on the 168-168 task of the Weather dataset.

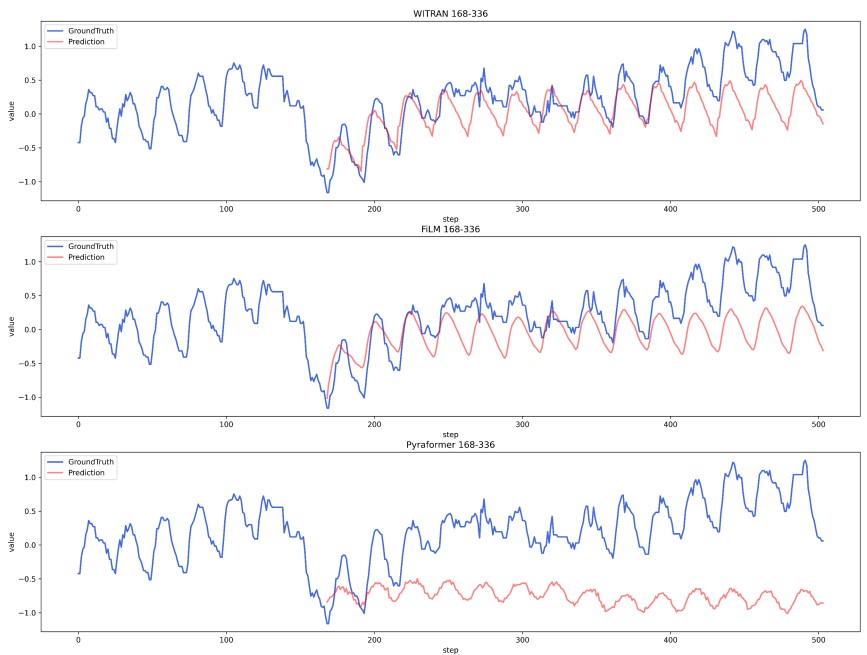

Figure 43: Forecasting cases comparison of WITRAN, FiLM and Pyraformer on the 168-336 task of the Weather dataset.

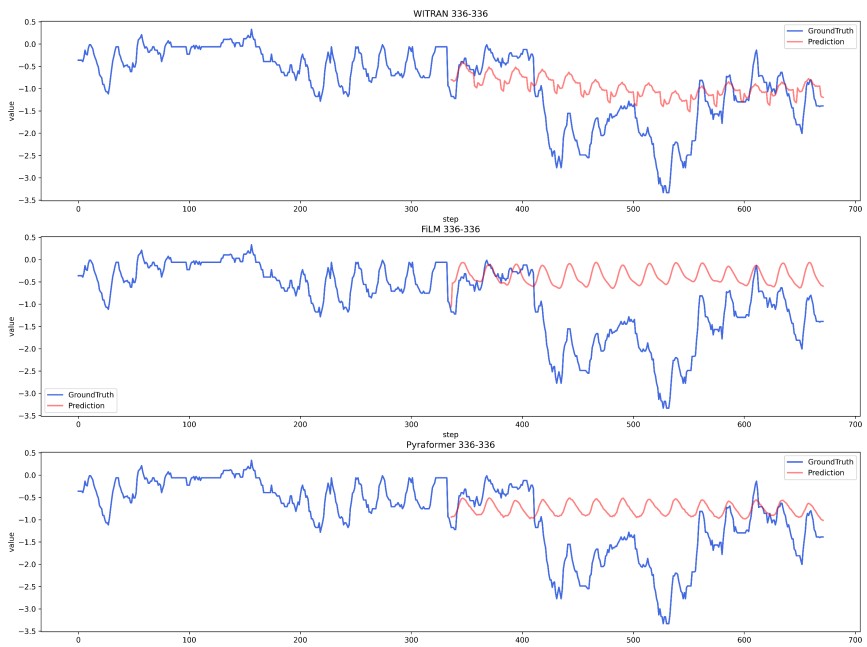

Figure 44: Forecasting cases comparison of WITRAN, FiLM and Pyraformer on the 336-336 task of the Weather dataset.

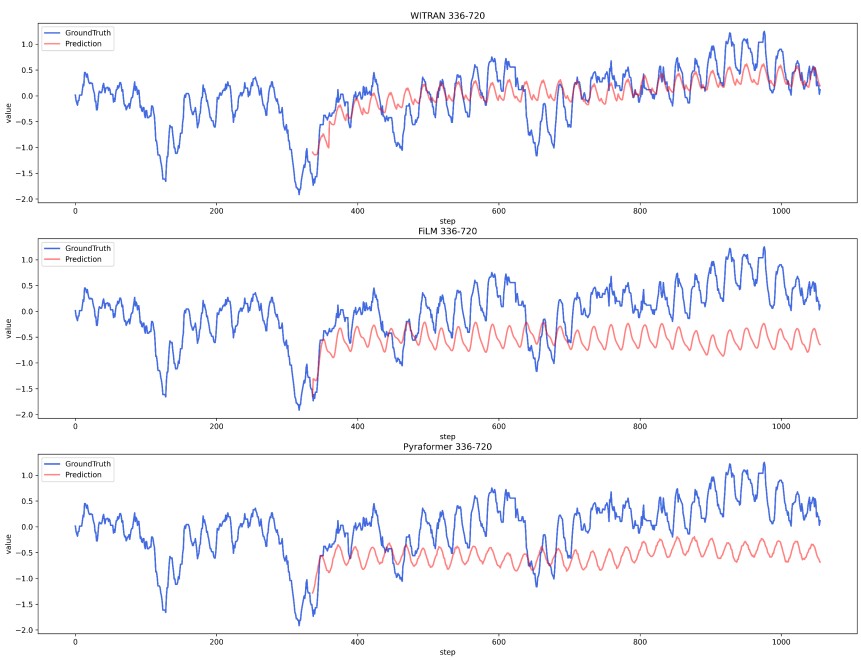

Figure 45: Forecasting cases comparison of WITRAN, FiLM and Pyraformer on the 336-720 task of the Weather dataset.

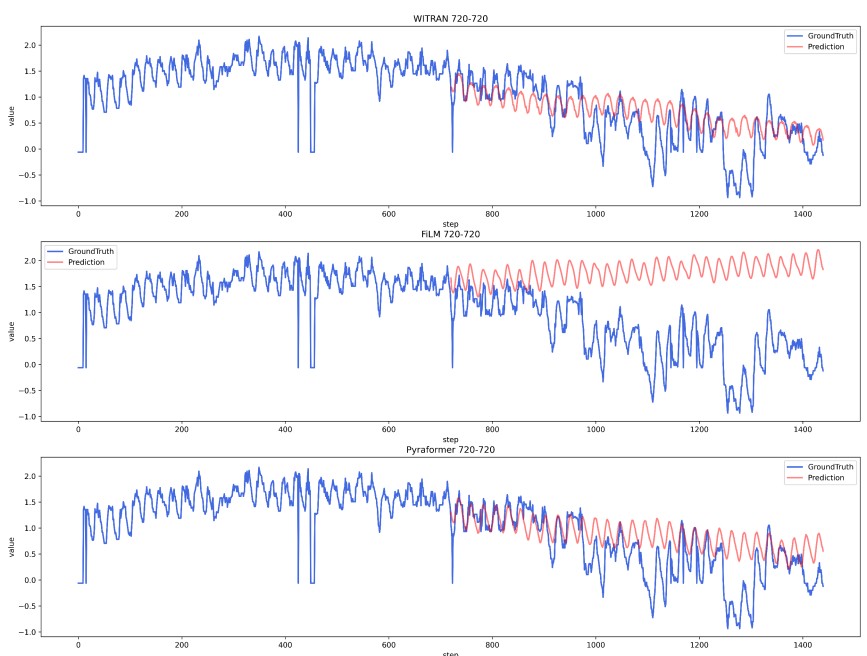

Figure 46: Forecasting cases comparison of WITRAN, FiLM and Pyraformer on the 720-720 task of the Weather dataset.

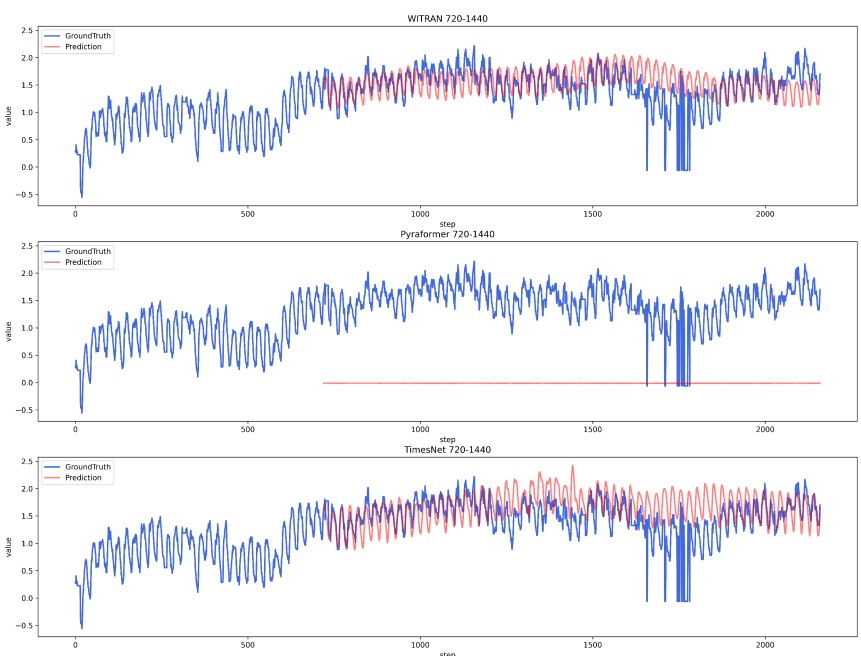

Figure 47: Forecasting cases comparison of WITRAN, Pyraformer and TimesNet on the 720-1440 task of the Weather dataset.

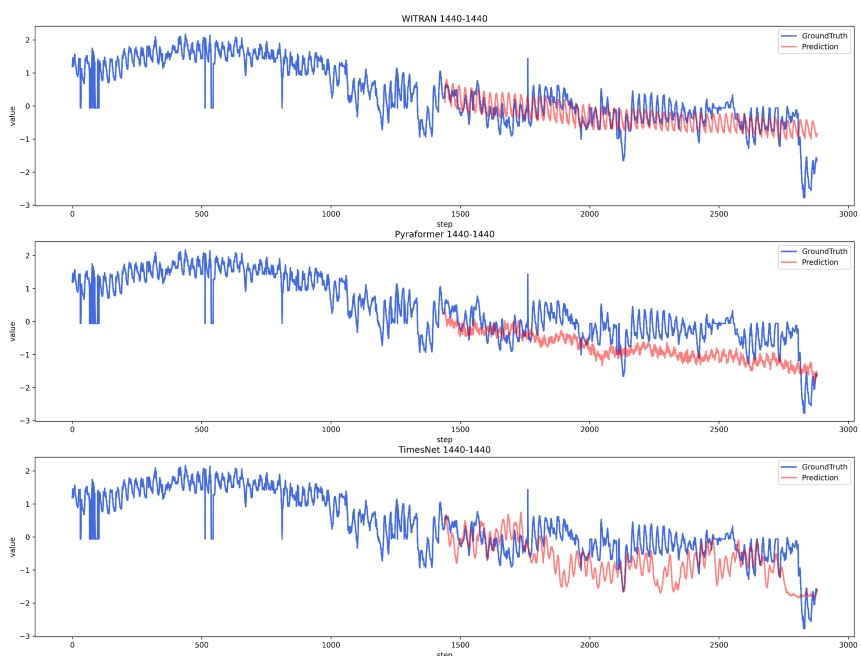

Figure 48: Forecasting cases comparison of WITRAN, Pyraformer and TimesNet on the 1440-1440 task of the Weather dataset.

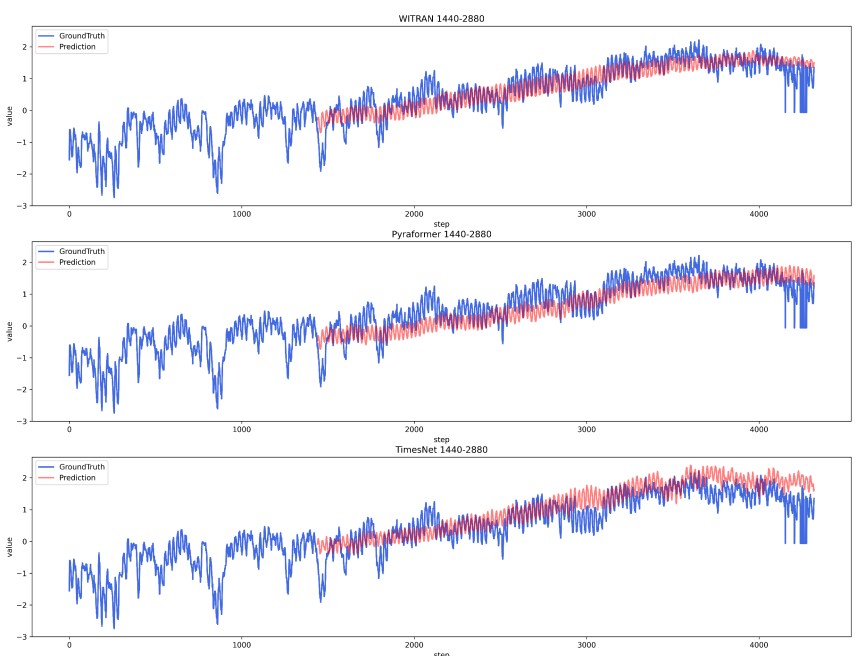

Figure 49: Forecasting cases comparison of WITRAN, Pyraformer and TimesNet on the 1440-2880 task of the Weather dataset.

**Algorithm 3** Recurrent Acceleration Network (RAN)

```python
import torch
import torch.nn as nn
import math
import torch.nn.functional as F

class WITRAN_HVGSU_Encoder(torch.nn.Module):
    def __init__(self, input_size, hidden_size, num_layers, dropout, water_rows, water_cols, res_mode='none'):
        super(WITRAN_HVGSU_Encoder, self).__init__()
        self.input_size = input_size
        self.hidden_size = hidden_size
        self.num_layers = num_layers
        self.dropout = dropout
        self.water_rows = water_rows
        self.water_cols = water_cols
        self.res_mode = res_mode
        # parameter of row cell
        self.W_first_layer = torch.nn.Parameter(torch.empty(6 * hidden_size, input_size + 2 * hidden_size))
        self.W_other_layer = torch.nn.Parameter(torch.empty(num_layers - 1, 6 * hidden_size, 4 * hidden_size))
        self.B = torch.nn.Parameter(torch.empty(num_layers, 6 * hidden_size))
        self.reset_parameters()

    def reset_parameters(self):
        stdv = 1.0 / math.sqrt(self.hidden_size)
        for weight in self.parameters():
            weight.data.uniform_(-stdv, +stdv)

    def linear(self, input, weight, bias, batch_size, slice, Water2sea_slice_num):
        a = F.linear(input, weight)
        if slice < Water2sea_slice_num:
            a[:batch_size * (slice + 1), :] = a[:batch_size * (slice + 1), :] + bias
        return a

    def forward(self, input, batch_size, input_size, flag):
        if flag == 1: # cols > rows
            input = input.permute(2, 0, 1, 3)
        else:
            input = input.permute(1, 0, 2, 3)
        Water2sea_slice_num, _, Original_slice_len, _ = input.shape
        Water2sea_slice_len = Water2sea_slice_num + Original_slice_len - 1
        hidden_slice_row = torch.zeros(Water2sea_slice_num * batch_size, self.hidden_size).to(input.device)
        hidden_slice_col = torch.zeros(Water2sea_slice_num * batch_size, self.hidden_size).to(input.device)
        input_transfer = torch.zeros(Water2sea_slice_num, batch_size, Water2sea_slice_len, input_size).to(input.device)
        for r in range(Water2sea_slice_num):
            input_transfer[r, :, r:r+Original_slice_len, :] = input[r, :, :, :]
        hidden_row_all_list = []
        hidden_col_all_list = []
```

**Algorithm 3** Recurrent Acceleration Network (RAN)

---

**for** layer **in range**(self.num_layers):
    **if** layer == 0:
        a = input_transfer.reshape(Water2sea_slice_num ∗ batch_size, Water2sea_slice_len,
            input_size)
        W = self.W_first_layer
    **else**:
        a = F.dropout(output_all_slice, self.dropout, self.training)
        **if** layer == 1:
            layer0_output = a
        W = self.W_other_layer[layer−1, :, :]
        hidden_slice_row = hidden_slice_row ∗ 0
        hidden_slice_col = hidden_slice_col ∗ 0
    B = self.B[layer, :]
    *# start every for all slice*
    output_all_slice_list = []
    **for slice in range** (Water2sea_slice_len):
        *# gate generate*
        gate = self.linear(torch.cat([hidden_slice_row, hidden_slice_col, a[:, **slice**, :]],
            dim = −1), W, B, batch_size, **slice**, Water2sea_slice_num)
        *# gate*
        sigmod_gate, tanh_gate = torch.split(gate, 4 ∗ self.hidden_size, dim = −1)
        sigmod_gate = torch.sigmoid(sigmod_gate)
        tanh_gate = torch.tanh(tanh_gate)
        update_gate_row, output_gate_row, update_gate_col, output_gate_col = sigmod_gate.chunk
          (4, dim = −1)
        input_gate_row, input_gate_col = tanh_gate.chunk(2, dim = −1)
        *# gate effect*
        hidden_slice_row = torch.tanh(
            (1−update_gate_row)∗hidden_slice_row + update_gate_row∗input_gate_row) ∗
               output_gate_row
        hidden_slice_col = torch.tanh(
            (1−update_gate_col)∗hidden_slice_col + update_gate_col∗input_gate_col) ∗
               output_gate_col
        *# output generate*
        output_slice = torch.cat([hidden_slice_row, hidden_slice_col], dim = −1)
        *# save output*
        output_all_slice_list.append(output_slice)
        *# save row hidden*
        **if slice** >= Original_slice_len − 1:
            need_save_row_loc = **slice** − Original_slice_len + 1
            hidden_row_all_list.append(
               hidden_slice_row[need_save_row_loc∗batch_size:(need_save_row_loc+1)∗
                  batch_size, :])
        *# save col hidden*
        **if slice** >= Water2sea_slice_num − 1:
            hidden_col_all_list.append(
               hidden_slice_col[(Water2sea_slice_num−1)∗batch_size:, :])
        *# hidden transfer*
        hidden_slice_col = torch.roll(hidden_slice_col, shifts=batch_size, dims = 0)
    **if** self.res_mode == 'layer_res' **and** layer >= 1: *# layer−res*
        output_all_slice = torch.stack(output_all_slice_list, dim = 1) + layer0_output
    **else**:
        output_all_slice = torch.stack(output_all_slice_list, dim = 1)
hidden_row_all = torch.stack(hidden_row_all_list, dim = 1)
hidden_col_all = torch.stack(hidden_col_all_list, dim = 1)
hidden_row_all = hidden_row_all.reshape(batch_size, self.num_layers, Water2sea_slice_num,
    hidden_row_all.shape[−1])
hidden_col_all = hidden_col_all.reshape(batch_size, self.num_layers, Original_slice_len,
    hidden_col_all.shape[−1])
**if** flag == 1:
    **return** output_all_slice, hidden_col_all, hidden_row_all
**else**:
    **return** output_all_slice, hidden_row_all, hidden_col_all

---

