# OpenReview forum: "WITRAN: Water-wave Information Transmission and Recurrent Acceleration Network for Long-range Time Series Forecasting"
_NeurIPS.cc/2023/Conference — NeurIPS 2023 spotlight_

### Official Review · Reviewer_2QPt · 2023-07-06

**Soundness:** 3 good
**Presentation:** 3 good
**Contribution:** 3 good
**Rating:** 5
**Confidence:** 4

**Summary:**

To capture semantic information and repetitive patterns concurrently, the authors propose WIT framework. By utilizing bi-granular information transmission and HVGSU, the framework can model the inherent repetitive pattern as well as correlation of time series. The author also use a generic RAN to reduce time complexity. Several experiments conducted by te authors demonstrate that their framework can outperform in time series forecasting.

**Strengths:**

S1: The logic and presentation of the essay is easy to follow.
S2: The problem they focus on is very essential as well as appealing.
S3: Experiments reveal the performance of the authors’ model outperforms existing SOTA baselines.


**Weaknesses:**

W1: The authors may include more experiments such as robustness check to further evaluate the performance of the framework.
W2: There are some typos and minor errors that do not influence the understanding of this work, which should be carefully checked. For instance, line 29, semantic information include-> semantic information includes


**Questions:**

Q1: Can authors include more experiments to further evaluate the performance of the framework.

---

> ### Author Rebuttal · Authors · 2023-08-10
>
> We are sincerely grateful to Reviewer 2QPt for their constructive feedback and recognition of our work.
>
> > Q1: The authors may include more experiments such as robustness check to further evaluate the performance of the framework. Can authors include more experiments to further evaluate the performance of the framework.
>
> Thank you for your suggestion. Here, we have followed MICN and introduced a simple white noise injection to demonstrate the robustness of our model. Specifically, we randomly select a proportion $\varepsilon$ of data from the original input sequence and apply random perturbations within the range $[-2X_{i}, 2X_{i}]$ to the selected data, where $X_{i}$ denotes the original data. After the noise injection, the data is then used for training, and the MSE and MAE metrics are recorded.
>
> Due to space constraints, we have included the table for this section in a separate PDF file. Please refer to Table B in the newly submitted PDF file for specific results.
>
> As the perturbation proportion $\varepsilon$ increases, there is a slight increase in the MSE and MAE metrics in terms of forecasting. It indicates that WITRAN demonstrates good robustness when dealing with less noisy data (up to 10%), and it possesses a significant advantage in effectively handling various abnormal data fluctuations.
>
> > Q2: There are some typos and minor errors that do not influence the understanding of this work, which should be carefully checked. For instance, line 29, semantic information include-> semantic information includes
>
> We greatly appreciate to the reviewer for conducting a meticulous and comprehensive review of our paper, including the appendix. We will conduct a careful check and correct these typos and minor errors in the subsequent versions of the paper.

---

> > ### Comment · Area_Chair_7e9m · 2023-08-18
> > **discussion**
> >
> > Dear Reviewer 2QPt,
> >
> > Thank you for being a reviewer for NeurIPS2023, your service is invaluable to the community!
> >
> > The authors have submitted their feedback.
> >
> > Could you check the rebuttal and other reviewers' comments and start a discussion with the authors and other reviewers?
> >
> > Regards,
> > Your AC

---

### Official Review · Reviewer_Ku6W · 2023-07-07

**Soundness:** 3 good
**Presentation:** 3 good
**Contribution:** 3 good
**Rating:** 7
**Confidence:** 4

**Summary:**

This paper focuses on the long-range time series forecasting problem. An interesting model, Water-wave Information Transmission and Recurrent Acceleration Network is proposed, which captures both short- and long-term recurrent patterns via bi-granular information transmission. The proposed model also captures global and local correlations using horizontal and vertical information transmission. This model is an interesting modification of the RNN network, and it significantly outperforms many transformer-based models.

**Strengths:**

S1. This paper reviews the shortcomings of transformer models and makes modifications to the RNN structure.

S2. A bi-granular information transmission is proposed to capture short- and long-term recurrent patterns, which is easy to understand and does not require additional methods (e.g., FFT) to extract periodicity.

S3. A recurrent acceleration network is proposed, which reduces the time complexity to O(√L) while maintaining the memory complexity at O(L).


**Weaknesses:**

W1. Some previous RNN-based models, such as ConvLSTM [1]/ PredRNN [2]/ PredRNN++ [3], have yet to be compared.

[1] Shi X, Chen Z, Wang H, et al. Convolutional LSTM network: A machine learning approach for precipitation nowcasting[J]. Advances in neural information processing systems, 2015, 28.
[2] Wang Y, Long M, Wang J, et al. Predrnn: Recurrent neural networks for predictive learning using spatiotemporal lstms[J]. Advances in neural information processing systems, 2017, 30.
[3] Wang Y, Gao Z, Long M, et al. Predrnn++: Towards a resolution of the deep-in-time dilemma in spatiotemporal predictive learning[C]//International Conference on Machine Learning. PMLR, 2018: 5123-5132.

W2. Some minor problems:
1) It is recommended to enlarge Figure 1(i) and make horizontal lines clearer.
2) It is recommended to unify TF_{en} and TFE_{de}.


**Questions:**

Q1. Please explicitly explain the advantages of the WITRAN model over other forecasting models from the perspective of information transmission process.

Q2. Please give the adaptation and limitation of the WITRAN model.

---

> ### Author Rebuttal · Authors · 2023-08-10
>
> We sincerely appreciate the valuable comments and recognition provided by Reviewer Ku6W regarding our work.
>
> > Q1: Some previous RNN-based models, such as ConvLSTM / PredRNN / PredRNN++, have yet to be compared.
>
> Thank you for your suggestion. We have supplemented the experimental results for these methods. Due to space limitations, we have included the specific experimental results of this section in the Global Rebuttal. From the experimental results in this section, combined with the findings from Tables 2 and 3, provide further comprehensive evidence that WITRAN outperforms the state-of-the-art methods.
>
> > Q2: Some minor problems: (1) It is recommended to enlarge Figure 1(i) and make horizontal lines clearer. (2) It is recommended to unify TF_{en} and TFE_{de}.
>
> Thanks a lot for reviewing our paper including appendix carefully and comprehensively. We will address the minor problems in the subsequent versions and conduct a more meticulous check of the paper.
>
> > Q3: Please explicitly explain the advantages of the WITRAN model over other forecasting models from the perspective of information transmission process.
>
> For long-range time series forecasting tasks, previous studies have highlighted two crucial aspects of semantic information. On one hand, there is the presence of long-term and short-term periodic semantic information. On the other hand, there is the consideration of local-global semantic information. WITRAN's advantages can be summarized as follows: (1) Through its bi-granular information transmission, has the capability to directly capture long- and short-term periodic semantic information in long-range time series. Additionally, it shortens the information transmission path, effectively solving the issues of gradient explosion/vanishing when dealing with long-range time series. (2) Utilizing its recurrent structure, WITRAN gradually captures local semantic information and integrates it into the global semantic context without the need for additional layers. (3) WITRAN can effectively capture both of the key semantic aspects mentioned in (1) and (2), which was not achievable with previous methods. (4) In WITRAN, RAN could serve as a universal framework for integrating other models to facilitate information fusion and transmission, and reduce the time complexity to $\mathcal{O}(\sqrt{L})$ while maintaining the memory complexity of $\mathcal{O}(L)$.
>
> > Q4: Please give the adaptation and limitation of the WITRAN model.
>
> Through extensive experiments on different datasets and tasks, we have demonstrated the strong performance of WITRAN in various domains, such as energy, traffic, and weather forecasting. The results presented in Tables 2 and 3 indicate that WITRAN performs exceptionally well in long-range and ultra-long-range sequence forecasting tasks, showcasing its adaptability to such scenarios. Additionally, the RAN framework of WITRAN serves as a generic acceleration framework, as effectively illustrated in Section 4.2. However, it is worth noting that WITRAN does have its limitations, as mentioned in the Conclusions section. Specifically, the Python-based implementation of WITRAN is not as efficient as the nn.GRU / nn.LSTM implementations in PyTorch, which are based on C++. Therefore, we plan to explore the integration of WITRAN into an interface using C++.

---

> > ### Comment · Area_Chair_7e9m · 2023-08-18
> > **discussion**
> >
> > Dear Reviewer Ku6W,
> >
> > Thank you for being a reviewer for NeurIPS2023, your service is invaluable to the community!
> >
> > The authors have submitted their feedback.
> >
> > Could you check the rebuttal and other reviewers' comments and start a discussion with the authors and other reviewers?
> >
> > Regards,
> > Your AC

---

> > > ### Comment · Reviewer_Ku6W · 2023-08-20
> > >
> > > The author has addressed all my previous concerns. I will keep my positive rating.

---

> > > > ### Author Response · Authors · 2023-08-20
> > > > **Thanks**
> > > >
> > > > Thank you for your time and positive comments!

---

### Official Review · Reviewer_r3j3 · 2023-07-07

**Soundness:** 2 fair
**Presentation:** 3 good
**Contribution:** 2 fair
**Rating:** 5
**Confidence:** 5

**Summary:**

The paper studies a Water-wave Information Transmission and Recurrent Acceleration Network (WITRAN) framework to model dependencies in a long historical time series. Inspired by Timesnet, the WITRAN introduces a water-wave information transmission strategy to model temporal information. This is implemented by the proposed Horizontal Vertical Gated Selective Unit (a kind of recurrent unit and  similar to GRU).

**Strengths:**

1) This paper is well-written
2) Figures illustrate model details in a good way
3) This paper studies the single-channel case, though most existing papers consider evaluations on multivariate time series. This makes sense as this can better evaluate the ability to learn temporal dependencies in time series.
4) Advanced baselines such as MICN, timesnet, patchTST, and film are included.

**Weaknesses:**

1) Although using the water-wave structure is new to me, I am still curious about why this design is needed in long-range time series forecasting. Especially, are there any special temporal structures that existing models in Figure a-g cannot handle?
2) The water-wave structure is a strategy to model time series dependencies. However, it is unclear how to decide the number of R and C and what are their effects on the final prediction.
3) In Figure 10 (appendix), the results of the proposed are much worse than other baselines: FiLM and Pyraformer. But you claimed that “our model WITRAN gives the best performance among different models” in Section J. Moreover, other plots in Figure 11-31 did not demonstrate the effectiveness of WITRAN.
4) The reference part is missing.

**Questions:**

Questions are as follows:
- Are there any special temporal structures that existing models in Figure a-g cannot handle?
- How to decide the number of R and C
- How to set the norm in (1).

**Limitations:**

Some limitations in terms of implementation have been mentioned in the Conclusion.

---

> ### Author Rebuttal · Authors · 2023-08-10
>
> We express our sincere gratitude to Reviewer r3j3 for their comprehensive review, which included thought-provoking questions and valuable insights.
>
> > Q1: Although using the water-wave structure is new to me, I am still curious about why this design is needed in long-range time series forecasting.
>
> We appreciate your recognition of the novelty of our work. The advantages of applying the water-wave structure to long-range time series forecasting can be summarized as follows:
>
> (1) Bi-granular Design: The water-wave structure employs a bi-granular design, which offers two advantages. Firstly, it shortens the information transmission path, allowing for efficient information flow. Secondly, it enables the exploration of hidden long-term and short-term repetitive patterns in the sequences. These advantages are particularly beneficial for long-range prediction tasks.
>
> (2) Simultaneous Information Transmission: The water-wave structure facilitates simultaneous horizontal and vertical information transmission. This concurrent transmission allows for accelerated processing within the model, thereby improving efficiency.
>
> (3) Global Relationship Modeling: With the water-wave structure, the global relationships in long-range time series can be captured effectively using a single layer, as illustrated in Figure 1. This reduces the complexity of the network and the training difficulty.
>
> By leveraging these advantages, the water-wave structure demonstrates its effectiveness and efficiency in long-range time series forecasting tasks.
>
> > Q2: Are there any special temporal structures that existing models in Figure a-g cannot handle?
>
> Modeling global and local correlations, and discovering long- and short-term repetitive patterns is crucial for accurate long-range time series forecasting. Previous works have partially addressed these issues separately, but have not been able to address all of them simultaneously, such as facing challenges in capturing hidden semantic information directly from point-wise input tokens and so on. In order to compare the differences between WITRAN and the model a-g in Figure 1 more clearly, we have prepared the following table to highlight the advantages of WITRAN.
>
> |                        **Advantages**                       | (a) RNN | (b) CNN | (c) Full Attention | (d) LogTrans | (e) Pyraformer | (f) MICN | (g) PatchTST | (h) TimesNet | (i)WITRAN(ours) |
> |:-----------------------------------------------------------:|:-------:|:-------:|:------------------:|:------------:|:--------------:|:--------:|:------------:|:------------:|:---------------:|
> |           **Non point-wise semantic information capture**           |    ✓    |    ✓    |          ✗         |       ✓      |        ✓       |     ✓    |       ✓      |       ✓      |        ✓        |
> | **Special design to capture long-term repetitive patterns** |    ✗    |    ✗    |          ✗         |       ✗      |        ✓       |     ✗    |       ✗      |       ✓      |        ✓        |
> |       **Efficiently (1 or 2 layers) model global correlations**       |  ✓ (1)  |    ✗    |        ✓ (1)       |       ✗      |        ✗       |   ✓ (2)  |     ✓ (2)    |       ✗      |      ✓ (1)      |
> |      **Well solve the gradient vanishing/exploding problem of RNN**     |    ✗    |    -    |          -         |       -      |        -       |     -    |       -      |       -      |        ✓        |
>
> > Q3: How to decide the number of R and C and what are their effects on the final prediction.
>
> $C$ reflects the periodicity of short-term repetitive patterns, while $R$ reflects the coarse-grained periodicity of long-term repetitive patterns. Therefore, the value of $C$ is related to the specific forecasting task, and we also referred to previous work by setting the value of $C$ as multiples of 12. The value of $R$ is determined by the sequence length $L$ and $C$, specifically $R = L/C$. For more detailed descriptions and experimental analysis, please refer to Section 3.1 and Section I.1.
>
> > Q4: How to set the norm in (1).
>
> The selection of the $norm$ in (1) is adaptively chosen by the model on the validation set based on the characteristics of the dataset and the task. For more details on this aspect, please refer to Section I.1.
>
> > Q5: In Figure 10 (appendix), the results of the proposed are much worse than other baselines: FiLM and Pyraformer. But you claimed that “our model WITRAN gives the best performance among different models” in Section J. Moreover, other plots in Figure 11-31 did not demonstrate the effectiveness of WITRAN.
>
> Thank you for pointing that out. The forecasting case figures in the appendix are segments randomly selected from various tasks of different datasets, aiming to demonstrate a fair overview of WITRAN's performance when combined with Tables 2 and 3. However, random selection may occasionally include segments that are not fully representative, such as the one you specifically mentioned, Figure 10, which can indeed lead to misunderstandings. Therefore, to further demonstrate the advantages of WITRAN, we have re-randomly selected four additional cases for this task. Please refer to the PDF file we have newly submitted for specific details, where the advantages of WITRAN become apparent.
>
> Additionally, we have conducted metric calculations for Figure 11 to Figure 31. Due to the character limit, we have also included this section in the PDF file. Please refer to Table A in the PDF file for the specific results.
>
> From the results in Table A, it can be observed that among all the randomly selected original cases, except for 5 cases, WITRAN consistently demonstrated superior performance. Thus, this provides ample evidence to support the overall superiority of WITRAN.
>
> > Q6: The reference part is missing.
>
> In the original version, we placed the references in the Supplementary Material. In the revision, we will place it in the main content.

---

> > ### Comment · Reviewer_r3j3 · 2023-08-18
> >
> > Thank you for your detailed response. The following are the remaining questions:
> >
> > Since the main contribution is the water-wave WITRAN structure (based on bi-granular Design), I believe **it is related to recurrent neural network variants like dilated RNN [1], sliced RNN [2], and other RNNs with residual/skip connections [3]**. It would be helpful if you could explore and discuss the connections between WITRAN and these related works.
> >
> > [1] Dilated Recurrent Neural Networks (2017)
> >
> > [2] Sliced Recurrent Neural Networks (2018)
> >
> > [3] End-to-end time series imputation via residual short paths  (2018)
> >
> > In Q2, it is mentioned that RNNs lack a specific design to capture long-term repetitive patterns and struggle with the gradient vanishing/exploding problem. What about LSTM/GRU? (I only found a discussion on their time and memory consumption in Appendix C.) LSTM/GRU architectures include cells and gates that aid in capturing both long-term and short-term information. Additionally, is there evidence provided to support that WITRAN can overcome the gradient vanishing/exploding problem of RNNs?
> >
> > Furthermore, in Q2, I believe the claim "Previous works have partially addressed these issues separately, but have not been able to address all of them simultaneously" is inaccurate. Numerous previous time-series prediction works and RNN variants (such as multiscale RNNs) have been developed to incorporate both long-term and short-term information.
> >
> > I have another question about the visualization results (Q5). Since the WITRAN model heavily relies on long-term repetitive patterns, I am unclear about its main advantages compared to methods that utilize periodic patterns or neural basis approximation, such as FiLM or NBeats. For instance, in Figures 29 and 43, for the ETTh1 and Weather datasets (where significant repetitive patterns appear to be absent), WITRAN and FiLM generate very similar predictions.
> >
> > Overall, I thoroughly enjoyed reading this paper, especially due to its bi-granular design and the use of univariate evaluation in the experiments. I believe addressing these questions will further enhance the overall quality of the paper.

---

> > > ### Author Response · Authors · 2023-08-19
> > > **Response to Reviewer r3j3 - Part 1**
> > >
> > > We greatly appreciate your recognition of our work and your suggestions to further enhance the overall quality of the paper. Due to space limitations, we will divide our response into three parts.
> > >
> > > **[Part 1]**
> > >
> > > > Q7: Since the main contribution is the water-wave WITRAN structure (based on bi-granular Design), I believe it is related to recurrent neural network variants like dilated RNN [1], sliced RNN [2], and other RNNs with residual/skip connections [3]. It would be helpful if you could explore and discuss the connections between WITRAN and these related works.
> > > [1] Dilated Recurrent Neural Networks (2017)
> > > [2] Sliced Recurrent Neural Networks (2018)
> > > [3] End-to-end time series imputation via residual short paths (2018)
> > >
> > > We sincerely appreciate your suggestions. We have compared these methods with WITRAN, and here is the comparison:
> > >
> > > |                                    **Advantages**                                    | DilatedRNN [1] | SlicedRNN [2] | RIMP-LSTM [3] | WITRAN (ours) |
> > > |:------------------------------------------------------------------------------------:|:--------------:|:--------:|:-------------:|:-------------:|
> > > |                  **Efficiently (1 layer) model global correlations**                 |        ⍻       |     ⍻    |     ✓ (1)     |     ✓ (1)     |
> > > |              **Special design to capture long-term repetitive patterns**             |        ✓       |     ✗    |       ✗       |       ✓       |
> > > | **Using 1 layer to capture long- and short-term repetitive patterns simultaneously** |        ✗       |     ✗    |       ✗       |       ✓       |
> > > |            **Well solve the gradient vanishing/exploding problem of RNN**            |        ✓       |     ✓    |       ✓       |       ✓       |
> > >
> > > (1) Efficiently (1 layer) model global correlations: (a) When the dilations of DilatedRNN does not include the value 1, multiple layers need to be constructed to extract global correlations. (b) SlicedRNN improves efficiency to some extent by parallel processing of minimum subsequences, but it still requires the introduction of multiple layers to capture the global correlations of the sequence.
> > >
> > > (2) Special design to capture long-term repetitive patterns: (a) SlicedRNN is unable to capture long-term repetitive patterns among elements of sub-sequences. (b) Although RIMP-LSTM incorporates Residual Paths and Residual Sum Unit designs, it still cannot effectively extract long-term repetitive patterns.
> > >
> > > (3) Using 1 layer to capture long- and short-term repetitive patterns simultaneously: (a) DilatedRNN can capture long- and short-term repetitive patterns, but it requires the use of multiple layers to achieve this. (b) SlicedRNN and RIMP-LSTM are not particularly adept at handling long-term repetitive patterns, as mentioned in (2).
> > >
> > > (4) Well solve the gradient vanishing/exploding problem of RNN: (a) In reference [1], the article states and provides formal proof that reducing the length of information paths between time steps can prevent the issues of gradient vanishing/exploding. DilatedRNN, SlicedRNN, and WITRAN tackle this problem by reducing the length of information transmission paths. (b) RIMP-LSTM addresses this issue by the designs of Residual Paths and Residual Sum Units.

---

> > > ### Author Response · Authors · 2023-08-19
> > > **Response to Reviewer r3j3 - Part 2**
> > >
> > > **[Part 2]**
> > >
> > > > Q8: In Q2, it is mentioned that RNNs lack a specific design to capture long-term repetitive patterns and struggle with the gradient vanishing/exploding problem. What about LSTM/GRU? (I only found a discussion on their time and memory consumption in Appendix C.) LSTM/GRU architectures include cells and gates that aid in capturing both long-term and short-term information. Additionally, is there evidence provided to support that WITRAN can overcome the gradient vanishing/exploding problem of RNNs?
> > >
> > > LSTM and GRU, through their gated designs, do help in mitigating the gradient vanishing/exploding problem to some extent. However, previous work has pointed out that, the approach (e.g. LSTM) that relies on a special type of linear unit with a self-connection does not explicitly address the issue of exploding gradients.
> > >
> > > Furthermore, LSTM and GRU are indeed capable of handling sequences of regular-range and capturing both long-term and short-term repetitive patterns within them. However, when applied to long-range forecasting tasks, they can suffer from the issue of information forgetting, and more details can be found in Appendix B.
> > >
> > > Therefore, these two points contribute to the difficulty of applying LSTM/GRU to long-range forecasting tasks.
> > >
> > > Regarding WITRAN's ability to overcome the gradient vanishing/exploding problem, it has been indicated and formally proven in reference [1] (DilatedRNN) that reducing the length of information transmission paths can effectively prevent gradient vanishing/exploding problems. WITRAN addresses the issue of gradient vanishing/exploding by reducing the length of information transmission paths. Furthermore, in order to further validate it experimentally, we conducted experiments on the ETTh1 dataset with 8 tasks to verify the ability of LSTM/GRU in solving gradient vanishing/exploding. The results are shown below:
> > >
> > > |  **Tasks** | **168-168** | **168-336** | **336-336** | **336-720** | **720-720** | **720-1440** | **1440-1440** | **1440-2880** |
> > > |:----------:|:-----------:|:-----------:|:-----------:|:-----------:|:-----------:|:------------:|:-------------:|:-------------:|
> > > |  **LSTM**  |   **6/18**  |   **6/18**  |   **3/18**  |   **3/18**  |     0/18    |   **1/18**   |      0/18     |    **3/18**   |
> > > |   **GRU**  |   **7/18**  |   **6/18**  |   **3/18**  |   **3/18**  |     0/18    |     0/18     |      0/18     |    **6/18**   |
> > > | **WITRAN** |     0/18    |     0/18    |     0/18    |     0/18    |     0/18    |     0/18     |      0/18     |      0/18     |
> > >
> > > The A/B ratio in the table represents the ratio of the total number of parameters with a loss of NaN during training (A) to the total number of parameters in the search space (B). The parameters being searched include $d_\mathrm{model}$ and $e_\mathrm{layer}$. And the tasks in which the loss becomes NaN during the training process have been highlighted in bold.
> > >
> > > From the above table, it can be observed that LSTM and GRU do help to some extent in mitigating the issues of gradient vanishing/exploding. However, they still face challenges in this regard. On the other hand, WITRAN addresses this problem effectively.

---

> > > ### Author Response · Authors · 2023-08-19
> > > **Response to Reviewer r3j3 - Part 3**
> > >
> > > **[Part 3]**
> > >
> > > > Q9: Furthermore, in Q2, I believe the claim "Previous works have partially addressed these issues separately, but have not been able to address all of them simultaneously" is inaccurate. Numerous previous time-series prediction works and RNN variants (such as multiscale RNNs) have been developed to incorporate both long-term and short-term information.
> > >
> > > The "these issues" refer to two aspects: (1) modeling global and local correlations, and (2) discovering long- and short-term repetitive patterns. Indeed, there have been some RNN variants proposed, as mentioned in your previous question (Q6). However, when applied to long-range time series forecasting tasks, they still struggle to effectively address both modeling global and local correlations and discovering long- and short-term repetitive patterns simultaneously:
> > >
> > > (1) DilatedRNN: DilatedRNN is capable of capturing both long-term and short-term repetitive patterns. However, due to its multi-layer design, DilatedRNN still faces challenges in effectively modeling global correlations or capturing long- and short-term repetitive patterns simultaneously.
> > >
> > > (2) SlicedRNN: SlicedRNN has the ability to parallelly process minimum subsequences. However, it still requires the design of multiple layers to capture global correlations. Moreover, it is unable to extract the long-term repeated pattern among elements of sub-sequences.
> > >
> > > (3) RIMP-LSTM: RIMP-LSTM can effectively capture the global correlations through its 1-layer design. However, it still struggles to effectively extract long-term repetitive patterns.
> > >
> > >
> > > > Q10: I have another question about the visualization results (Q5). Since the WITRAN model heavily relies on long-term repetitive patterns, I am unclear about its main advantages compared to methods that utilize periodic patterns or neural basis approximation, such as FiLM or NBeats. For instance, in Figures 29 and 43, for the ETTh1 and Weather datasets (where significant repetitive patterns appear to be absent), WITRAN and FiLM generate very similar predictions.
> > >
> > > We sincerely thank you for your thorough review once again. Capturing long-term repetitive patterns is crucial for long-range forecasting tasks. However, at the same time, capturing short-term repetitive patterns is also crucial for the accuracy of the forecasting.
> > >
> > > WITRAN relies on both long-term and short-term repetitive patterns in historical sequences for capturing periodic semantic information. In particular, as you mentioned in Figures 29 and 43, both WITRAN and FiLM are capable of forecasting a periodic fluctuation of peaks and valleys. However, FiLM's forecasting tend to be smoother, while WITRAN excels in predicting the upward/downward fluctuations present (high frequency data) in short-term periods. This further attests to the advantages of WITRAN.

---

> > > > ### Comment · Reviewer_r3j3 · 2023-08-21
> > > >
> > > > Thank you for your kind response. My concerns have been addressed. I'll happily move my score to 5. I hope the above discussions can be included in your final version.

---

> > > > > ### Author Response · Authors · 2023-08-21
> > > > > **Thanks**
> > > > >
> > > > > Thank you once again for your time and suggestions. We will include above discussions in the final version.

---

### Official Review · Reviewer_1moF · 2023-07-07

**Soundness:** 3 good
**Presentation:** 3 good
**Contribution:** 3 good
**Rating:** 8
**Confidence:** 5

**Summary:**

This paper studies the problem of long-range time series forecasting problem and proposes a WITRAN model. The paper analyzes and compares previous forecasting methods from the perspective of information transmission process and design a water-wave information transmission mechanism, which simultaneously capture global and local correlations via a bi-granular information transmission. And a recurrent acceleration network is designed to reduce the computation complexity.

**Strengths:**

1) WITRAN is an ingeniously crafted framework, capable of simultaneously capturing two aspects of semantic information within long-range time series - global-local correlations and both long- and short-term periodic patterns. The WIT includes two innovative modules, namely, HVGSU and GSU, which augment forecasting accuracy while enhancing explainability. Additionally, the RAN segment is designed to markedly improve efficiency. Finally, the WIT and RAN segments are seamlessly integrated together.

2) The paper presents an exhaustive theoretical substantiation of the RAN segment's operating efficiency, shedding light on its capacity to significantly boost model efficiency.

3) WITRAN exhibits superior performance, as confirmed by a series of comprehensive experiments with equitable settings across all baselines. WITRAN outperforms SOTA methods in both long-range and ultra-long-range forecasting tasks. Moreover, the incorporation of RAN indeed enhances computational efficiency and reduces memory footprint, in alignment with theoretical proof.

**Weaknesses:**

1) It seems that the proposed model is suitable for the time series that naturally contains bi-granular periodicity, such as traffic flow with daily and weekly periodicity. So the generalization of the proposed model is unclear.

2) In the experimental section, many Transformer-based methods are taken as baselines, but few RNN-based methods are not compared. Since the proposed model is RNN-based, so it is necessary to add more RNN-based baselines.


**Questions:**

1) If the five datasets have enough representativeness among all kinds of time series?

2) The material included in Appendix A appears to be rather elementary for the target audience in this domain. I would be intrigued to understand the rationale behind the authors' decision to incorporate this content.


**Limitations:**

See weakness.

---

> ### Author Rebuttal · Authors · 2023-08-10
>
> We extend our sincere appreciation to Reviewer 1moF for their valuable insights and for recognizing the significance of our research.
>
> > Q1: It seems that the proposed model is suitable for the time series that naturally contains bi-granular periodicity, such as traffic flow with daily and weekly periodicity. So the generalization of the proposed model is unclear.
>
> WITRAN effectively captures both long- and short-term repetitive patterns through bi-granular information transmission. As long as the dataset contains periodicity and coarser-grained periodicity, WITRAN can effectively capture them. Our extensive experiments have demonstrated that WITRAN achieves outstanding performance in various fields, such as energy, traffic, and weather. Additionally, the RAN framework in WITRAN can serve as a generic framework for acceleration, highlighting its generality.
>
> > Q2: In the experimental section, many Transformer-based methods are taken as baselines, but few RNN-based methods are not compared. Since the proposed model is RNN-based, so it is necessary to add more RNN-based baselines.
>
> Thank you very much for your valuable suggestions. We have included several classic RNN-based methods, including convLSTM, PredRNN, and predRNN++. Due to space limitations, we have included the specific experimental results of this section in the Global Rebuttal. From the experimental results in this section and the results in Tables 2 and 3 provide further evidence that our method is optimal.
>
> > Q3: If the five datasets have enough representativeness among all kinds of time series?
>
> The benchmark datasets we used covers various domains and have different levels of granularity in their original collection. The previous works, including the baseline methods we selected, mostly use these datasets. Therefore, these datasets should have sufficient representativeness in covering time series.
>
> > Q4: The material included in Appendix A appears to be rather elementary for the target audience in this domain. I would be intrigued to understand the rationale behind the authors' decision to incorporate this content.
>
> This section's detailed description of LSTM and GRU does indeed seem somewhat elementary. Our main intention is twofold: (1) to provide a foundation in Appendix B, where we elaborate on the inspiration drawn from LSTM and GRU, and (2) to facilitate a clear comparison between our proposed HVGSU and the differences between LSTM and GRU for the convenience of readers.

---

> > ### Comment · Area_Chair_7e9m · 2023-08-18
> > **discussion**
> >
> > Dear Reviewer 1moF,
> >
> > Thank you for being a reviewer for NeurIPS2023, your service is invaluable to the community!
> >
> > The authors have submitted their feedback.
> >
> > Could you check the rebuttal and other reviewers' comments and start a discussion with the authors and other reviewers?
> >
> > Regards,
> > Your AC

---

> > ### Comment · Reviewer_1moF · 2023-08-19
> >
> > The authors have addressed all my previous concerns.
> > I will keep my score as 8.

---

> > > ### Author Response · Authors · 2023-08-19
> > > **Thanks**
> > >
> > > Thanks for your time and positive comments!

---

### Author Rebuttal · Authors · 2023-08-10

We thank the reviewers for their careful reading, and detailed and considerate feedback.

# 1 The Supplementary Baseline Experimental Results

Due to space limitations, we will report the results of the baseline experiments we conducted in this section. We have included several classic RNN-based methods, including convLSTM, PredRNN, and PredRNN++. The performance of these methods is as follows:

| Dataset |    Task   | ConvLSTM |         | PredRNN |         | PredRNN++ |         |
|:-------:|:---------:|:---------:|:---------:|--------:|:---------:|:----------:|:---------:|
|         |           |    MSE   |   MAE   |   MSE   |   MAE   |    MSE    |   MAE   |
|   ECL   |  168-168  |  1.10061 | 0.86419 | 1.59968 | 0.97815 |  1.05324  | 0.81715 |
|         |  168-336  |  0.95128 | 0.79262 |  1.3981 | 0.92003 |  1.14599  | 0.84769 |
|         |  336-336  |  1.06283 | 0.84397 | 1.24618 | 0.86953 |  1.14628  | 0.84583 |
|         |  336-720  |  1.02897 | 0.83296 | 1.38004 | 0.88513 |  1.12283  | 0.84775 |
|         |  720-720  |  1.16781 | 0.89208 | 1.14053 | 0.81585 |   1.0375  | 0.81916 |
|         |  720-1440 |  1.19013 | 0.90671 | 1.38331 | 0.91472 |     -     |    -    |
|         | 1440-1440 |  1.08107 | 0.86057 | 1.22349 | 0.88084 |     -     |    -    |
|         | 1440-2880 |  1.12110  | 0.86696 | 1.23118 | 0.84768 |     -     |    -    |

| Dataset |    Task   | ConvLSTM |         | PredRNN |         | PredRNN++ |         |
|:-------:|:---------:|:---------:|:---------:|--------:|:---------:|:----------:|:---------:|
|         |           |    MSE   |   MAE   |   MSE   |   MAE   |    MSE    |   MAE   |
| traffic |  168-168  |  1.9568  | 1.20513 | 2.18789 | 1.26295 |  2.03032  | 1.22555 |
|         |  168-336  |  1.56089 | 1.00843 | 1.94531 | 1.19838 |  2.15307  | 1.25684 |
|         |  336-336  |  1.53403 |  1.0047 | 1.94237 | 1.20825 |  2.01507  | 1.22229 |
|         |  336-720  |  1.97317 |  1.1942 | 2.10299 | 1.23738 |  2.07428  | 1.23589 |
|         |  720-720  |  1.97934 | 1.19838 | 2.24516 | 1.27154 |  2.05511  | 1.23179 |
|         |  720-1440 |  1.94060  | 1.17002 | 2.12212 | 1.23488 |     -     |    -    |
|         | 1440-1440 |  2.05290  |  1.2179 | 3.43876 | 1.49211 |     -     |    -    |
|         | 1440-2880 |  2.11828 | 1.24669 | 1.94494 | 1.20526 |     -     |    -    |

| Dataset |    Task   | ConvLSTM      |    | PredRNN      |    | PredRNN++      |    |
|:-------:|:---------:|:---------:|:---------:|--------:|:---------:|:----------:|:---------:|
|         |           |    MSE   |   MAE   |   MSE   |   MAE   |    MSE    |   MAE   |
|  ETTh1  |  168-168  |  1.11838 | 0.95673 | 0.20783 | 0.36933 |  0.19706  | 0.35199 |
|         |  168-336  |  0.42289 | 0.53454 | 0.60663 |  0.6639 |  0.18579  | 0.34186 |
|         |  336-336  |  0.28599 | 0.43453 | 0.90852 | 0.83825 |  0.23011  | 0.38086 |
|         |  336-720  |  0.18302 | 0.35023 | 0.29477 | 0.44067 |  0.28792  | 0.41855 |
|         |  720-720  |  0.21331 | 0.36852 | 0.92726 | 0.87761 |  0.34586  | 0.56421 |
|         |  720-1440 |  0.26016 | 0.40767 |  0.2097 | 0.35739 |     -     |    -    |
|         | 1440-1440 |  0.46201 | 0.55664 | 0.25375 | 0.39662 |     -     |    -    |
|         | 1440-2880 |  0.68657 | 0.73995 |    -    |    -    |     -     |    -    |

| Dataset |    Task   |      ConvLSTM |    |      PredRNN |    |      PredRNN++ |    |
|:-------:|:---------:|:---------:|:---------:|--------:|:---------:|:----------:|:---------:|
|         |           | MSE      | MAE     | MSE     | MAE     | MSE       | MAE     |
|  ETTh2  |  168-168  |  0.67911 | 0.67424 | 2.71814 | 1.24722 |  0.46365  | 0.53826 |
|         |  168-336  |  0.57245 |  0.6036 |  7.0023 | 1.94492 |  0.50444  | 0.56156 |
|         |  336-336  |  0.43214 | 0.52404 | 1.21141 | 0.90117 |  0.53269  | 0.57922 |
|         |  336-720  |  0.44047 | 0.52043 | 0.77007 | 0.71523 |  0.59645  | 0.61303 |
|         |  720-720  |  0.61122 | 0.61653 | 0.68664 | 0.66567 |  0.65089  | 0.65267 |
|         |  720-1440 |  0.68432 | 0.67343 |  0.5682 | 0.63012 |  0.52903  | 0.59123 |
|         | 1440-1440 |  0.69629 | 0.67345 | 0.84719 | 0.77302 |     -     |    -    |
|         | 1440-2880 |  0.49551 | 0.57418 | 0.81936 |  0.7682 |     -     |    -    |

| Dataset |    Task   | ConvLSTM |         | PredRNN |         | PredRNN++ |         |
|:-------:|:---------:|:---------:|:---------:|--------:|:---------:|:----------:|:---------:|
|         |           | MSE      | MAE     | MSE     | MAE     | MSE       | MAE     |
| WTH     | 168-168   | 0.23746  | 0.37102 | 0.70020  | 0.66288 | 0.32711   | 0.43675 |
|         | 168-336   | 0.27096  | 0.40337 | 0.96749 | 0.77150  | 0.40391   | 0.48547 |
|         | 336-336   | 0.30900    | 0.42290  | 0.77884 | 0.70297 | 0.40797   | 0.49392 |
|         | 336-720   | 0.39303  | 0.49367 | 0.77279 | 0.70146 | 0.50707   | 0.56308 |
|         | 720-720   | 0.40034  | 0.47207 | 0.72229 | 0.68889 | 0.61416   | 0.65392 |
|         | 720-1440  | 0.46196  | 0.55477 | 0.61551 | 0.6625  | -         | -       |
|         | 1440-1440 | 0.39681  | 0.48828 | 0.7735  | 0.74801 | -         | -       |
|         | 1440-2880 | 0.45973  | 0.54664 | -       | -       | -         | -       |

Among them, "-" represents the cases of gradient explosion/vanishing. Specifically, during the training process, the loss value may become NaN.

# 2 The Supplementary Forecasting Cases

We randomly selected an additional four forecasting cases to compare the performance of WITRAN, FiLM, and Pyraformer on the 168-168 task of the ECL dataset. For more details, please refer to Figure A in the newly submitted PDF.

# 3 The MAE Metric of Figure 11 to Figure 31

We have calculated the metrics for different methods in each case, more details can be found in Table A in the PDF.

# 4 The Supplementary Robustness Experiments

We have included the table for this section in a separate PDF file. Please refer to Table B in the PDF for specific results.

---

### Decision · Program_Chairs · 2023-09-21

**Decision:**

Accept (spotlight)

**Comment:**

All reviewers gave this paper a positive rating. All but one reviewer were satisfied with the authors' feedback. The concerns of reviewer 2QPt, who did not respond, appear to have been adequately addressed by the authors. Based on the above, the ACs consider it appropriate to accept the paper. The ACs recommend that the feedback and discussion be included in the final version.